# Biomechanical control of vascular morphogenesis by the surrounding stiffness

Yasuyuki Hanada[1,2,3], Semanti Halder[1], Yuichiro Arima [2], Misato Haruta[1], Honami Ogoh[1], Shuntaro Ogura[4], Yukihiko Shiraki[4], Sota Nakano[1], Yuka Ozeki[1], Shigetomo Fukuhara [5], Akiyoshi Uemura[4], Toyoaki Murohara [3] & Koichi Nishiyama [1,2,6] ✉

Sprouting angiogenesis is a form of morphogenesis which expands vascular networks from preexisting networks. However, the precise mechanism governing efficient branch elongation driven by directional movement of endothelial cells (ECs), while the lumen develops under the influence of blood inflow, remains unknown. Herein, we show perivascular stiffening to be a major factor that integrates branch elongation and lumen development. The lumen expansion seen during lumen development inhibits directional EC movement driving branch elongation. This process is counter-regulated by the presence of pericytes, which induces perivascular stiffening by promoting the deposition of EC-derived collagen-IV (Col-IV) on the vascular basement membrane (VBM), thereby preventing excessive lumen expansion. Furthermore, inhibition of forward directional movement of the tip EC during lumen development is associated with decreased localization of the F-BAR proteins and Arp2/3 complexes at the leading front. Our results demonstrate how ECs elongate branches, while the lumen develops, by properly building the surrounding physical environment in coordination with pericytes during angiogenesis.

The tissues of living organisms have complex but functional shapes and structures. The individual components such as cord-like, branching and lumen structures are formed through distinct morphogenetic cell behaviors, including collective cell movement, in response to internal and external signals[1–4]. The individual morphogenetic stages are interrelated and, ultimately, are integrated into different tissue morphologies. These facts raise essential questions that include whether or not there are common principles underlying the integration of distinct morphogenetic stages and how the individual stages simultaneously proceed while affecting each other at cellular and multicellular levels.

Sprouting angiogenesis is a form of morphogenesis which expands vascular networks from the preexisting ones in both developmental and postnatal settings[5–7]. During this process, vascular endothelial cells (ECs) collectively elongate the cord-like structure forming new branches in coordination with perivascular pericytes. Almost simultaneously, the lumen structure forms de novo in the newly established branch, followed by circumferential expansion of the lumen with the inflow of blood[8,9]. Efficient extension of a cord-like structure during branch elongation is driven by the directional movement of ECs[7,10,11]. Therein, individual ECs become polarized along the antero-posterior axis[12–15], and then move forward and backward,

[1]Laboratory for Vascular and Cellular Dynamics, Department of Medical Sciences, University of Miyazaki, Miyazaki, Miyazaki, Japan. [2]International Research Center for Medical Sciences, Kumamoto University, Kumamoto, Kumamoto, Japan. [3]Department of Cardiology, Nagoya University Graduate School of Medicine, Nagoya, Aichi, Japan. [4]Department of Retinal Vascular Biology, Nagoya City University Graduate School of Medical Sciences, Nagoya, Aichi, Japan. [5]Department of Molecular Pathophysiology, Institute for Advanced Medical Sciences, Nippon Medical School, 1-1-5 Sendagi, Bunkyo-ku, Tokyo, Japan. [6]The Frontier Research Center, University of Miyazaki, Miyazaki, Miyazaki, Japan. ✉e-mail: koichi_nishiyama@med.miyazaki-u.ac.jp

and also stop cell-autonomously and via communications with neighboring cells, though all eventually move toward the direction of branch elongation[10,11,16]. However, the mechanisms underlying EC-promoted elongation of branches via their directional movement in conjunction with lumen development during angiogenic morphogenesis, are still largely unknown.

In addition to biochemical regulations by various molecules including VEGF (vascular endothelial cell growth factor)[5–7,17], angiogenic morphogenetic processes are biomechanically controlled by physical properties of the surrounding environment such as blood flow and the extracellular matrix (ECM)[18–21]. Blood flow reportedly causes mechanical stimiuli[18,19], among which pressure and wall stretch are assumed to be loaded onto the vascular wall due to the blind-end structure of the tip of the angiogenic branch. Thereby, intraluminal pressure expands the lumen, resulting in vascular wall stretching which in turn generates wall tension, which is then counter-regulated by the stiffness of the surrounding ECM[22]. We recently identified intraluminal pressure derived from blood flow as being a negative regulator of angiogenic branch elongation, an effect induced by lumen expansion[12]. Ectopic and excessive lumen expansion followed by vascular wall stretching, induced by intraluminal pressure loading, resulted in failed formation of antero-posterior cell polarity, thereby impairing directional EC movement and branch elongation[12]. Furthermore, we identified Fes/Cdc42 interacting protein-4 (CIP4) homology-Bin/Amphiphysin/Rvs (F-BAR) proteins[23], CIP4 and Transducer of Cdc42 dependent actin assembly-1 (TOCA1), as key regulators mediating the actin-related protein 2/3 (Arp2/3) complex-dependent-actin polymerization required for directional EC migration during angiogenesis[12]. In addition, blood flow-derived intraluminal pressure was reported to be essential for promoting lumen development via inverse membrane blebbing[24]. These observations suggest the existence of a mechanism that integrates branch elongation and lumen development by further controlling these distinct actions of intraluminal pressure. This mechanism would presumably promote proper angiogenic morphogenesis in physiological settings seen during development. However, this possibility has not yet been fully addressed, at least in part because of methodological difficulties in vivo such as direct observations of the involved physical properties and experimental perturbation of these properties.

To tackle these issues, we reconstituted angiogenic morphogenesis and its surrounding physical environment in vitro, enabling us to directly monitor and intervene in the processes involved, utilizing a unique microfluidic system. This approach combining in vitro on-chip reconstitutions and in vivo validations showed perivascular stiffness to be a major driver that integrally allows branch elongation and lumen development to proceed during angiogenic morphogenesis. We found the lumen and branch expansions seen during lumen development to inhibit directional tip EC movement, driving branch elongation, which was counter-regulated by perivascular stiffening which restricts lumen expansion in the presence of pericytes. We further demonstrated enhanced deposition of EC-derived type-IV collagen (Col-IV) on vascular basement membranes (VBM) to account for spatiotemporal perivascular stiffening by pericytes. Finally, we demonstrated that decreased localization of the F-BAR proteins and Arp2/3 complexes at the leading front might be mechanistically involved in inhibiting the forward directional movement of tip ECs during lumen development.

## Results

### Visualization of the dynamics of branch elongation and lumen development
To explore the relationship between branch elongation and lumen development during angiogenic morphogenesis, we visualized and examined these dynamic processes, using a unique on-chip assay system with a microfluidic device[12] (Supplementary Fig. 1a). This assay system can mimic sprouting angiogenesis of ECs three-dimensionally.

ECs collectively sprout into the ECM and form a vessel-like structure having cell-cell junctions composed of VE-cadherin as well as branching and anastomotic structures (Fig. 1a and Supplementary Fig. 1b, c). Furthermore, this system reproduces development of the vascular lumen (Fig. 1a–c). Time-lapse imaging of on-chip angiogenesis demonstrated a dynamic process in which the de novo vascular lumen appeared within an elongating branch, followed by circumferential growth of the branch with lumen expansion (Fig. 1a, b and Supplementary Movie 1).

### Relationship between branch elongation and lumen development during angiogenic morphogenesis of ECs
We analyzed the relationship between branch elongation and lumen development, in terms of temporal dynamics, employing time-lapse imaging of our on-chip angiogenesis assay of ECs. We identified two patterns of de novo lumen development around the tip (Fig. 1d). One pattern involves the pre-existing proximal lumen extending toward the tip of the branch and expanding circumferentially; this was morphologically characterized by a tapered lumen shape toward the distal tip (Proximal-to-Distal extension) (Supplementary Fig. 2a and Supplementary Movie 2). The other pattern showed a large vacuole-like structure initially emerging around the tip of the branch and then becoming connected to the pre-existing lumen (Distal-to-Proximal fusion) (Supplementary Fig. 2b and Supplementary Movie 3), which was documented 9 times in 8 branches during the 63-h period of observation. Two similar lumen tip morphology patterns were observed in murine retinal angiogenesis, showing a tapered lumen shape toward the distal tip (50% of 122 observed vessel tips, Supplementary Fig. 2c) as well as vacuole-like structures positive for ICAM-2, an apical membrane marker of the vascular lumen, around the tips of the angiogenic branches which had not yet connected to the proximal lumen (11%, Supplementary Fig. 2d), findings consistent with those described in an earlier report[24].

We tracked tip ECs contributing to de novo lumen formation with the nucleus serving as a landmark on time-lapse movies. Simultaneously, we traced changes in the nearby lumen area of the tracked tip ECs. Mosaic analysis with a small population of genetically GFP-prelabeled HUVECs (GFP-HUVECs) showed ECs with anterior-posterior cell polarity to have cell body lengths, from the center of the nucleus to either the front or the rear edge of an elongating angiogenic branch, that ranged from 27 to 205 μm (Supplementary Fig. 3a, b). Therefore, one analyzed lumen area was set as the range within 25 μm from the center of the nucleus of the tracked tip ECs to both the proximal and the distal side (Fig. 1e), the site at which the lumen is assumed to be composed of the tracked tip ECs. It is noteworthy that we found directional forward movement of tip ECs, which also reflects branch elongation, to decelerate immediately after lumen expansion, regardless of whether the lumen emerged from the distal or the proximal side (Fig. 1f, and Supplementary Fig. 3c and Supplementary Movie 4, 5), whereas the lumen did not expand, instead narrowing, in tip ECs maintaining forward movement without deceleration (Fig. 1g and Supplementary Movie 6). A quantitative analysis showed that, in majority of cases, lumens emerged or expanded 30 min prior to deceleration during forward directional movement of tip ECs, while, in clear contrast, this was not seen in tip ECs without deceleration (Fig. 1h, i). These results strongly suggest an inverse relationship between branch elongation and lumen development, i.e., that lumen expansion possibly inhibits both forward directional movement of nearby tip ECs and branch elongation during angiogenic morphogenesis.

### Inhibitory actions of lumen and branch expansions on EC movement and branch elongation
To verify the possible causal relationship between branch elongation and lumen development, we investigated changes in the dynamics of directional EC movement and branch elongation when the degrees of

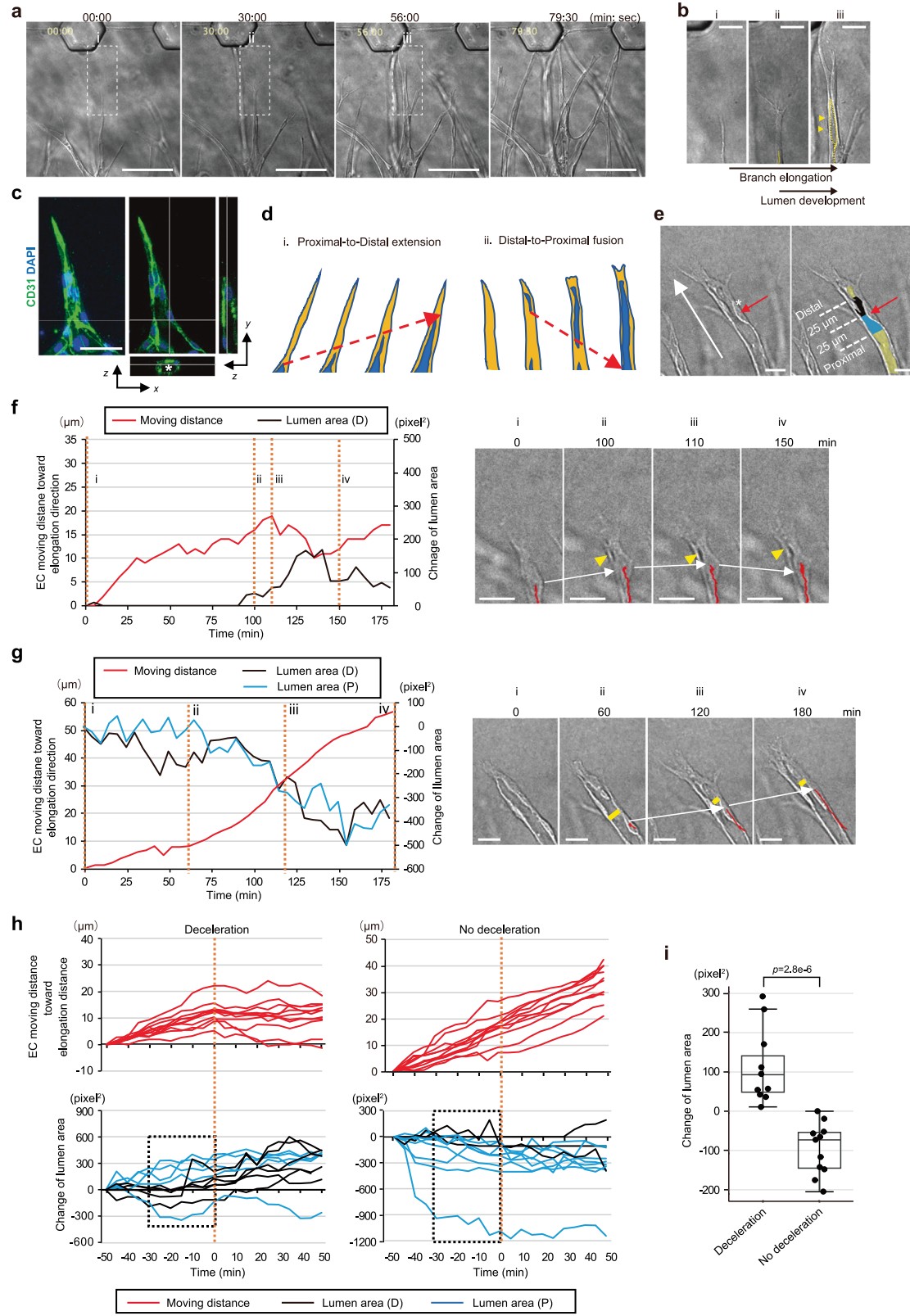

lumen and branch expansions were externally altered in different ways, in the on-chip angiogenesis of ECs. Micro-vessel (lumen) diameters are generally determined by the balance between intraluminal and extraluminal pressures and by the stiffness of the vessel wall itself as well as that of the surrounding ECM. If intraluminal pressure is increased ($\Delta P$) with no other parameter changing, the micro-vessel expands circumferentially (Fig. 2a).

Thus, to intentionally restrict branch and lumen expansions in the on-chip angiogenesis assay, we first increased the stiffness of ECM by treatment with Transglutaminase (TG)[25,26], a crosslinking enzyme of fibrin gels, without changing the biochemical compositions of the on-chip assay of ECs. To confirm the increase in stiffness of fibrin-collagen gel in response to TG treatment, local viscoelasticity was measured. For the measurements, the rheological method known as particle-

**Fig. 1 | Relationship between branch elongation and lumen development during on-chip angiogenic morphogenesis of ECs. a, b,** Time lapse imaging of on-chip angiogenesis of ECs, shown in DIC images at the elapsed times indicated at the top (**a**, See also Supplementary Movie 1, a representative data from more than 3 independent experiments) and in magnified DIC images of squares with dotted boundaries in **a** (**b**). Yellow dotted lines and arrows indicate the lumen. **c**, Representative confocal z-projection (left) and its orthogonal views (right) showing vascular lumen structures (white asterisk) in the angiogenic branch (from more than 3 independent experiments). **d**, Schematic representations of two different patterns of de novo vascular lumen formation in an angiogenic branch (See also Supplementary Fig. 2). **e–i** Evaluation of relationships in dynamics between lumen development and tip EC movement in the elongating angiogenic branch. **e** Setting of analyzed lumen area, shown in DIC images of an angiogenic branch, in which an example of a tracked cell (white asterisk) and the analyzed lumen areas were superimposed. Lumen areas (yellow) at the distal (black) and proximal (blue) sides relative to the center of the nucleus of the tracked cell (red arrows). White arrow indicates the direction of branch elongation. **f, g** Kymographs of tip EC movement (red) and lumen area (distal area (D) in black, proximal area (P) in blue) in an elongating branch (left) and DIC images of elongating branches with trajectory (red line) of the tip EC displacement at timepoints indicated in numbers

(orange dotted line) on the kymograph (right): a representative pattern with deceleration of tip EC movement (white arrows, line iii indicates deceleration timepoint) when the lumen which had developed by Distal-to-Proximal fusion expands (yellow arrowheads) (**f**, See also Supplementary Movie 4) and a representative pattern without deceleration of tip EC movement (white arrows) during Proximal-to-Distal lumen extension, with concurrent development of the lumen without circumferential expansion (yellow bars) (**g**, See also Supplementary Movie 6). **h** Kymographs showing relationships in dynamics between tip EC movement (top) and lumen area (bottom) for 50 min before and after the deceleration timepoint (0 min, orange dotted line) in the presence and the absence of deceleration of forward tip EC movements (deceleration: $n = 11$; no deceleration: $n = 11$, from 3 independent experiments). As to changes of lumen area, data are shown for predominantly expanded proximal (P) or distal (D) lumens. **i** Box and individual plots of changes in the lumen area for 30 min prior to the deceleration timepoint (the dotted boxes in **h**). Box plots show the interquartile range, with the middle line defining the median, and whiskers show the minimum and maximum values, excluding outliers. Outlier values are defined as being 1.5 times the interquartile range above and below the third and first quartile, respectively. Scale bars: 20 μm (**b**), 25 μm (**e, f, g**), 50 μm (**c**) and 100 μm (**a**). Two-sided Mann-Whitney U test (**i**). Source data are provided as a Source Data file.

tracking microrheology (PTM)[27] was incorporated into the on-chip angiogenesis assay. For PTM, fluorescent beads with a diameter of 1 μm were embedded randomly in the fibrin-collagen gel before the induction of angiogenic morphogenesis and the thermal fluctuation of each bead was then measured (Fig. 2b). The degree of this fluctuation, which is denoted as the Creep compliance index, reflects the local stiffness of the gel just around the measured bead (Fig. 2b). Lower values of the index reflect greater gel stiffness. The PTM analysis showed Creep compliance indices to decrease, in response to a rise in the fibrinogen level in the gel (Fig. 2c), which would be expected to increase local stiffness. The data obtained suggested the PTM method to be applicable to evaluating ECM stiffness around the measured bead in the assay system. The PTM analysis showed Creep compliance indices to decrease in response to treatment with TG (Fig. 2c). We also conducted a vascular distensibility test to verify whether TG treatment inhibits lumen and branch expansions in response to intraluminal pressure loading. For this test, 60 Pa (~ 0.45 mmHg) of additional hydrostatic pressure ($\Delta P$) was loaded onto the intraluminal vascular wall (Supplementary Fig. 1d). This pressure loading caused lumen and branch expansions around the tip ECs without TG treatment (13.6 ± 5.5 % increase in branch diameter), whereas the branches rarely expanded in response to the same pressure loading in the ECM with TG treatment (−0.52 ± 2.5 % increase in branch diameter) (Fig. 2d), showing reduced vessel distensibility due to increased ECM stiffness. The on-chip angiogenesis assay using the stiffened ECM treated with TG showed angiogenic branches to elongate more efficiently, maintaining smaller branch and lumen diameters around the leading edge where the de novo lumen develops (evaluated by the index termed Tip expandability), as compared to those with the non-treated ECM (Fig. 2e–g).

We further analyzed tip EC movement changes by reducing lumen expansion. Time-lapse imaging showed tip ECs to frequently slow or even halt after moving forward for a period of time. This move forward and stop movement pattern was repeated during branch elongation in on-chip angiogenesis of ECs using the non-treated ECM (Fig. 2h and Supplementary Fig. 4). In contrast, when the ECM was treated with TG, most of the tip ECs tended to continue moving forward, as characterized by a higher frequency of forward movement with greater speed, such that the aforementioned movement pattern was suppressed (Fig. 2h, i and Supplementary Fig. 4, See also Supplementary Movie 7). However, the sizes of nearby lumens remained essentially constant, though occasional decreases were seen, regardless of whether the lumen was distal or proximal to the nuclei, and expansion was rarely noted (Fig. 2j–l and Supplementary Fig. 5), in clear contrast to the condition without TG treatment (Fig. 2l). These results indicate that restriction of excessive lumen expansion promotes branch

elongation with a higher frequency of forward directional movement of tip ECs.

Next, we ectopically altered the expansion states of both lumen and branch by increasing intraluminal pressure externally and then restoring it. To achieve this, additional hydrostatic pressure ($\Delta P = 41 \pm 14$ Pa) with culture media was loaded into the intraluminal spaces of elongating branches and it was then released 3 h after the loading in on-chip angiogenesis of ECs, and, in some cases, the pressure load-release cycle was repeated (Fig. 3a). Time-lapse imaging data and its quantification confirmed the lumen and branch to expand and shrink, respectively, in immediate response to loading and release of the additional intraluminal pressure (Fig. 3b, c). We found that, consistent with our previous results[12], branches ceased elongating or even retracted (Fig. 3d, and left panel of Fig. 3e), accompanied by abrupt loss of forward directional movement of the tip ECs, just after additional intraluminal pressure loading (Fig. 3f, g, See also Supplementary Movie 8). On the other hand, in control on-chip angiogenesis without pressure loading, forward directional movement of tip ECs leading to branch elongation was essentially maintained over the same observation period (right panel of Fig. 3e and Supplementary Fig. 6, See also Supplementary Movie 9). Notably, in response to release of the loaded pressure, most tip ECs gradually started to move forward again at least within 100 min after the release and thereby contributed to branch elongation (Fig. 3d, h, i, left panel of Fig. 3e, See also Supplementary Movie 8). Similar dynamic changes in directional tip EC movement and branch elongation in response to external loading and release of intraluminal pressure could be repeatedly reproduced in the second round of the pressure load-release cycle, although the number of branches in which this could be confirmed was methodologically limited (Supplementary Fig. 7, See also Supplementary Movie 10). These results clearly indicate that ectopic lumen and branch expansions can inhibit branch elongation driven by forward directional EC movement.

Finally, we attempted to induce lumen and branch expansions by pharmacological interventions. It was previously reported that, in both epithelial and endothelial tubulogenesis, expansion of the de novo lumen is limited by apical actomyosin contractility via Ras homologue gene family member A (RhoA) - Rho-associated protein kinase (ROCK) −non-muscle myosin II (NMII) signaling[24,28–30]. Therefore, we blocked the RhoA-ROCK-NMII signaling using an inhibitor of ROCK, Y27632, and an inhibitor of NMII, Blebbistatin, from one day after induction of on-chip angiogenesis of ECs, a time point when most angiogenic branches had sprouted into the ECM. Inhibition of the ROCK induced branch and lumen expansions around the tip (Fig. 3j, k, right panel of Fig. 3l), which was accompanied by disappearance of phosphorylated

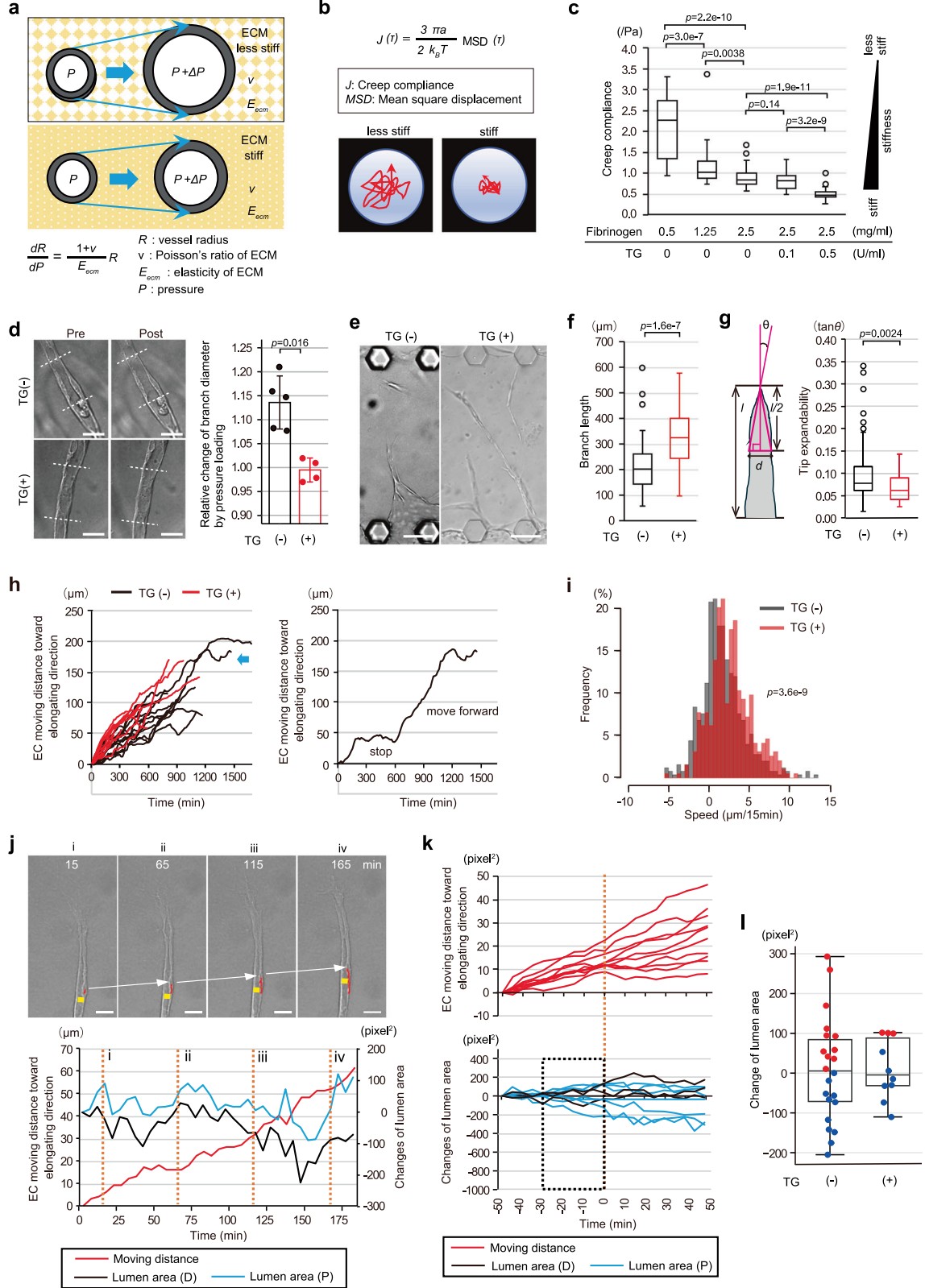

myosin light chain 2 (pMLC2), a parameter of activated NMII (Fig. 3k and Supplementary Fig. 8), and reduced branch elongation (Fig. 3j, left panel of Fig. 3l). Furthermore, it should be noted that the reduction in branch elongation as an effect of the ROCK inhibitor was rescued by physically restricting branch and lumen expansions via stiffening of the surrounding ECM by TG treatment (Fig. 3j, l). Similar effects were also obtained with NMII inhibitor treatment (Fig. 3j, m), indicating

branch and lumen expansions to possibly serve a negative regulatory function as regards branch elongation, although the involvement of a direct inhibitory effect of both inhibitors on directional EC movement cannot be ruled out. Collectively, these findings strongly suggest a causal link between lumen development and branch elongation, in which lumen and branch expansions inhibit forward directional movement of the tip ECs, resulting in retardation of branch elongation.

**Fig. 2 | Perivascular stiffening suppresses lumen and branch expansions and enhances tip EC movement and angiogenic branch elongation. a** Schematic illustrations showing branch diameter changes in response to intraluminal pressure loading ($\Delta P$) depending on different stiffnesses of the perivascular environment. The formula indicates relationships among parameters contributing to vascular distensibility. **b** The PTM method. Creep compliance ($J$), a parameter for quantifying the stiffness of the gel, was calculated from the tracking data of pre-implanted fluorescent beads. **c** Direct measurement of the stiffnesses of fibrin-collagen gel with various concentrations of fibrinogen and evaluation of the stiffening of gels by treatment with transglutaminase (TG) in the on-chip angiogenesis assay using the PTM method (Fibrinogen (F) 0.5/TG 0: $n = 30$; F 1.25/TG 0: $n = 29$; F 2.5/TG 0: $n = 28$; F 2.5/TG 0.1: $n = 30$; F 2.5/TG 0.5: $n = 30$, from 4 independent experiment). **d**–**l** Effects of ECM stiffening by TG treatment on on-chip angiogenesis of ECs. **d** A vascular distensibility test. Representative DIC images of angiogenic branches before and after loading additional intraluminal hydrostatic pressure when gel without and with TG treatment was used (left). White dotted lines indicate the distal and proximal sides of the quantified lumen. Comparison of branch diameter changes (right, TG (−): $n = 5$; TG (+): $n = 4$, from 4 independent experiments). **e**–**g** Representative DIC images (**e**) of angiogenic EC branches with and without TG treatment, and quantification by parameters of branch lengths (**f**) and tip expandability (**g**) (TG (−): $n = 70$; TG (+): $n = 47$, from 3 independent experiments). The Tip expandability was quantified as $\tan\theta$, obtained by dividing $l/2$ by $d/2$, where $l$ and $d$ indicate total branch length and branch diameter at the half-way point of the branch, respectively. **h**–**l** Quantification of tip EC movement and lumen development using the gel without and with TG treatment. **h** Kymograph of tip EC movements (left, TG (−): $n = 20$; TG (+): $n = 11$, from 3 independent experiments) and typical move forward and stop pattern of tip EC movement (right), as in the left panel of **h** (blue arrow). **i** Frequency distribution of EC movement speeds. **j**, **k**, Kymographs showing relationships in dynamics between tip EC movement (red) and the nearby lumen area change (distal (D) and proximal (P) areas in black and blue, respectively) using the gel with TG treatment. **j** Representative tip EC movement and lumen area change. Serial DIC images of elongating branches, with trajectory (red line) of the tip EC displacement and lumen diameter (yellow bars) at timepoints indicated by numbers in the kymograph (orange dotted line) (top, See also Supplementary Movie 7). White arrows indicate displacements of the nuclei of tip ECs. **k** Data for 100 min are shown and the half-way timepoint is set as 0 min (orange dotted line) ($n = 10$, from 3 independent experiments). As to lumen area changes, data are shown for predominantly expanded proximal (P) or distal (D) lumens. **l** Box and individual plots of lumen area changes for 30 min prior to the half-way timepoint ($p = 0.8691$ between values with (red) and without (blue) deceleration of tip EC movement). Data are expressed as means ± SD (**d**). All box plots show the interquartile range, with the middle line defining the median, and whiskers show the minimum and maximum values, excluding outliers (**c**, **f**, **g**, **l**). Outlier values are defined as being 1.5 times the interquartile range above and below the third and first quartile, respectively. Scale bars: 20 μm (**d**), 25 μm (**j**) and 100 μm (**e**). Two-sided Mann-Whitney U test with a Holm correction (**c**), Two-sided Mann-Whitney U test (**d**, **f**, **g**, **l**) and two-sided Kolmogorov-Smirnov test (**i**). Source data are provided as a Source Data file.

## Pericyte effects on angiogenic morphogenesis driven by ECs

The next question was whether there is an additional regulatory system further controlling branch elongation efficiently, while lumen formation progresses, in the process of angiogenic morphogenesis seen during normal developmental stages. To address the question, we focused on morphological differences between angiogenic branches with and without pericyte coverage. Pericytes, one of the cell types comprising the vasculature, is present adjacent to ECs even in elongating angiogenic branches (Fig. 4a)[31–33]. Consistent with a previous report[33], in murine retinal angiogenic branches lacking pericyte coverage due to treatment with neutralizing PDGFRβantibodies (APB5)[33] (Fig. 4a), radial growth of angiogenic branches was delayed, markedly on postnatal day 3 (P3) and on P4 (Fig. 4b, c). Also, the branch diameters, which generally correlate with lumen diameters regardless of whether or not pericytes are present and which portion of the branch is examined (Supplementary Fig. 9), were enlarged even around the tip (Fig. 4d, e). In on-chip angiogenesis of ECs obtained by adding pericytes, angiogenic branches covered by pericytes and with a luminal structure were induced (Fig. 4f, g), and the actions of pericytes on vascular morphogenesis observed in murine retina, such as enhanced branch elongation (Fig. 4h, i) and reduced branch diameter including around the tip of the branch (Fig. 4h, j, k), were reproducible. These observations confirmed that pericytes control branch elongation as well as lumen and branch expansions during angiogenic morphogenesis.

We further analyzed tip EC movement changes in the on-chip angiogenesis assay performed with pericyte coculture. Time-lapse imaging showed that, in clear contrast to the dynamics of tip EC movement in EC-only on-chip angiogenesis, in the presence of pericytes, most of the tip ECs tended to continue moving forward, characterized by a higher frequency of forward movement with greater speed (Fig. 5a–c and Supplementary Fig. 10, See also Supplementary Movie 11), which is highly similar to the pattern of tip EC movement when lumen expansion was restricted using TG-treated stiffened ECM (Figs. 2h, i, 5E and Supplementary Fig. 4, See also Supplementary Movie 7). On the other hand, the sizes of nearby lumens remained essentially constant, though occasional decreases were seen, regardless of whether the lumen was distal or proximal to the nuclei, and excessive expansion was rarely noted (Fig. 5c–e), in contrast to the condition without pericytes (Fig. 1h, i). Notably, one of 20 tip ECs showed apparent deceleration of forward movement after nearby lumen expansion, despite having been cocultured with pericytes,

supporting the presence of an inverse correlation between lumen expansion and tip EC forward movement driving branch elongation.

Also, to clarify the relationships between the forward directional movement of tip ECs and lumen and branch expansions in vivo, we examined the position of the Golgi apparatus relative to the nucleus, an anterior side marker in moving cells with anterior-posterior cell polarity[14,15] Consistent with the results obtained employing the on-chip angiogenesis assay (Fig. 5f, g), whole-mount immunostaining images of murine P4 retina showed the ratio of tip ECs with distal localization of the Golgi apparatus to be reduced in angiogenic branches lacking pericyte coverage and displaying expanded branches (Fig. 5h, i), showing pericytes to promote forward directional movement of tip ECs with anterior-posterior cell polarity, thereby leading to enhancement of branch elongation in vivo. These observations suggest pericytes to efficiently drive angiogenic morphogenesis by integrating branch elongation and lumen development. Taken together with the aforementioned results of TG treatment, these results also suggest that for the integrating mechanism, pericytes maintain the forward directional movement of tip ECs possibly by restricting lumen and branch expansions, although we cannot completely rule out the possibility that pericytes control branch elongation and lumen expansion independently.

## Perivascular stiffening by pericytes via deposition of type-IV collagen on vascular basement membrane

A previous study demonstrated micro-vessel distensibility in response to increased intraluminal pressure ($\Delta P$) to, at least theoretically, depend on physical properties of the surrounding ECM in vivo (Fig. 2a)[22]. Indeed, in a vascular distensibility test, the branches around the tip ECs rarely expanded in response to additional pressure loading when pericytes were present (−0.12 ± 2.6 % increase in branch diameter), whereas the same pressure loading caused lumen and branch expansions in the absence of pericytes (7.7 ± 2.7 % increase in branch diameter) (Fig. 6a), suggesting increased perivascular stiffening with coexistent pericytes. Thus, we determined whether pericytes stiffen the extravascular environment surrounding the tips of angiogenic sprouts, where de novo lumens emerge and expand, using the PTM method. Employing the PTM test, we found the stiffness of the areas around (<5 μm from branch, Near) angiogenic branches at the tip to be increased in the presence of pericytes while that in the areas far from angiogenic branches was not (outside of the branch (>10 μm), Far)

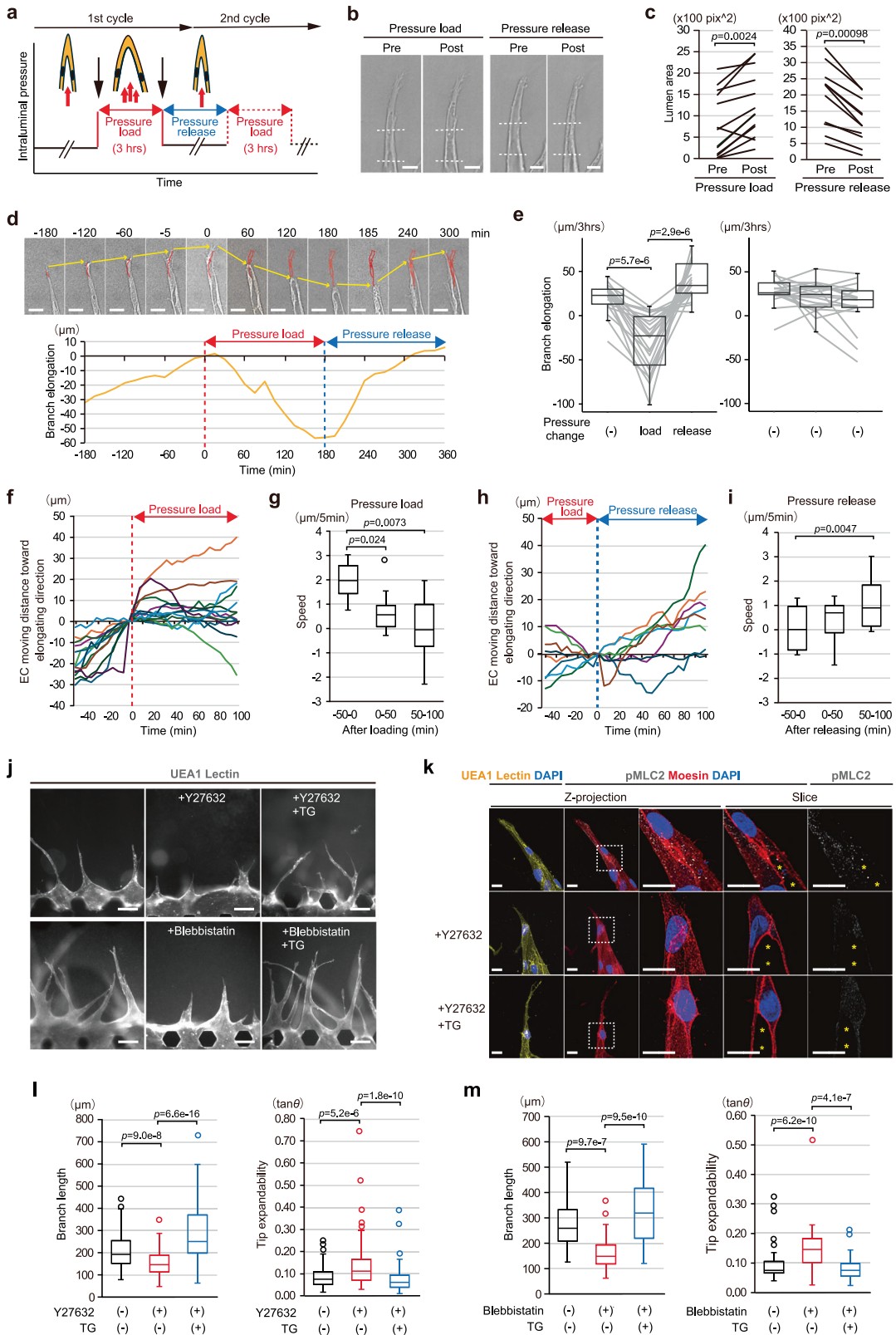

(Fig. 6b). In contrast, the stiffness was not significantly changed regardless of the distance from branches in the absence of pericytes (Fig. 6b). Collectively, these results indicate that coculturing ECs with pericytes increased the stiffness of the ECM surrounding the angiogenic branches, thereby restricting lumen and branch expansions, possibly resulting in effective forward directional movement of tip ECs and, consequently, elongation of angiogenic branches.

We subsequently tackled the issue of how the perivascular structure around the tips of angiogenic branches became stiffened in the presence of pericytes. Mural cells including pericytes were previously reported to stimulate vascular basement membrane (VBM) formation via interactions with ECs in vascular morphogenesis in vitro[34] as well as in zebrafish dorsal aorta development in vivo[35]. The VBM, which is connected directly to the surrounding ECM, provides

**Fig. 3 | Influences of interventions for lumen and branch expansion altering tip EC movement and branch elongation. a–i** Physical interventions altering lumen and branch expansions by externally loading intraluminal pressure in on-chip angiogenesis of ECs. **a** A schematic diagram showing the timeline of the loading and release cycles of intraluminal pressure. **b, c** Changes of lumens and branches when loaded followed by release of the additional intraluminal pressure.
**b** Representative DIC images. White dotted lines indicate the proximal and distal boundaries of the analyzed luminal region. **c** Quantification of the changes in luminal area (loading: $n = 12$; release: $n = 11$, from 3 independent experiments).
**d–i** Time-lapse analyses of on-chip angiogenesis during the intervention.
**d** Representative serial DIC images at the elapsed time indicated at the top before and after loading and the release of additional intraluminal pressure (top, See also Supplementary Movie 8) and kymograph of branch elongation (bottom). In the DIC images, yellow arrows and red lines indicate the trajectory of the tip of the branch and the tip EC displacement, respectively. In the kymograph, red and blue dotted lines indicate time points at which additional intraluminal pressure was loaded and then released, respectively. **e** Box plots of averaged branch elongation and the individual plots 3 h before (pressure change (−)) and after (load) the 1st pressure loading and 3 h after its release (release) (left), and the corresponding box and individual plots in the control group without interventions (right) (pressure loading and release: $n = 21$ from 4 independent experiments, control: $n = 19$, from 3 independent experiments, See also Supplementary Fig. 6). **f–i** Kymograph showing the dynamics of tip EC movement for 50 min before and 100 min after loading (**f**) and

releasing (**h**) additional intraluminal pressure (time=0, dotted red line), and the quantification showing in box plots of averaged tip EC movement speed (**g**, each group: $n = 13$ from 4 independent experiments; **i**, each group: $n = 8$ from 4 independent experiments). **j–m** Pharmacological interventions altering lumen and branch expansions of on-chip angiogenesis of ECs by treatment with Y27632 (top) and Blebbistatin (bottom), and without and with TG treatment of the gel.
**j** Representative fluorescent images of the angiogenic branches. **k** Representative confocal z-projection images, with magnified images of square areas bound by dots (z-projection) and their corresponding confocal x-y slice images at specific z-portions (slice), showing localization pattern of pMLC2 in the angiogenic branches around the area where the de novo vascular lumen (yellow asterisks) develops. **l, m**, Box plots of branch length (left) and Tip expandability (right), without and with Y27632 (**l**) or Blebbistatin (**m**) and without and with TG treatment of the gel (**l**, Y27632 (−) TG (−): $n = 117$; Y27632 (+) TG (−): $n = 108$; Y27632 (+) TG (+): $n = 102$, from 3 independent experiments, (**m**), Blebbistatin (−) TG (−): $n = 39$; Blebbistatin (+) TG (−): n = 35; Blebbistatin (+) TG (+): n = 64, from 3 independent experiments). All box plots show the interquartile range, with the middle line defining the median, and whiskers show the minimum and maximum values, excluding outliers (**g, i, l, m**). Outlier values are defined as being 1.5 times the interquartile range above and below the third and first quartile, respectively. Scale bars: 20 μm (**d, k**), 25 μm (**b**) and 100 μm (**j**). Two-sided Wilcoxon signed rank test (**c**), Wilcoxon signed rank test with a Bonferroni correction (**e**) and two-sided Mann-Whitney U test with a Bonferroni correction (**g, i, l, m**). Source data are provided as a Source Data file.

the structural and mechanical support required by micro-vessels[21,36]. The mechanical properties depend upon ECM components and their abundances[21,36]. Of the core components that include laminin and type-IV collagen (Col-IV), the latter is regarded as being a primary determinant of the mechanical properties of the VBM because of its extensive cross-linkages[21,36]. Studies producing both in vitro and in vivo data have raised the possibility that Col-IV of VBM may make a more important contribution to determining branch diameter than other VBM components[34,35]. We thus hypothesized that pericytes may affect VBM formation by inducing spatiotemporally-enhanced Col-IV deposition, which in turn leads to appropriate stiffening of the perivascular structure of angiogenic branches.

To test this hypothesis, we compared the degree of Col-IV deposition on VBM in proximity to the leading edge of an angiogenic branch, the site of de novo vascular lumen development, between on-chip angiogenesis assay conditions with and without pericyte coculture. A detergent-free whole mount immunostaining procedure, that enables examination of only the extracellular component, clearly demonstrated Col-IV deposition on VBM, positioned just outside of the EC membrane at the basal side, in proximity to the branch where the de novo lumen was developing (Fig. 6c, d) in the presence of pericytes. In contrast, the Col-IV deposition on VBM around the position of de novo lumen development was reduced in the absence of pericytes (Fig. 6c, d), as confirmed by quantification analyses of both the Col-IV coverage ratio and the fluorescence intensity per volume (Fig. 6e). We also observed similar pericyte effects on VBM deposition of Col-IV in murine retinal angiogenesis in vivo. The Col-IV coverage ratio on VBM around the tip of the branch was decreased in the P4 retinal angiogenic branches lacking pericyte coverage due to APB5 treatment (Fig. 6f, g and Supplementary Fig. 11). These data, viewed collectively, support our hypothesis that pericytes enhance Col-IV deposition on VBM around the angiogenic branches where the de novo vascular lumen develops. It should be noted that deposition of laminin, one of the other main VBM components, was also enhanced at the leading front of the angiogenic branches with pericyte coverage, as described in a prior report[34], though the degree of enhancement was smaller than that seen with Col-IV deposition (Supplementary Fig. 12).

## Functional involvement of Col-IV deposition on VBM in angiogenic morphogenesis

Subsequently, we examined the functional involvement of Col-IV deposition on VBM in the enhancement of branch elongation via

stiffening of the perivascular structure employing interference in the gene expression of Col-IV. Consistent with the findings of a previous study[34], both ECs and pericytes, used for the on-chip angiogenesis assay, expressed *COL4A1* and *COL4A2*, Col-IV subunits of VBM (Supplementary Fig. 13a), though ECs are reportedly more likely to be a major contributor to Col-IV deposition on VBM[34]. We knocked down both *COL4A1* and *COL4A2* in either ECs (*COL4A1/A2* DKD ECs) or pericytes using siRNA (Supplementary Fig. 13b), then carried out the on-chip angiogenesis assay with pericytes. As we expected, the double knock-down in ECs reduced Col-IV deposition on VBM around angiogenic branches, especially around the portion where the de novo vascular lumen was developing (Fig. 7a–c), and further delayed branch elongation with an enlarged branch diameter even when cocultured with pericytes (Fig. 7a, d, e), as seen in the absence of pericytes (Fig. 4h–k). Notably, branch regression, characterized by VBM structure lacking an internal vessel structure composed of ECs, was frequently seen after abnormal lumen expansion in an on-chip angiogenic branch composed of *COL4A1/A2* DKD ECs (Fig. 7f). On the other hand, the double knock-down in pericytes did not significantly reduce Col-IV deposition on VBM around branches, even around the tips, although branch elongation was slightly reduced and both branches and lumens also displayed slight expansion (Fig. 7a-e). These data revealed the functional involvement of Col-IV deposition on VBM in angiogenic morphogenesis, and also suggest that mainly EC-derived Col-IV was involved in the formation and function of VBM at least around the tip portion where the de novo vascular lumen develops. We further assessed whether or not Col-IV deposition on VBM contributed to angiogenic morphogenesis via stiffening of the perivascular structure. The PTM analysis demonstrated the perivascular structure around the leading fronts of on-chip angiogenic branches composed of *COL4A1/A2* DKD ECs to exhibit reduced stiffness (Fig. 7g). Additionally, retardation of branch elongation as well as branch and lumen expansions observed in on-chip angiogenesis with *COL4A1/A2* DKD ECs and untreated pericytes were rescued, though not completely, by using TG-treated stiffened ECM (confirmed by the PTM analysis), with no changes in Col-IV deposition on the VBM (Fig. 7h–m). These results collectively suggest that spatiotemporally-enhanced deposition of EC-derived Col-IV on VBM in the presence of pericytes controls perivascular stiffness appropriately, which promotes branch elongation by physically restricting both lumen and branch expansions.

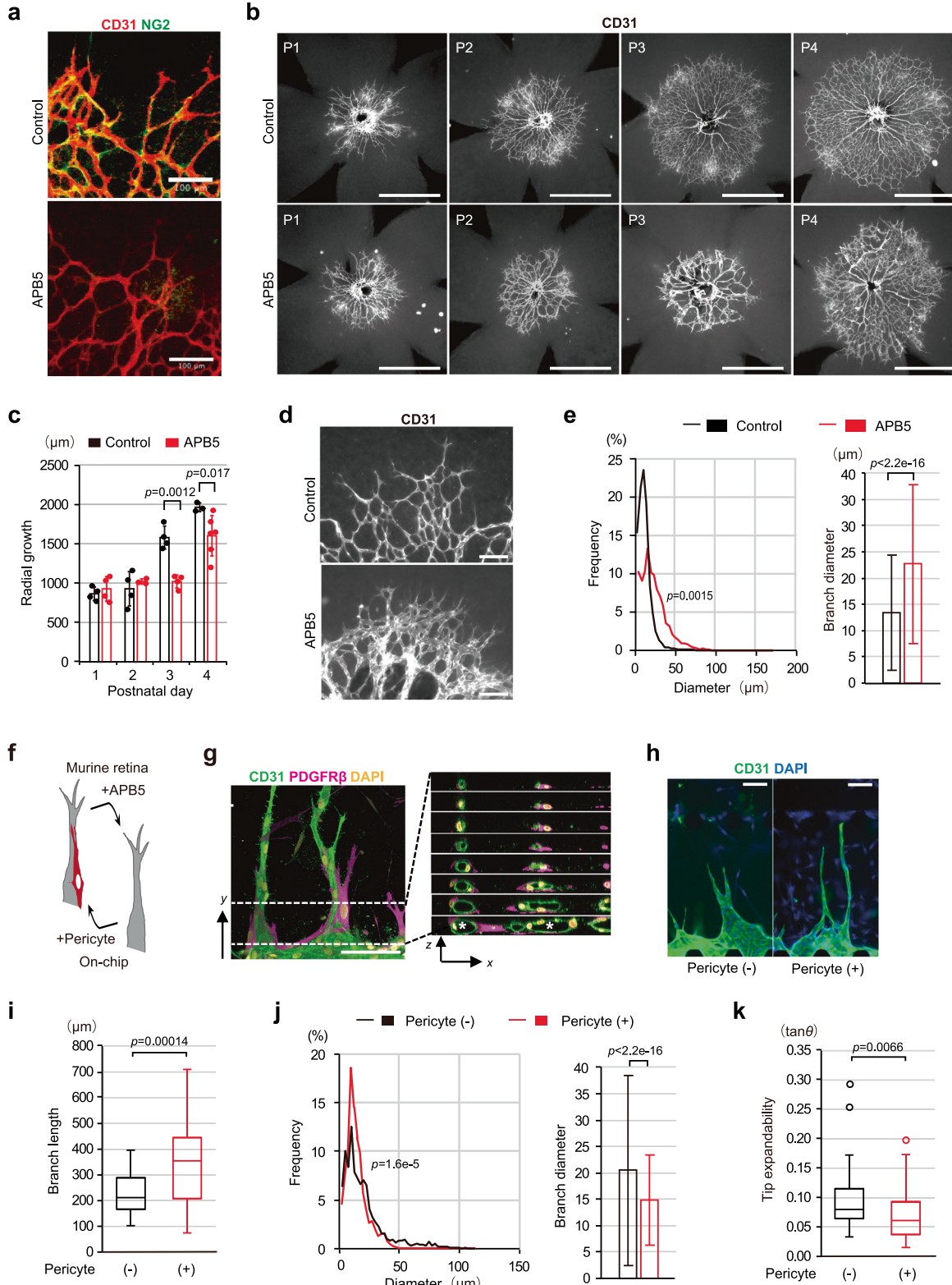

## Mechanism controlling tip EC movement during lumen and branch expansions

Finally, we addressed the mechanisms by which the directional movement of tip ECs is controlled in response to lumen and branch expansions. We previously showed that the F-BAR proteins, CIP4 and TOCA1, localized to the leading edge of tip ECs, to promote Arp2/3 complex-mediated actin polymerization, which is essential for the directional movement leading to angiogenic branch elongation[12]. We further identified CIP4 and TOCA1 to function as a mechano-sensor of EC membrane stretching, in which failure of membrane localization of CIP4 and TOCA1 induced by ectopic intraluminal pressure loading-dependent vessel expansion resulted in impairment of Arp2/3 complex-driven directional movement of tip ECs[12]. Therefore, we hypothesized that a similar mechanism(s) may operate during

**Fig. 4 | Effects of pericytes on angiogenic morphogenesis by ECs.**
**a**–**e** Morphological differences between angiogenic branches with and without APB5 treatment which removes pericyte coverage in the murine retina.
**a** Representative confocal z-projection images of the angiogenic fronts on postnatal day 4 (P4). **b**, **c**, Representative fluorescent images (**b**) showing whole angiogenic branches on P1, 2, 3 and 4 and quantification for radial growth of angiogenic branches (**c**, for P1, control (CTL): $n = 4$; APB5: $n = 4$, for P2, CTL: $n = 4$; APB5: $n = 3$, for P3, CTL: $n = 4$; APB5: $n = 4$, for P4, CTL: $n = 3$; APB5: $n = 6$, from 3 independent experiments). **d**, **e**, Representative magnified views of the angiogenic fronts on P3 (**d**) and quantification of the branch diameters of retinal angiogenic branches: frequency distributions (left of **e**) and average values (right of **e**) of branch diameter ($n = 3$ retinas each group from 3 independent experiments).
**f** Schematic diagram showing reversal of the murine retinal assay and the on-chip assay to examine the effects of pericytes on angiogenic morphogenesis; in the murine retinal assay, the effect was analyzed by removing pericytes with APB5 treatment, whereas in the on-chip assay pericytes were added. **g**–**k** Morphological differences, in the on-chip assay, between angiogenic branches without and with coculturing with pericytes. **g** A representative confocal 3D image (left) and its orthogonal views (right) showing vascular lumen structures (white asterisks) in the angiogenic branches of ECs with adjacent pericytes. **h**–**k** Representative fluorescent images of angiogenic EC branches (**h**) and quantification of branch lengths (**i**), branch diameters (**j**) (frequency distributions (left) and average values (right)) and Tip expandability (**k**) (for **i** and **k**; $n = 33$ branches without pericytes and 50 branches with pericytes, from 3 independent experiments, for **j**; $n = 46$ branches without pericytes and 39 branches with pericytes, from 3 independent experiments). Data are expressed as means ± SD (**c**, right panels of **e** and **j**). All box plots show the interquartile range, with the middle line defining the median, and whiskers show the minimum and maximum values, excluding outliers (**i**, **k**). Outlier values are defined as being 1.5 times the interquartile range above and below the third and first quartile, respectively. Scale bars: 100 μm (**a**, **d**, **g**, **h**), 500 μm (**b**). Two-sided Welch's *t* test (**c**), Two-sided Kolmogorov-Smirnov test (left panels of **e** and **j**) and two-sided Mann-Whitney U test (right panels of **e** and **j**, **i**, **k**). Source data are provided as a Source Data file.

physiological lumen and branch expansions in angiogenesis to suppress the directional movement of tip ECs.

To test this hypothesis, we evaluated the localization patterns of Arp2/3 complexes and the F-BAR proteins in tip ECs of on-chip angiogenesis of ECs comparing these parameters with the lumen developmental state, and further examined changes in both, when lumen and branch expansions were restricted intentionally by stiffening the surrounding ECM with TG treatment, when additional intraluminal pressure was loaded, and both. Whole-mount immunostaining demonstrated variations in the localization patterns of Arp2/3 complexes (detected by ARPC2) dependent on angiogenic branches, with some tip ECs showing their localization predominantly at the leading front, while others did not (Fig. 8a). On the other hand, when branch and lumen expansions were restricted by stiffening the ECM with TG treatment, more tip ECs showed predominant Arp2/3 complex localization at the leading front (Fig. 8a, b). We further found an inverse correlation between predominant Arp2/3 complex localization at the leading front of tip ECs and the lumen developmental state around the tip ECs ($r = -0.4486$); in the tip ECs of the branch that largely occupied the lumen, a smaller number of Arp2/3 complexes was localized at the leading front regardless of whether or not the ECM gel had been treated with TG (Fig. 8c, d and Supplementary Fig. 14). Similar localization patterns in CIP4 as well as TOCA1 tagged with EGFP in the N-terminus (EGFP-TOCA1) were observed (Fig. 8e–g, and Supplementary Fig. 15); CIP4 and EGFP-TOCA1, which were colocalized with Arp2/3 complexes (Supplementary Fig. 15b, c and 16), were more dominantly localized at the leading front of tip ECs when the ECM contained TG (Fig. 8f and Supplementary Fig. 15d), and, in more lumen-dominated branches, smaller numbers of CIP4 and EGFP-TOCA1 were localized at the leading front of tip ECs ($r = -0.5547$, Fig. 8g and Supplementary Fig. 15e). Furthermore, when additional intraluminal pressure was loaded, the majority of the Arp2/3 complex and F-BAR protein disappeared from the leading front of tip ECs in conjunction with lumen and branch expansion, resulting in retarded branch elongation, which was inhibited by stiffening the ECM with the TG treatment (Fig. 8h–j and Supplementary Fig. 17). These data indicate suppression of localizations of Arp2/3 complexes and the F-BAR proteins at the leading front of tip ECs to be closely related to lumen emergence and its expansion, and that these localizations might be biomechanically controlled.

Moreover, we similarly investigated CIP4 and Arp2/3 complex localization at the leading front of tip ECs in on-chip angiogenesis during coculture with pericytes and examined the relationships between their localizations and lumen development. Consistent with results when we limited branch and lumen expansions by treating the ECM with TG, greater numbers of CIP4 colocalized with Arp2/3 complexes were localized at the leading front of tip ECs in angiogenic branches with pericytes, as compared to those without pericytes (Fig. 9a–c), in conjunction with less lumen occupation adjacent to the tip ECs in the branch ($r = -0.4638$ for CIP4, $r = -0.4417$ for Arp2/3 complex, Fig. 9d). Taken together with our previous data[12], these results strongly support our hypothesis and suggest that forward directional movement of the tip ECs might be suppressed via impairment of localization of the F-BAR proteins at the leading front in response to lumen and branch expansions, followed by failure of Arp2/3 complex-mediated actin polymerization, which is counter-regulated biomechanically by the presence of pericytes.

## Discussion

During angiogenic morphogenesis, ECs and pericytes coordinate to fashion and elongate their cord-like structures via directional movement of the ECs[6,7,10,17]. Almost simultaneously, new luminal structures develop while exposed to mechanical stimuli due to inflow of blood[6–9,17–19]. Here, we identified perivascular stiffening as a major factor that integrates branch elongation and lumen development to allow angiogenesis to proceed properly in the presence of blood flow-induced intraluminal pressure. We showed that lumen emergence and its ensuing expansion inhibits polarization and directional movement of tip ECs, resulting in delayed branch elongation. This process was counter-regulated by pericyte-mediated perivascular stiffening that prevented excessive lumen expansion. We demonstrated that pericytes mediate proper Col-IV deposition onto the VBM by ECs, which accounts for perivascular stiffening and reduction in vascular distensibility. Finally, we found that the leading front localization of F-BAR proteins and Arp2/3 complexes in tip ECs decreases with emergence and expansion of the nearby lumen, suggesting a possible molecular mechanism for the biomechanical inhibition of the directional migration of tip EC's, and that this process is counter-regulated by the presence of pericytes. Together with recent finding[12,24], the present results allow us to propose a model for the biomechanical control of angiogenic morphogenesis, wherein ECs efficiently elongate angiogenic branches via their directional movement, while the lumen develops, by properly controlling the balance among surrounding mechanical factors (Supplementary Fig. 18a). Our results also highlight a coordinated role played by pericytes in promoting angiogenesis via control of this mechanical balance.

Intraluminal pressure is essential to promoting physiological lumen development[24]. On the other hand, our recent work demonstrated that, in wound angiogenesis, ectopic loading of intraluminal pressure inhibited branch elongation by inducing abnormal lumen expansion[12]. In the present study, we showed the presence of a similar inhibitory mechanism in developmental angiogenesis, and revealed a mechanism that regulates the inhibitory actions of intraluminal pressure by pericyte-mediated control of perivascular stiffening. These

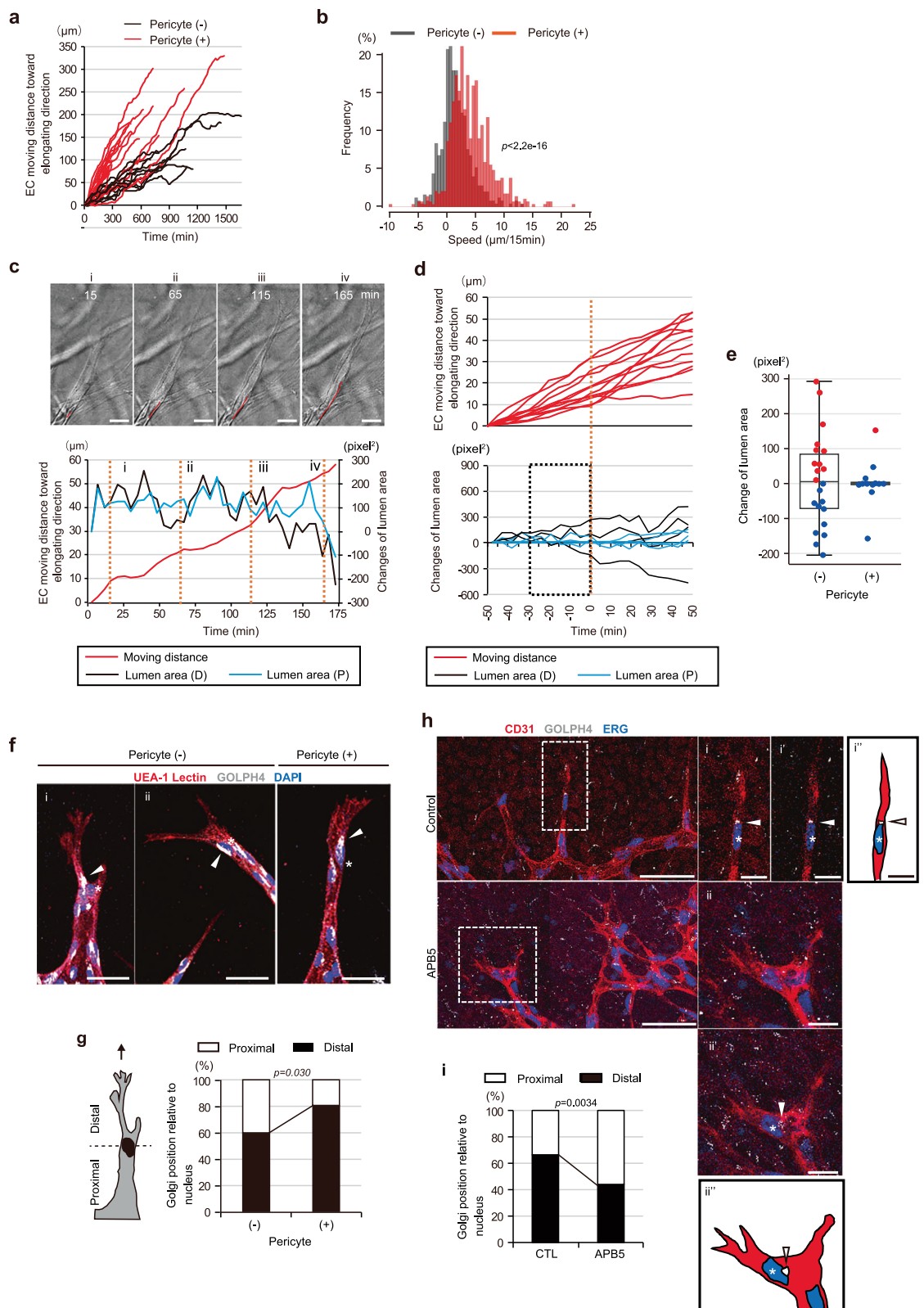

results raise the possibility of the existence of a fundamental machinery constructing dynamic tissue morphology, beyond angiogenesis, which operates via biomechanical integration of multi-morphogenetic steps.

Mural cells including pericytes contribute to vascular maturation and maintenance during development, processes that are closely associated with mural cell-mediated VBM formation and function, including Col-IV deposition[35,37–40]. However, the roles pericytes play in the early stage of vascular development have not been fully elucidated. The present study builds on our previous findings[33] that demonstrated that pericytes promote branch elongation, keeping the branch thin, during sprouting angiogenesis. We identified a causal relationship between these morphogenetic steps, supporting the idea of a pericyte-dependent promotion of branch elongation via branch thinning. We

**Fig. 5 | Pericytes restrict lumen expansion and enhance forward directional movement of tip ECs. a–e** Quantification of tip EC movement and lumen development in the presence and the absence of pericytes in an elongating branch as demonstrated by on-chip angiogenesis. **a** Kymograph of tip EC movements ($n = 20$ with pericytes; $n = 11$ without pericytes, from 3 independent experiments). **b** Frequency distribution of EC movement speeds with and without pericytes. **c**, **d** Kymographs showing relationships in dynamics between tip EC movement (red) and the nearby lumen area change (distal (D) and proximal (P) areas in black and blue, respectively) in the presence of pericytes. **c** Representative tip EC movement and lumen area change. Serial DIC images of elongating branches, with trajectory (red line) of the tip EC displacement at timepoints indicated by numbers in the kymograph (orange dotted line) (top, See also Supplementary Movie 11). **d** Data for 100 min are shown and the half-way timepoint is set as 0 min (orange dotted line). As to lumen area changes, data are shown for predominantly expanded proximal (P, blue) or distal (D, black) lumens ($n = 12$ from 3 independent experiments). **e** Box and individual plots of lumen area changes for 30 min prior to the half-way timepoint in the absence and the presence of pericytes ($p = 0.8591$ between values with (red) and without (blue) deceleration of tip EC movement). All

box plots show the interquartile range, with the middle line defining the median, and whiskers show the minimum and maximum values, excluding outliers. Outlier values are defined as being 1.5 times the interquartile range above and below the third and first quartile, respectively. **f–i** Localization patterns of the Golgi apparatus in the tip ECs. **f**, **h** Representative confocal z-projection images showing localization of the Golgi apparatus (white arrowheads) at the distal (i) or the proximal (ii) side of the nucleus toward the direction of branch elongation in the tip ECs (asterisks) of on-chip (**f**) or murine retinal (**h**, P4) angiogenic branches with and without pericytes. In **h** confocal $x$-$y$ slice images at specific z-positions in areas within squares bounded by dots are shown in i and i′ or ii and ii′ and drawings of i′ and ii′ are shown in i″ and ii″, respectively. **g**, **i** Quantification of the ratio of tip ECs having the Golgi apparatus positioned at the distal (arrowheads in the left (i) and the right panels of **f** as examples) or the proximal (arrowhead in the middle panel (ii) of **f** as an example) side position relative to the center of the nucleus (for **g**, pericyte (−): $n = 35$; pericyte (+): $n = 57$, from 3 independent experiments, for **i**, control (CTL): $n = 80$; APB5: $n = 85$, from 3 independent experiments). Scale bars: 10 μm (i, i′, ii and ii′ of **h**), 25 μm (**c**), 50 μm (**f**, left panel of **h**). Two-sided Kolmogorov-Smirnov test (**b**) and two-sided Chi-squared test (**g**, **i**). Source data are provided as a Source Data file.

further revealed that VBM stiffening via enhanced deposition of Col-IV, mainly secreted by ECs, causes branch thinning in the presence of pericytes. Similarly, normal Col-IV deposition on VBM via mural cell-EC interactions controls vessel diameter in zebrafish dorsal aorta development[35]. Pericytes are not always just adjacent to ECs in the angiogenic branches, as observed in cranial vessels and intersegmental vessels during development in zebrafish embryo[41,42]. However, in vivo imaging in zebrafish embryo[42] and single cell RNAseq analyses in mice[43–46] suggest that perivascular cells, such as fibroblast-like cells adjacent to ECs that later giving rise to pericytes, can act as a substitute for pericytes to form VBM in coordination with ECs. This may allow other perivascular cells, similar to pericytes, to maintain perivascular stiffening and integrally control vessel elongation and lumen development via biomechanical processes.

Col-IV deposition on VBM is regulated by a range of steps, including expressions of the constituent genes, *COL4A1* and *COL4A2*, their assemblies into the VBM and their eventual degradation[21]. Pericytes contribute to enhanced Col-IV deposition on VBM by increasing *COL4A1* and *COL4A2* expression in both ECs and pericytes[21], and by suppressing the degradation by TIMP (tissue inhibitor of metalloproteinase)−3 secreted by pericytes themselves[47]. The present data advance our previous understanding and suggest the importance of EC-derived Col-IV deposition via EC-pericyte interactions in achieving adequate VBM function in angiogenesis, although the molecular mechanism underlying these interactions remains unknown.

Our present work highlights the importance of the physical properties surrounding nascent vessels during angiogenesis, especially the mechanical balance among factors that include intraluminal pressure, vascular wall stretching (tension) and perivascular stiffness. Modes of angiogenesis may differ, reflecting variations in these physical parameters under different physiological and pathological conditions, allowing the process of vessel formation to be explained in a unified manner (Supplementary Fig. 18b). In physiological angiogenesis, the mechanical balance is properly controlled to maintain lumen and branch expansion at adequate levels, but this balance appears to be disrupted in pathological states. For example, in wound angiogenesis, an unbalanced mechanical state, i.e., increased vascular wall stretching due to pressure overloading, inhibits branch elongation with the lumen of the upstream injured branch becoming dilated[12]. In contrast, in tumor angiogenesis, an unbalanced mechanical state of reduced vascular wall stretching due to increased stiffness of the tumor microenvironment and/or the interstitial pressure, enhances elongation via thinning of branches, which is associated with abnormal tumor neovascular development and poor outcomes[48,49]. Restoring mechanical balance may therefore constitute a possible therapeutic target for controlling pathological angiogenesis. High interstitial

pressure in tumors is caused by enhanced permeability of the neo-vasculature[50], suggesting that suppression of vascular permeability is a possible strategy for an anti-angiogenesis therapy via normalization of the mechanical balance. In addition to the surrounding physical properties, cell-autonomous roles of the constituent cells are also important for determining the mechanical balance controlling lumen expansion during, not only for the formation of epithelial, but also endothelial, tubular structures[24,28–30]. In zebrafish, active actomyosin contractility in the apical cortex increases cortical stiffness and causes retraction of membrane blebbing, thereby limiting lumen and branch expansions during angiogenic morphogenesis[24,30]. Indeed, in the present study, suppression of apical actomyosin contractility by ROCK or NMII inhibition induced both branch and lumen expansions around the tips of ECs undergoing on-chip angiogenesis and reduced branch elongation. How cell-autonomous mechanisms are involved in mechanical unbalance and pathological angiogenesis remains to be investigated.

The physical properties of the surrounds are quite important for sculpting the complex shapes and structures of tissues and organs[2,3,51,52]. The basement membrane (BM), which surrounds most tissues and organs, serves as a major determinant of the surrounding physical properties[36,53]. Dynamic spatiotemporal construction and remodeling of the BM is required to achieve controlled outgrowth of tissues, such as anisotropic growth, via generating mechanical heterogeneity during tissue sculpting[28,54]. The present data showed that, in angiogenic morphogenesis, the BM mechanically guides anisotropic growth-like elongation of tubular structure along the longitudinal axis, which emphasizes our previous understanding of the involvement of the surrounding physical properties in tissue and organ morphogenesis. Our data advances this concept by showing the importance of the mechanical balance among the surrounding physical factors and their involvement in the integration of distinct morphogenetic steps.

The present work revealed a mechanism possibly integrating branch elongation and lumen development at the cellular level, which in turn governs how circumferential lumen and branch expansions inhibit the forward directional movement of ECs during lumen development. Our previous work showed that ECs sense abnormal membrane tension, generated by lumen expansion followed by vascular wall stretching, via the F-BAR family of proteins, TOCA-1 and CIP4, and thereby lose the antero-posterior cell polarity essential for their directional movement[12]. The present data suggest that a similar mechanism mediated by F-BAR proteins might function in response to physiological lumen and branch expansions during lumen development, which is further controlled biomechanically by pericytes, although this would need to be confirmed by simultaneously

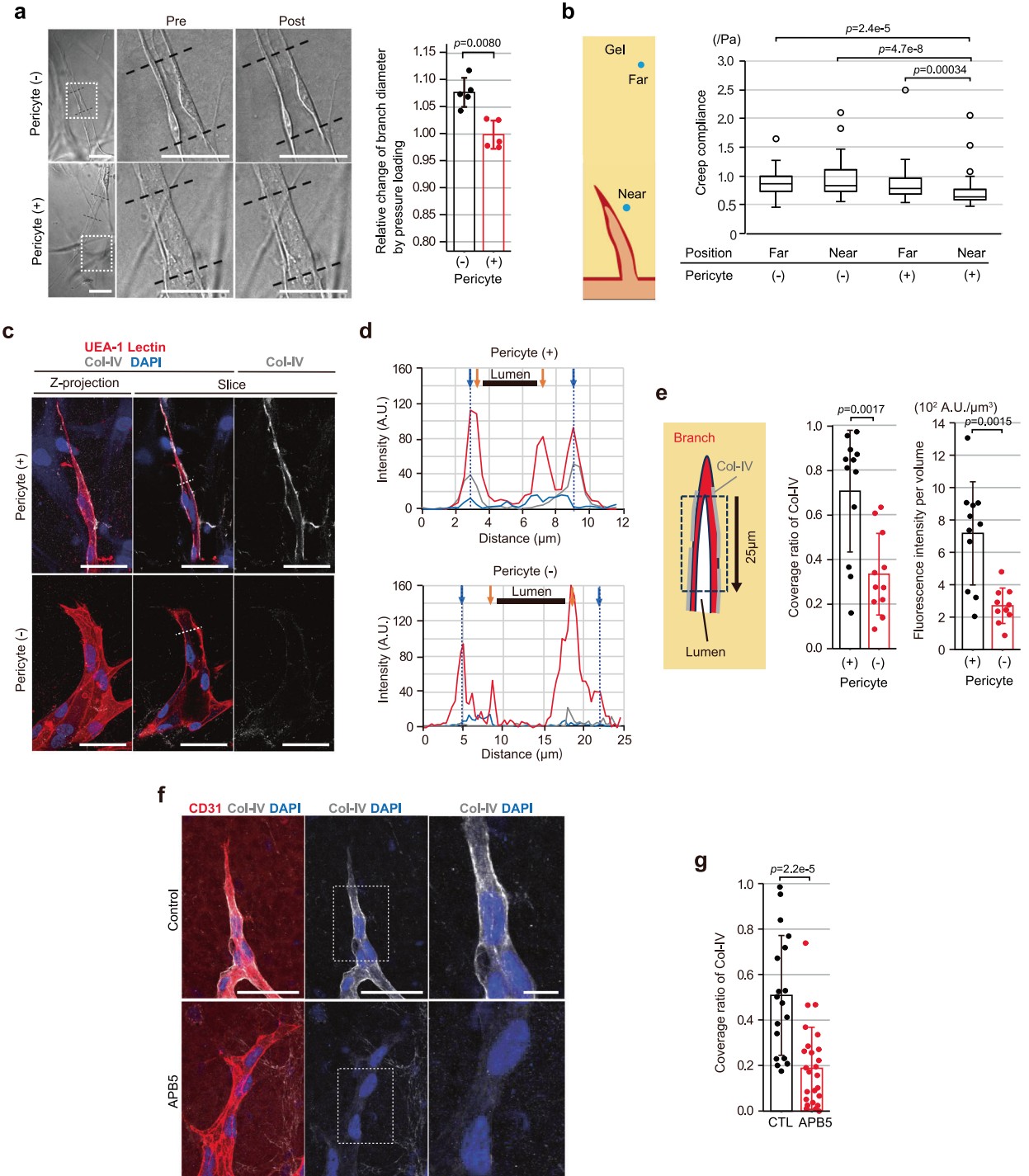

monitoring the dynamics of the F-BAR protein localization and lumen development. Moreover, the involvement of membrane tension in such a mechanism remains unclear. On the other hand, lumen expansion can also raise the intercellular tension that reinforces EC-EC adhesion via the cadherin machinery and promotes contact extension, which then leads to the loss of antero-posterior cell polarity[55–57]. Future studies should assess the involvement of different tensions generated by lumen expansion with the impairment of directional EC movement. Furthermore, we cannot exclude the possibility that ECs directly sense the perivascular stiffening, leading to efficiently elongating branch, while narrowing branch diameter.

The two individual morphogenetic processes examined herein, branch elongation and lumen development, are based on distinct

cellular and multi-cellular behaviors. Branch elongation is driven by directional and collective movements of ECs polarizing along the antero-posterior axis[10–13]. For lumen development, on the other hand, ECs must acquire apico-basal cell polarity, creating a central space followed by lumen expansion through cell shape changes, including cell flattening along the vascular wall[8,9]. Therefore, to achieve elongation of the branch while the lumen develops, appropriate numbers of ECs need to change their cell polarity to apico-basal, assuring commitment to lumen formation[8,9], while the remaining ECs must keep their antero-posterior cell polarity to drive branch elongation[10–13]. It remains unclear whether or not the degree or/and the spatial control of the different tensions can explain the cell polarity transition in the portion of the cell

**Fig. 6 | Perivascular stiffening by pericytes via Col-IV deposition on VBM.**
**a** Representative DIC images of on-chip angiogenic branches before and after
loading of additional intraluminal hydrostatic pressure (60 Pa) in the absence and
the presence of pericytes in a vascular distensibility test (left). Areas in squares
bounded by dots are magnified in the mid and right. Black dotted lines indicate
distal and proximal sides of analyzed lumen areas for quantification of branch
diameter changes. Comparison of branch diameter changes between the absence
and the presence of pericytes (right, $n = 5$ branches from 5 independent experiments). **b** Direct measurement of fibrin-collagen gel stiffness in the on-chip
angiogenesis assay using the PTM method. Stiffnesses of the gels far from (>10 μm)
or near (<5 μm) angiogenic branches (as shown in a drawing in the left) with or
without pericytes, evaluated with Creep compliance (Pericyte (P) (−)/Far: $n = 55$; P
(−)/Near: $n = 50$; P (+)/Far: $n = 55$; P (+)/Near: $n = 50$, from 3 independent experiments). All box plots show the interquartile range, with the middle line defining the
median, and whiskers show the minimum and maximum values, excluding outliers.
Outlier values are defined as being 1.5 times the interquartile range above and
below the third and first quartile, respectively. **c–g** Col-IV deposition on VBM
around the tip where the de novo vascular lumen develops. **c** Representative
confocal z-projection images and their corresponding confocal x-y slice images at a
specific z-position (slice) in the on-chip angiogenesis. **d** Line scan profiles of
fluorescence intensity of UEA-1 Lectin, Col-IV and DAPI along the dotted line indicated in the confocal x-y slice image of **c**. Orange and blue arrows indicate apical
and basal sides of the vascular wall detected by UEA-Lectin staining (red), respectively. **e** Quantitative analysis of Col-IV deposition on VBM in the perivascular area
between the tip of the lumen and its 25 μm proximal side, shown in the square with
the black dotted boundary in the drawing (left), using the following parameters:
Coverage ratio (middle) (pericyte (−): $n = 11$; pericyte (+): $n = 12$, from 3 independent
experiments) and Fluorescence intensity per volume (right) (pericyte (−): $n = 10$;
pericyte (+): $n = 11$, from 3 independent experiments). **f**, Representative confocal
z-projection images, with and without APB5 treatment that results in removal of
pericyte coverage (left and middle), and the magnified views of the areas in squares
bounded by dots in the middle (right) in murine retinal angiogenesis. **g** Quantitative
analysis of Col-IV deposition on VBM in the perivascular area around the site of de
novo lumen formation with (CTL) and without (APB5) pericyte coverage (CTL:
$n = 18$; APB5: n = 26, from 3 independent experiments). Data are expressed as
means ± SD (**a**, **e**, **g**). Scale bars: 10 μm (**a**) and 50 μm (**c**, **f**). Two-sided Mann-Whitney
U test (**a**, **e**, **g**). Two-sided Mann-Whitney U test with a Holm correction (**b**). Source
data are provided as a Source Data file.

---

population involved in branch elongation, or whether there are any
other major factors involved, such as intraluminal pressure[24].

Our study offers direct evidence that biomechanical regulation of
angiogenic morphogenesis is mediated by the perivascular environment. These insights enhance, not only our understanding of the
principles biomechanically integrating different morphogenetic processes in angiogenesis and other forms of tissue development, but also
open avenues to novel applications in clinical management and
regenerative medicine based on controlling angiogenesis.

## Methods

All genetic modification experiments in this study were approved by
the Gene Recombination Experiment Committee established based on
the University of Miyazaki internal regulations. All animal experiments
using mice were carried out in accordance with the animal care
guidelines of University of Miyazaki, Kumamoto University and
Nagoya City University.

### Plasmid

A human TOCA1 cDNA was amplified by PCR using a pCS2+MT-hToca1-
wt plasmid (Addgene plasmid #33030) as the template and cloned into
a pEGFP-C1 vector (Clontech, Takara Bio Inc.) to construct the pEGFP-
C1-TOCA1 encoding N-terminally EGFP-tagged TOCA1 (EGFP-TOCA1).
The EGFP-TOCA1 cDNA was also subcloned into the CSII-CMV-MCS-
IRES2-Bsd (to construct the CSII-CMV-EGFP-TOCA1-IRES2-Bsd plasmid). The CSII-CMV-MCS-IRES-Bsd lentivirus expression vector and the
packaging plasmid (pCAG-HIVgp, pCMV-VSV-G) were kindly provided
by Dr. H. Miyoshi (BioResource Center, RIKEN).

### Cell culture and lentivirus production and infection

Human umbilical vein endothelial cells (HUVECs) and human lung
fibroblasts (hLFs) were obtained from Lonza. Human pericytes from
placenta (hPlPCs) were purchased from PromoCell. HUVECs expressing GFP (GFP-HUVECs) were purchased from Angio-Proteomie.
HUVECs and GFP-HUVECs were cultured in EGM-2 (Lonza) and used at
passages 4 to 5. hLFs were cultured in FGM-2 (Lonza) and used at
passages 4 to 5. hPlPCs were cultured in Pericyte Growth Medium 2
(PromoCell) and used at passages 4 to 6. Authentication for each cell
line was conducted by each manufacturer.

Production of and infection with recombinant lentivirus encoding
EGFP-TOCA1 were performed as we previously reported[12]. Briefly,
EGFP-TOCA1 lentivirus vectors were transfected with the packaging
plasmids (pCAG-HIVgp and pCMV-VSV-G) into 293 T cells using Lipo-
fectamine 3000 reagent according to the manufacturer's instructions
(Thermo Fischer Scientific). Five h after transfection, transfection
media were replaced with growth media (Dulbecco's modified Eagle's
medium with 10% fetal bovine serum (FBS)). After 48 h, the conditioned media were collected and centrifuged at 1500 x $g$ for 5 min.
Then, the supernatants were filtered through a 0.45-μm filter to
remove floating cells and debris and stored at −80°C until use. The
lentivirus encoding EGFP-TOCA1 were further concentrated using PEG-
it™ Virus Precipitation Solution (System Biosciences) according to the
manufacturer's protocol and stored at −80°C until use. HUVECs at the
third or fourth passage were infected with lentivirus at the appropriate
multiplicities of infection and used at passages 5-6.

### Antibodies

The antibodies used were as follows: mouse anti-human CD31 (WM59,
BioLegend, 303102), rabbit anti-human PDGFRβ (Y92, OriGene), rabbit
anti-GOLPH4 (Abcam, ab28049), rabbit anti-VE-cadherin (D87F2, Cell
Signaling, 2500), goat anti-collagen type IV (SouthernBiotech, 1340-
01), rabbit anti-laminin (Abcam, ab11575), rabbit anti-NG2 (Millipore,
AB5320), rat anti-mouse CD31 (MEC13.3, BD Pharmingen, 552074), rat
anti-ICAM2 (3C4, BioLegend, 105601), goat anti-human ARPC2 (Novus
Biologicals, NB100-137), rabbit anti-human TRIP10 (CIP4, Proteintech,
10798-1-AP), mouse anti-human pMLC2 (Ser19) (Cell Signaling, #3675),
rabbit anti-human Moesin (EPR3864, Abcam, ab52490), Alexa Fluor
488-conjugated rabbit anti-ERG2 (EPR3864, Abcam, ab196374), Alexa
Fluor 488-conjugated goat anti-mouse IgG and goat anti-rat IgG
(Thermo Fischer Scientific), Cy3-conjugated goat anti-rabbit IgG and
goat anti-rat IgG (Thermo Fischer Scientific) and Cy3-conjugated
donkey anti-rabbit IgG (Thermo Fischer Scientific). Alexa Fluor 633-
conjugated goat anti-rat IgG and donkey anti-goat IgG (Thermo Fisher
Scientific), and Alexa Fluor 647-conjugated goat anti-rabbit IgG
(Abcam). The monoclonal antibody against murine PDGFRβ was purified from the supernatant of serum-free culture of the hybridoma cells
(clone APB5) by using HiTrap™ Protein G HP column (Cytiva).

### RNA extraction, qPCR and siRNA transduction

Total RNA was extracted from sub-confluent HUVECs and hPlPCs and
purified using an RNA extraction kit, Nucleospin RNA plus (MACHERRY-
NAGEL). Then, 500 ng of total RNA were reverse transcribed to cDNA
using PrimerScript RT Master Mix (Takara). Quantitative PCR (qPCR)
was performed with the QuantStudio 5 real-time PCR system (ThermoFisher Scientific) and Brilliant III Ultra-Fast SYBR Green QPCR Master
Mix (Agilent Technologies). The relative values were determined by the
$2^{-\Delta\Delta Ct}$ method, utilizing GAPDH as a control gene. Sequences of the
primers employed are shown in Supplementary Table 1.

Accell siRNAs against human *COL4A1* and *COL42*, termed Accell
Human *COL4A1* siRNA SMARTPool (E-011618-00-0005) and Accell

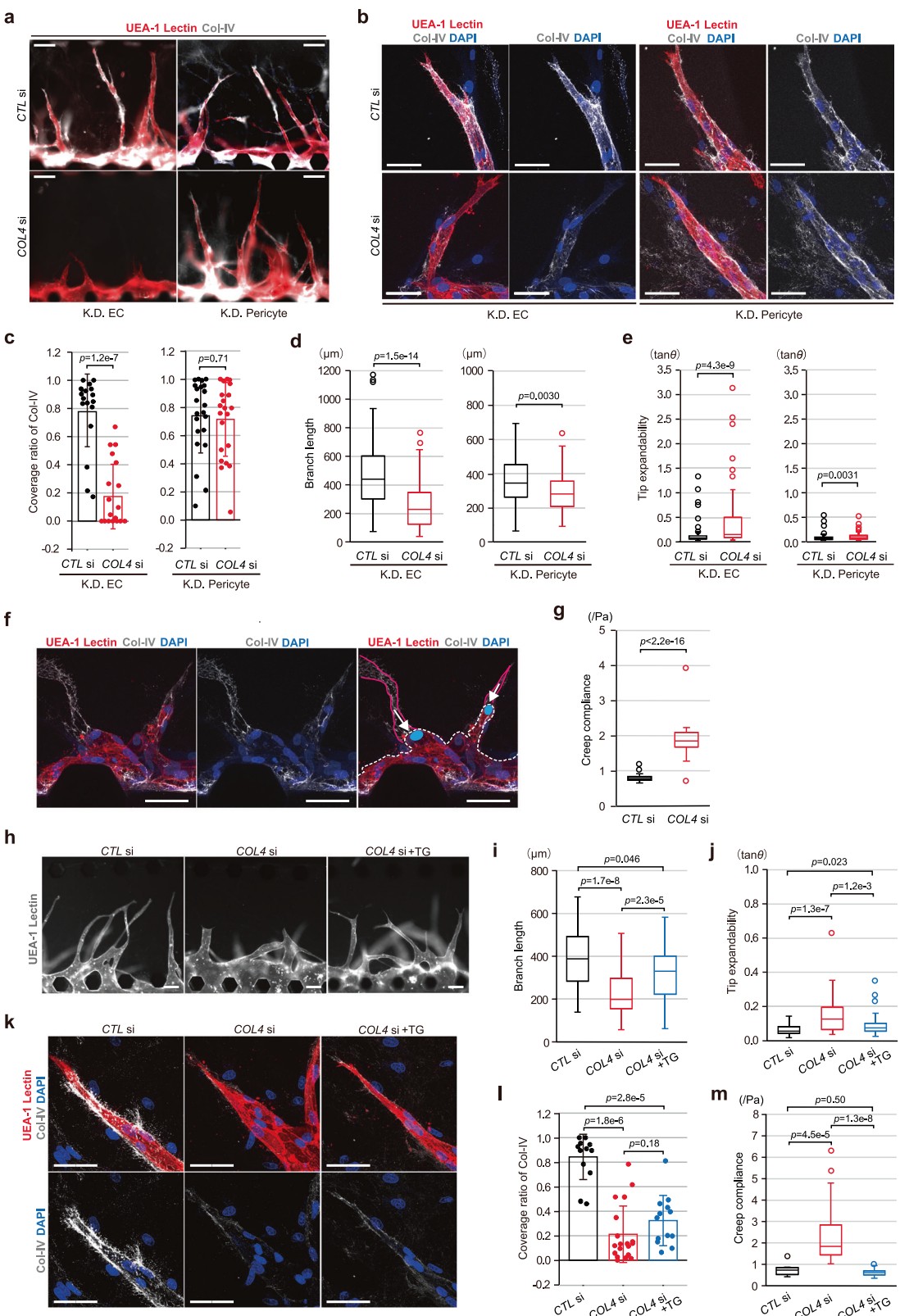

Human *COL4A2* siRNA SMARTPool (E-003645-00-0005), respectively, and negative control siRNA, termed Accell Non-targeting Control Pool (D-001910-10-05), were purchased from Horizon Discovery Ltd. Accell siRNA Delivery Media (B-005000-500, Horizon Discovery) with SingleQuots™ Supplements and with (for HUVECs) or without (for hPlPCs). Growth Factors (CC-4176, Lonza) served as the culture medium for the knock-down procedure. The supplements and the growth factors were diluted in the medium to the concentration recommended by the manufacturer. For double knock-down of *COL4A1* and *COL42*, cells were cultured with 0.5 (for HUVECs) or 1 (for PlPCs) μM *COL4A1* siRNA and 0.5 (for HUVECs) or 1 (for PlPCs) μM *COL4A2* siRNA. As the negative control, cells were cultured with 1 (for HUVECs) or 2 (for PlPCs) μM non-targeting siRNA. After a 24-h culture period, the cells were used for the experiments.

**Fig. 7 | Functional involvement of Col-IV deposition on VBM in on-chip angiogenic morphogenesis with pericytes. a, b** Representative fluorescent images (**a**) and magnified confocal z-projection images (**b**), with (*COL4* si) and without (*CTL* si) knock-down of both *COL4A1* and *COL4A2* in either ECs (K.D. EC) or pericytes (K.D. Pericyte) using siRNAs. **c–e** Knock-down effects. **c**, Quantification of Col-IV deposition on VBM (for ECs, *CTL* si: $n = 18$, *COL4* si: $n = 20$, from 3 independent experiments, for pericytes, *CTL* si: $n = 22$; *COL4* si: $n = 22$, from 3 independent experiments). **d**, Quantification of branch length (for K.D. ECs, *CTL* si: $n = 116$, *COL4* si: $n = 138$, from 3 independent experiments, for K.D. pericytes, *CTL* si: $n = 95$, *COL4* si: $n = 69$, from 3 independent experiments). **e** Quantification of Tip expandability (for K.D. ECs, *CTL* si: $n = 116$; *COL4* si: $n = 138$, from 5 independent experiments, for K.D. pericytes, *CTL* si: $n = 95$; *COL4* si: n = 69, from 4 independent experiments). **f** Representative confocal z-projection images showing regression process of angiogenic branches composed of *COL4* si ECs. In the right, Col-IV depositions on VBM (red lines), distal edge of the angiogenic branch (white dotted line) and tip cell nuclei (white arrows) were schematically superimposed on the confocal z-projection image in the left. **g** Evaluation of changes in perivascular stiffness in angiogenesis with *COL4* si ECs by PTM analysis based on Creep compliance (*CTL* si: n = 42; *COL4* si: n = 45, from 3 independent experiments). **h–m** Rescue experiments of *COL4* si in ECs by stiffening the gel with TG treatment. **h** Representative

fluorescent images showing the morphogenetic changes in angiogenic branches of *COL4A1* and *COL4A2* double knock-down ECs and pericytes and the rescue effects of TG treatment. **i, j** Quantification of the rescue effects by TG treatment, shown in box plots of branch length (**i**) and Tip expandability (**j**) (for **i** *CTL* si: $n = 53$; *COL4* si: $n = 53$; *COL4* si + TG: $n = 64$, from 4 independent experiments, for **j** *CTL* si: $n = 53$ from; *COL4* si: $n = 53$; *COL4* si + TG: $n = 64$, from 4 independent experiments). **k** Representative confocal z-projection images showing the differences in Col-IV deposition on VBM around areas where the de novo vascular lumen forms. **l** Quantification of Col-IV deposition on VBM (*CTL* si: $n = 13$; *COL4* si: $n = 19$; *COL4* si + TG: $n = 19$, from 3 independent experiments). **m**, Evaluation of changes in perivascular stiffness by the PTM method (*CTL* si: $n = 14$; *COL4* si: $n = 14$; *COL4* si + TG: $n = 18$, from 4 independent experiments). Data are expressed as means ± SD (**c, l**). All box plots show the interquartile range, with the middle line defining the median, and whiskers show the minimum and maximum values, excluding outliers (**d, e, g, i, j**). Outlier values are defined as being 1.5 times the interquartile range above and below the third and first quartile, respectively. Scale bars: 50 μm (**b, f, k**), 100 μm (**a, h**). Two-sided Mann-Whitney U test (**c–e, g, i, j**) and two-sided Mann-Whitney U test with a Bonferroni correction (**l, m**). Source data are provided as a Source Data file.

## On-chip angiogenesis assay

An on-chip angiogenesis assay was performed using the microfluidic device, as previously reported[12]. The device has five parallel flow channels with two sizes of wells at each side. Channels 2 to 4 have small wells for the introduction of hydrogel, culture media and cells, while channels 1 and 5 have larger wells on one side, thereby functioning as the main reservoir for the culture medium (Supplementary Fig. 1a). The microfluidic device was fabricated employing polydimethylsiloxane (PDMS) by conventional soft lithography and replica molding. A 100 μm thick SU-8 3050 (MicronChem, Westborough, MA) resist was photolithographically patterned on a silicon wafer and used as a master mold after treatment with trichloro-(1H, 1H, 2H-perfluorooctyl) silane (Sigma-Aldrich). A PDMS prepolymer [PDMS base: curing agent = 10: 1 (w/w)] (SILPOT184, Dow Corning Toray) cast on the mold was degassed in a vacuum chamber for 1 h and then incubated at 80 °C for 4–8 h. After incubation, the PDMS slab was separated from the mold and wells 1.5 mm or 6 mm in diameter were created by biopsy punch (Sterile Dermal Biopsy Punch, Kai Industries). Dust was removed from the PDMS slab and the glass bottom dish (3.5 cm diameter, Matsunami Glass) with adhesive tape, and the slab and glass surfaces were then treated with oxygen plasma for 40 s and 2 min, respectively. This allows irreversible bonding between the slabs after incubation at 80 °C overnight. The microfluidic device was sterilized by applying UV irradiation prior to each experiment.

For the angiogenesis assay (Supplementary Fig. 1b), hLFs suspended at a concentration of $5.0 \times 10^6$ cells/ml in phosphate buffered saline (PBS) including fibrinogen and collagen type 1 [2.5 mg/ml fibrinogen (Sigma-Aldrich) and 0.2 mg/ml collagen type 1 (Corning)] were prepared on ice. Immediately after addition of thrombin (0.5 U/ml, Sigma-Aldrich), hLFs were introduced into channel 1. Fibrin-collagen gels with no cells were introduced into channels 3 and 5. In some experiments, fibrin-collagen gels were crosslinked employing transglutaminase (TG, Sigma-Aldrich) and/or the concentration of fibrin in the gel was changed to the indicated dose. PBS with fibrinogen, collagen type 1 and TG [0.1 or 0.5 U/ml] was prepared and mixed with thrombin and $CaCl_2$ [2.5 mmol/l], and then the gels were similarly introduced into channels 3 and 5. After gelation at 37 °C in a 5% $CO_2$ incubator for 15 min, EGM-2 was injected into channels 2 and 4 and poured into all reservoirs, and the device was incubated at 37 °C in the 5% $CO_2$ incubator overnight to remove bubbles at the interface between the gel and the medium. Next, HUVECs suspended in EGM-2 at a concentration of $5.0 \times 10^6$ cells/ml were introduced into channel 4 and the device was tilted 90° for a few minutes to allow the cells to adhere to the gel-medium interface. Then, the device was incubated at 37 °C in a 5% $CO_2$ incubator to induce angiogenesis. For coculture

experiments with hPlPCs, $1.0 \times 10^6$ cells/ml of hPlPCs in EGM-2 were introduced into channel 4 in the same manner one day before the introduction of HUVECs. The medium was replaced on days 2 and 4 after the introduction of HUVECs. The devices were used for the following experiments, 3 or 4 days after introducing HUVECs, unless otherwise noted.

In the experiments to examine the effects of an inhibitor of ROCK, Y27632 (FUJIFILM Wako Pure Chemical Corp.), and an inhibitor of NMII, Blebbistatin (Merck), on on-chip angiogenesis, EGM-2 were replaced with new culture media including Y27632 (10 nM) or Blebbistatin (10 μM) or DMSO (Wako), twice at a 30-min interval one day after introducing HUVECs. Two days later, the devices were subjected to whole-mount staining.

## Murine retinal angiogenesis assay

Mice were housed in 12:12 light:dark light cycles at ambient temperature ranging between 20 °C and 23 °C and humidities between 30% and 70%. All experiments were carried out employing C57BL/6 WT mice (Japan SLC and CREA Japan). Sex was not considered in this study because sex differences were not thought to have significant effect on vascular development in the mice used in the experiment, which were less than 4 days old. Inductions of pericyte removal from murine retinal angiogenic branches and whole-mount preparations were performed as previously described[58]. Briefly, the monoclonal antibodies against murine PDGFRβ (APB5) [1 mg/ml in PBS] were intraperitoneally injected once on postnatal day 1 (P1). The eye was extracted on the indicated postnatal days after perfusion fixation with 4% paraformaldehyde (PFA), followed by additional fixation with 4% PFA for 30 min. Finally, the retinal cup was dissected from other parts of the eye and then subjected to whole-mount staining.

## Whole-mount staining

Angiogenic branches in the on-chip angiogenesis assay were fixed with 4% PFA (Wako) at 4 °C for 1 h. After being extensively washed with PBS, the primary antibodies (1:500) in permeabilization/blocking buffer [PBS with 0.1% TritonX-100 (Sigma-Aldrich) and 1% bovine serum albumin (BSA) (Sigma-Aldrich)] were injected into all channels, twice at a 30-min interval, and then incubated overnight at 4 °C. After at least 3 washes with wash buffer [PBS with 0.1 % TritonX-100], appropriate secondary antibodies (1:1000) in permeabilization/blocking buffer were added and replaced with fresh buffer solutions 30 min later, followed by overnight incubation at 4 °C. After extensive washing with the aforementioned wash buffer, nuclei were stained with DAPI (Dojin) in PBS at 4 °C for 2 h. Specifically, in experiments to stain the components of VBM (Col-IV and laminin), no detergents containing Tritonx-

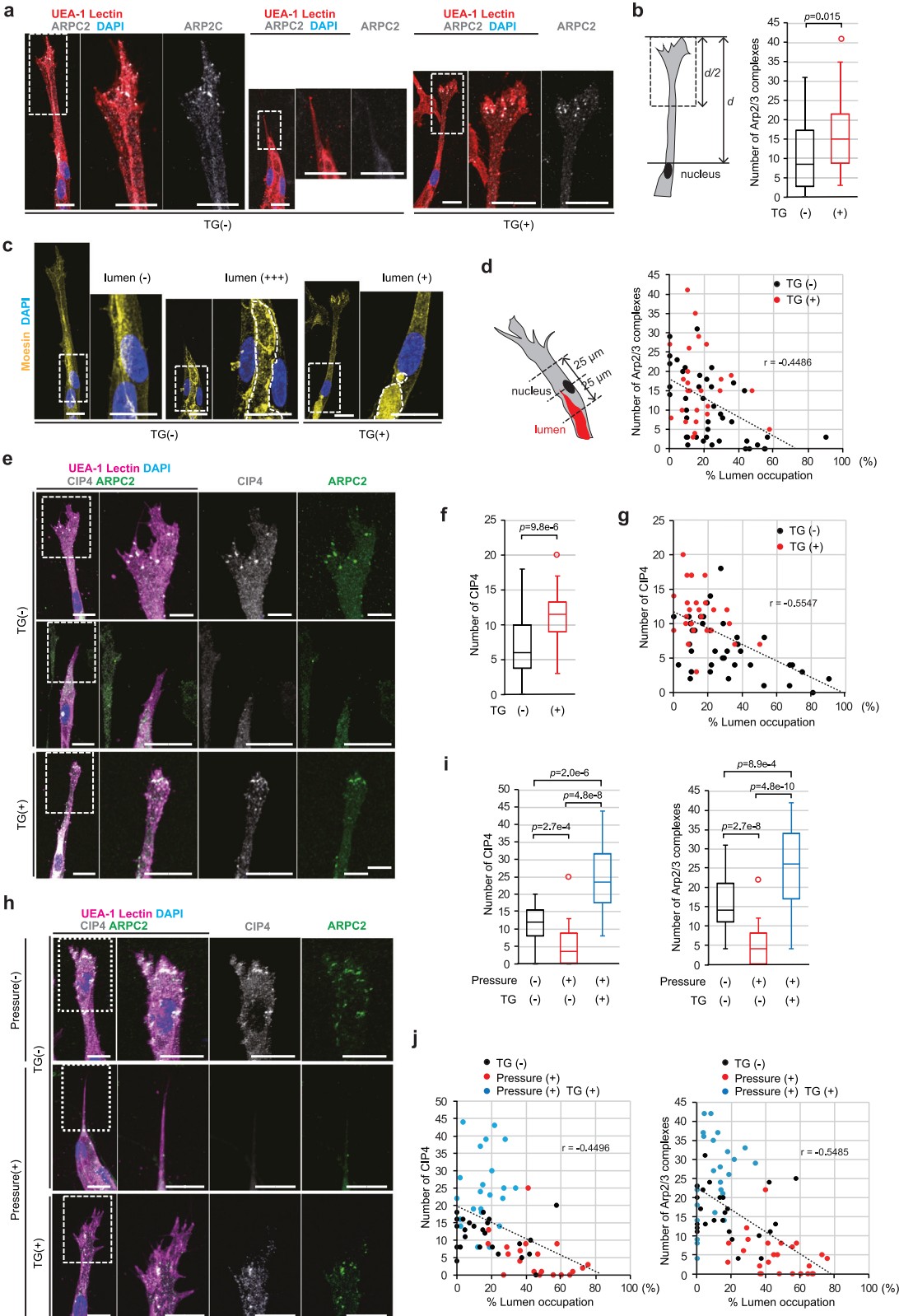

100 were used for staining and washing buffers, in order to avoid cell membrane permeabilization. To detect the EC membrane, FITC-conjugated UEA-1 Lectin [10 µg/ml] (Vector) was added to the primary antibody buffer.

For the murine retinal angiogenesis assay, the extracted retinal cup was additionally fixed with 4% PFA at 4 °C overnight. After being extensively washed with PBS, the retinal cup was processed in a manner similar to that for immunostaining of the on-chip angiogenesis assay. In some experiments, FITC-conjugated BS-1 Lectin [10 µg/ml] (Vector) in permeabilization/blocking buffer was added for detection of the retinal vasculature. If deemed necessary, the nuclei were visualized with DAPI.

**Fig. 8 | Loss in localization of F-BAR protein clusters and Arp2/3 complexes at the leading front of tip ECs in relation to lumen development. a–d** Relationships between Arp2/3 complex localization patterns around the tips of branches and the nearby lumen development in on-chip angiogenesis without (−) or with (+) TG treatment of the gel. **a** Representative z-projection confocal images, showing different localization patterns of Arp2/3 complexes. Areas within squares bounded by white dots are shown magnified on the right. **b** Quantification of Arp2/3 complex localization (TG (−): n = 42 branches; TG (+): n = 26 branches, from 3 independent experiments). d indicates the distance from the nuclear edge to the tip.
**c** Z-projection confocal images, showing different extents of lumen development (Lumen, (−): none, (+++): well developed, (+): less developed), which corresponds to images of **a** (See also Supplementary Fig. 14). **d** Distribution plot showing relationship between Arp2/3 complex localization and lumen development (TG (−): n = 42 branches, TG (+): n = 26 branches, from 3 independent experiments).
**e–g** Relationships between CIP4 localization patterns around the tips of branches and the nearby lumen development in on-chip angiogenesis without (−) or with (+) TG treatment of the gel. **e** Representative z-projection confocal images, showing different localization patterns of CIP4. Areas within squares bounded by white dots are shown magnified on the right. **f, g** Quantification of CIP4 localization (**f**) and

distribution plot (**g**) showing relationship between CIP4 localization and lumen development (**f, g** TG (−): n = 42 branches; TG (+): n = 26 branches, from 3 independent experiments). **h–i** Changes in localization patterns of CIP4 and Arp2/3 complexes around the tip of on-chip angiogenic branches without or with TG treatment of the gel by intraluminal pressure loading. **h** Representative z-projection confocal images, showing different localization patterns of CIP4 and Arp2/3 complexes. Areas within squares bounded by white dots are shown magnified on the right. **i, j,** Quantification of CIP4 and Arp2/3 complex localization (**i**) and distribution plot (**j**) showing relationship between CIP4 or Arp2/3 complex localization and lumen development (**i, j,** for CIP4, Pressure (P) (−)/TG (−): n = 25 branches; P (+)/TG (−): n = 20 branches; P (+)/TG (+): n = 22 branches, from 3 independent experiments, for Arp2/3, P (−)/TG (−): n = 25 branches; P (+)/TG (−): n = 25 branches; P (+)/TG (+): n = 25 branches, from 3 independent experiments). All box plots show the interquartile range, with the middle line defining the median, and whiskers show the minimum and maximum values, excluding outliers (**b, f, i**). Outlier values are defined as being 1.5 times the interquartile range above and below the third and first quartile, respectively. Scale bars: 20 μm. Two-sided Mann-Whitney U test (**b, f**) and two-sided Mann-Whitney U test with a Bonferroni correction (**i**). Source data are provided as a Source Data file.

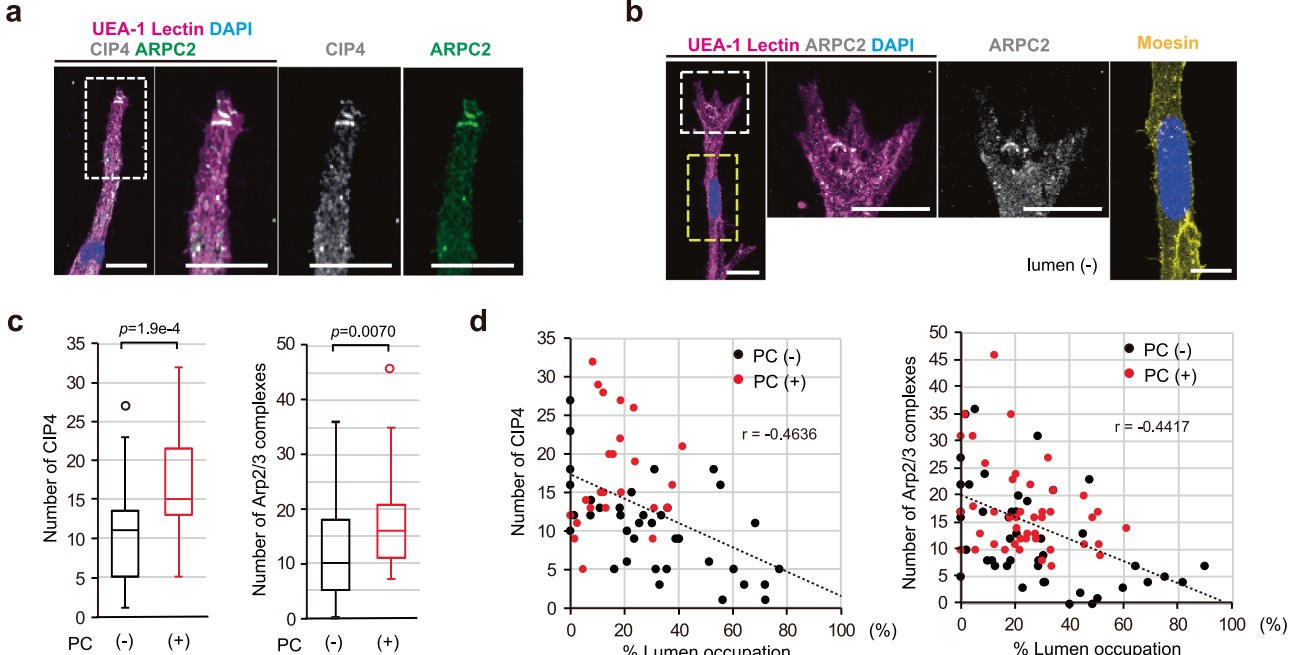

**Fig. 9 | Localization patterns of F-BAR protein clusters and Arp2/3 complexes at the leading front of tip ECs in the presence of pericyte. a–d** Relationship between CIP4 and Arp2/3 complex localization patterns around the tips of branches and the nearby lumen development in on-chip angiogenesis in the absence (PC (−)) or the presence (PC (+)) of pericytes. **a, b** Representative z-projection confocal images showing colocalization of CIP4 with Arp2/3 complexes in the presence of pericytes (**a**), and images showing relationship between Arp2/3 complex localization pattern and lumen development in the presence of pericytes (**b**). Regions indicated in squares bounded by white dots were magnified on the right in **a** and **b**, and the region within the square bounded by yellow dots is shown

magnified in the right panel of **b. c, d** Quantification of CIP4 and Arp2/3 complexes localization (**c**) and distribution plot showing relationship between CIP4 or Arp2/3 complexes localization around and lumen development (**d**) (**c, d**, for CIP4, PC (−): n = 37 branches; PC (+): n = 25 branches, for Arp2/3, PC (−): n = 45 branches; PC (+): n = 44 branches, from 3 independent experiments). All box plots show the interquartile range, with the middle line defining the median, and whiskers show the minimum and maximum values, excluding outliers (**c**). Outlier values are defined as being 1.5 times the interquartile range above and below the third and first quartile, respectively. Scale bars: 20 μm. Two-sided Mann-Whitney U test (**c**). Source data are provided as a Source Data file.

## Imaging and data processing

Time-lapse and streaming images were obtained using a fluorescent inverted microscope (IX83, Olympus) which is equipped with a CMOS camera (ORCA-Flash4.0, Hamamatsu Photonics) and is operated by MetaMorph software (Molecular Devices). Before time-lapse imaging of on-chip angiogenesis, the culture dish was filled with culture media, which caused the PDMS slab to be completely immersed in the culture medium, thereby stabilizing the media within the channel in terms of both temperature and volume. The microfluidic devices were maintained at 37 °C with 5% $CO_2$ using a stage top incubator (Tokai Hit

IX3W-STXG) throughout the experiments. For simultaneous imaging of EC movement and vascular lumen development in elongating angiogenic branches, differential interference contrast (DIC) images were obtained every 5 or 15 min for 12–99 h, using a 20x objective lens (UPLSAPO, 0.75 NA, Olympus). For the vascular distensibility test, DIC images were taken with a 40x objective lens (UPLSAPO, 0.75 NA, Olympus) before and after intraluminal hydrostatic pressure loading. For the particle tracking microrheology analysis, continuous fluorescent images of the beads were taken at 1000 frames per second for 30 s using a 100x objective lens (UPLSAPO100XO, 1.4 NA, Olympus)

under the control of the Stream Acquisition function of the Meta-Morph (Molecular Devices). Meticulous attention was paid, throughout the imaging process, to keeping the gels in the microfluidic devices at 37 °C using a 5% $CO_2$ stage top incubator and to prevent the gels from being exposed to any external vibrations.

In the on-chip angiogenesis assay, fluorescence imaging analyses were performed using a confocal laser-scanning microscope (TCS-SP8, Leica), equipped with a Hybrid Detector (Leica HyD, Leica), operated by LASX software version 2.0.1.14392 (Leica) or using a fluorescent inverted microscope (IX83, Olympus) equipped with a CMOS camera (ORCA-Flash4.0, Hamamatsu Photonics) and operated employing the MetaMorph software (Molecular Devices). For localization analysis of the Golgi apparatus, fluorescent confocal images were obtained at 1 μm intervals along the z-axis using a 20x objective lens (HC PL APO CS2, 1.40 NA, Leica). For the VBM analysis, localization patterns of F-BAR protein clusters, Arp2/3 complexes and pMLC2 as well as lumen development, fluorescent confocal images were obtained at 0.3 to 0.5 μm intervals along the z-axis using a 63x objective lens (Oil immersion, HC PL APO CS2, 1.40 NA, Leica). For morphological analyses, fluorescent images were obtained with a fluorescent inverted microscope (IX83, Olympus) using a 20x objective lens (UPLSAPO, 0.75 NA, Olympus).

In the murine retinal angiogenesis assay, for morphological analyses of the retinal vasculature, fluorescent images were obtained using a fluorescent microscope (BZ-X810, Keyence, 4x objective lens, Keyence) and, as necessary, fluorescent confocal images were obtained at 0.5 to 1 μm intervals along the z-axis using a confocal laser-scanning microscope (FluoView FV1200, Olympus, 20x objective lens, UPLSAPO, 0.75 NA, Olympus or oil-immersion 60x objective lens, UPLSAPO, 1.35 NA, Olympus), equipped with a GaAsP detector (Olympus). For localization analysis of the Golgi apparatus, images were obtained at 0.3 μm intervals along the z-axis using a confocal laser-scanning microscope (TCS-SP8, Oil immersion, 63x objective lens, HC PL APO CS2, 1.40 NA, Leica)), equipped with a Hybrid Detector (Leica HyD, Leica), operated using LASX software version 2.0.1.14392 (Leica).

All obtained images were processed employing the Fiji (http://fiji.sc.) image processing package, and the processed images were subjected to quantitative analysis, if deemed necessary.

**Interventions altering vascular intraluminal pressure**
To evaluate vascular distensibility (vascular distensibility test), 60 Pa of hydrostatic pressure was loaded onto the intraluminal space of angiogenic branches in the on-chip angiogenesis assay, as previously described in detail[12]. Briefly, a pressure-resistant tube was inserted into the well of channel 4 of the device, and the tube was connected to a 2.5 ml syringe (Supplementary Fig. 1d). The well at the opposite side of channel 4 was connected to a differential pressure gauge (DPG-01U, CUSTOM) via a resistant tube. The tube and syringe were then filled with the culture medium. Other wells were plugged using silicone (REPLISIL 22 N, dent-e-con) to maintain the channels as a closed circuit. The syringe was fixed to a universal laboratory stand, allowing adjustment of the degree of intraluminal hydrostatic pressure by changing the height of the syringe. DIC images of the vessels before and after pressure loading were taken when the pressure value became constant in the pressure gauge. To assess the influences of lumen and branch expansions on the dynamics of tip EC movement and branch elongation, after acquiring at least 12 h of time-lapse imaging data, similarly, ~40 Pa of hydrostatic pressure was loaded onto the intraluminal space of angiogenic branches to intentionally induce lumen and branch expansions, and time-lapse imaging was then restarted (Fig. 3a). After the imaging for another 3 h under additional intraluminal pressure loading, the pressure was released to relieve the lumen and branch expansions, followed by time-lapse imaging without intraluminal pressure loading for the following 20 h. In some

experiments, the load and release cycles of additional intraluminal pressure were repeated twice. For the control group, the same procedures were performed except for hydrostatic pressure loading and its release. Streaming DIC images were obtained during hydrostatic pressure loading and release to confirm the lumen and branch expansions and the cessation of these responses. Furthermore, after completion of the time-lapse imaging study, the actual values of the applied hydrostatic pressure were measured using the pressure gauge through a similar hydrostatic pressure loading procedure. The average values were $41 \pm 14$ Pa ($n = 6$). In experiments to evaluate the changes in localization patterns of the Arp2/3 complex and F-BAR protein by intraluminal pressure loading, ~40 Pa of hydrostatic pressure was loaded by placing capillary glass tubes (Hirshman) filled with the media (~11 mm height from the center of vascular lumen) at the outlet and the inlet of channel 4 (Supplementary Fig. 17a), similarly as we previously reported[12].

Using the DIC images obtained before and after hydrostatic pressure loading or its release, focusing on the vascular lumen area where the average branch diameter was ~10–20 μm, we manually traced the lumen structure over a range of 50 μm along the longitudinal axis (between the dotted lines shown in Figs. 2d, 3b, 6a). The traced images were binarized employing the Default thresholding function of the Fiji program, and the skeletonized image and subsequent distance map image were then obtained using the Skeletonize and Distance map functions, respectively, in Fiji. After the 8-bit skeletonized image had been divided by 255, the image was multiplied by the distance map image using the Image Calculator function in Fiji. The image thus obtained showed the distribution of the radius along the branches in pixel units. For the vascular distensibility test, relative changes in branch diameter were further calculated by dividing the radius distribution image after pressure loading by that before pressure loading using the Image Calculator and by averaging the distribution data for each analyzed vessel.

**Particle-tracking microrheology analysis**
In order to compare the local stiffness of the perivascular environment, the passive particle-tracking microrheology (PTM) method was used (Fig. 2b)[27]. In passive PTM, the thermal fluctuation of small probes in a material reflects the local mechanical properties of viscoelasticity. For this experiment, 1 μm fluorescent carboxylate-modified polystyrene beads (FluoSpheres, F8821, Molecular Probes), serving as probes, were pre-embedded in the fibrin-collagen gel of channel 3 of the microfluidic device. To achieve adherence of the probes to the fibrin fibers, the probes were treated with BSA solution [10 mg/ml in distilled water] overnight at room temperature, as previously described[59]. After rinsing with distilled water, the probes were suspended in fibrin-collagen solution at a concentration of 0.04 % solids and then introduced into the microfluidic device. The applicability of the PTM method for measuring fibrin-collagen gel viscoelasticity was validated using gels made employing various concentrations of fibrinogen with or without TG, a crosslinker (Fig. 2c). In order to compare the local stiffnesses of the perivascular environment in the on-chip angiogenesis assay, imaging of the fluorescent beads was performed at portions far from (at a distance of more than 10 μm distance) and near (within 5 μm) the angiogenic branches, both those with and those without pericytes on day 4 after the induction of angiogenesis (Fig. 6b). The imaging was also performed at portions near those of ECs with and without *COL4A1/A2* knock down in ECs (Fig. 7g). The tracked beads selected were located $15 \pm 5$ μm above the coverslip.

Movements of the beads on streaming images were tracked using TrackMate[60], a plug-in software program from Fiji, with which the trajectories of the beads are obtainable in sub-pixel resolution. The mean square displacement (MSD) for each bead was semi-automatically calculated from the tracking data, using an in-house

Java program. The MSD of each bead is expressed by the formula,

$$MSD(\tau) = \left\langle \{x(t+\tau) - x(t)\}^2 + \{y(t+\tau) - y(t)\}^2 \right\rangle \quad (1)$$

where $x(t)$ and $y(t)$ are the coordinates of the bead at time $t$, $\tau$ is the lag time, and the brackets mean time averaging. The MSD of the bead is proportional to a parameter termed the Creep compliance ($J(\tau)$) of the material:

$$J(\tau) = \frac{3\pi a}{2 k_B T} MSD(\tau) \quad (2)$$

where $a$ is the radius of the bead, $k_B$ is the Boltzmann constant and $T$ is the temperature. The static error of MSD was quantified as having a value of $7 \times 10^{-5}\ \mu m^2$ by the method shown in the relevant reference[61]. A few beads which showed directed motions were excluded from the analysis. The values of the MSD at lag time 10 s were used to calculate the creep compliance.

### Cell size analysis

In order to measure the distance from the EC nucleus to the distal and proximal edges of the cell, a mosaic analysis was performed in the on-chip angiogenesis assays with and without hPIPC, in which HUVECs, mixed with GFP-HUVEC at a ratio of 3:1 or 5:1, were introduced. Four days after starting cultivation, angiogenic EC branches and nuclei were visualized by staining with UEA-1 Lectin and DAPI, respectively. Confocal z-projection images of GFP-HUVECs in angiogenic branches were used for quantification (Supplementary Fig. 3a, b). In the quantitative analysis, the shapes of the cells and their nuclei were traced manually, and from the tracing data obtained, distances from the centroid of the nucleus to the distal and proximal edges of the cell were calculated semiautomatically, with the centroid of the nucleus being determined using Particle analysis, a function provided by the Fiji program.

### Quantitative analyses of the dynamics of EC movement and vascular lumen transformation, and their relationships in on-chip angiogenesis

To assess the movement dynamics of tip EC comprising the analyzed lumen, we manually tracked the centroid of the EC nucleus on DIC images obtained every 5 min by time-lapse imaging of the on-chip angiogenesis assay, using MTrackJ[62], a plugin program of Fiji, as shown in Fig. 1e. In some analyses, subsequently, the existence frequency of EC moving at the indicated speed (distance per 15 min) toward the direction of the tip (plus) or the bottoms (minus) of the branches was calculated. In the intervention study assessing vascular intraluminal pressure, in addition to tip ECs, the tips of the angiogenic branches were simultaneously tracked in a similar manner on DIC images obtained every 15 min. From the tracking data, we determined the average speed (/min) of individual tip ECs during specified periods after loading and the release of intraluminal pressure, as well as the average branch elongation for 3 h before and after intraluminal pressure loading and 3 h after its release. Furthermore, to quantitatively link the dynamics of EC movement to vascular lumen changes, two dimensional shapes of the vascular lumen were manually traced simultaneously on the same DIC images. From the traced images, lumen areas within 25 μm of the tip and bottom sides, from the centroid of the tracked tip EC nucleus, were measured at each timestep using a computer program devised employing Java (Fig. 1e). This analyzed lumen area was determined by reference to the average size of ECs, comprising on-chip angiogenic branches, as calculated in the cell size analysis (Supplementary Fig. 3a, b).

Based on these data, the dynamics of changes in the lumen area just before EC deceleration (point of deceleration in EC movement) were obtained. The deceleration point of EC movement was determined as follows. Local maximum time points were identified based on

the EC movement distance-time plot. The local maximum timepoint $T$ is defined as the point satisfying $y(T - 5\ min) < y(T)$ and $y(T) \geq y(T+5\ min)$, where $y(T)$ is the position of the EC at time $T$. Additionally, the following criteria were set to select the deceleration point from the local maximum points:

1. $y(T) - y(T - 50\ min) > 5\ \mu m$,
2. $y(T + 50\ min) - y(T) < 5\ \mu m$,
3. $y(T) - y(T + 15\ min) > 0\ \mu m$.

Then, the change in lumen area during the 30-min period before the deceleration point was calculated.

### Quantitative evaluations of angiogenic branch morphology

**Branch elongation.** Effects of the removal or addition of pericytes on angiogenesis were morphologically analyzed in terms of branch elongation and the circumferential growth of the branches in a murine retinal assay as well as in the on-chip assay. In the retinal assay, branch elongation was evaluated by an index of the average radius of the vascular network. To achieve this, an approximate ellipse of the retinal vascular network was calculated from a polygon of which the vertices were the vessel tips on the obtained fluorescent image, using the Fit Ellipse function of Fiji. Then, major and minor radii of the approximate ellipse were averaged. In the on-chip assay, branch elongation was evaluated as branch length based on straight-line distance ($l$, shown in Fig. 2g) from the base of the branch to the tip.

**Branch diameter.** To calculate the frequency distribution of the branch diameter, the fluorescent images in both assays were binarized by the Default thresholding program of Fiji. After dilation and erosion of the binarized image, a skeletonized image and subsequent distance map were obtained using the Skeletonize and Distance map functions, respectively, of Fiji. The branch diameter data were extracted from the processed image, in which an 8-bit skeletonized image divided by 255 was multiplied by the distance map image using the Image Calculator function of Fiji. Then, the existence probability of the indicated vessel diameter and the average value were calculated from the extracted data. The unit of obtained values was converted from pixels to micrometers. In the retinal assay, 400 μm circular areas around the optic disc and, in the on-chip assay, areas where ECs had invaded Channel 3, forming a sheet-like structure rather than a vessel structure, were omitted from the analysis.

**Tip expandability.** To more sensitively evaluate the extent of the angiogenic branch diameter around the tip, the parameter Tip expandability was selected. For this, we initially calculated the strait-line distance from the base of the branch to the tip (corresponding to branch length ($l$)), and then drew a line intersecting with it at the half-way point to determine both ends of the width of the branch. Tip expandability ($\tan \theta$) was calculated by the following formula (See also Fig. 2g),

$$\tan \theta = (l/2)/(d/2) = l/d \quad (3)$$

where $d$ is a distance between both ends of the width of the branch at the half-way point.

### Quantification for deposition of Col-IV and laminin on VBM

In the on-chip angiogenesis assay, we first confirmed deposition of Col-IV on VBM around the angiogenic branches, by measuring fluorescence intensity profiles along indicated lines in confocal images using the Plot Profile function of Fiji (Fig. 6e). Next, the extent of Col-IV and laminin depositions on VBM around the leading edges of angiogenic branches, where the de novo vascular lumen develops, were quantitatively evaluated using 2 parameters: coverage ratio and fluorescence intensity per volume. For analyses of both parameters, the abluminal surface of angiogenic EC branches from the tip of the vessel lumen to

the 25 μm proximal side were manually selected as a line on a confocal z-projection image (Fig.6e). Then, the coverage ratio *C* was calculated by the formula,

$$C = b/a \quad (4)$$

where *a* is the total length of the selected line, and *b* is the length which has the fluorescent intensity of Cy3 (Col-IV or laminin) over a threshold value on the selected line of the confocal z-stack images. We set 30 and 200 (A.U.) as the threshold values for Col-IV and laminin, respectively.

The other parameter, fluorescent intensity of Cy3 (Col-IV or laminin) per volume, was calculated, using confocal z-projection images, in which nuclei, angiogenic EC branches and Col-IV (or laminin) (Cy3) were visualized. Perivascular areas within 1 μm from the abluminal surface of an angiogenic EC branch from the tip of the lumen to the 25 μm proximal were determined and then analyzed by manually tracing the abluminal surfaces of the branches using the Fiji program. In this analyzed area, the total amount of Cy3 fluorescence intensity was calculated from a confocal z-projection image obtained by projecting 5 serial images over the central part of the branch. The fluorescent intensity of Cy3 per volume in the confocal z-projection image was calculated by the formula,

$$I_{avg} = S/V \quad (5)$$

where *S* is total amount of the intensity of Cy3 in the area and *V* is the volume of the projected area.

In the murine retinal angiogenesis assay, the extents of Col-IV and laminin depositions on VBM around the leading edge of the angiogenic branch, where the de novo vascular lumen develops, were quantitatively evaluated using one parameter, the coverage ratio, essentially as described above. Note that for the parameter analysis, the abluminal surfaces of angiogenic EC branches from the proximal end of the nucleus of the tip EC to the 31 μm proximal side were manually selected as a line on a confocal z-projection image (Supplementary Fig. 11c). The analyzed area was determined based on the data showing the average distance between the tip of the lumen and the proximal end of the nucleus of the tip EC to be ~31 μm, a value similar to those in control retinas and in APB5-treated retinas (Supplementary Fig. 11b).

### Correlation analysis between vascular diameter and lumen diameter in murine retinal angiogenesis

The correlation between the outer diameter of an angiogenic branch and its lumen diameter was assessed in the murine retinal vasculature on P4. Abluminal and luminal surfaces of the angiogenic branch were visualized by whole mount staining with BS-1 Lectin and ICAM2, respectively. For this analysis, 3 angiogenic branches which did not bifurcate or merge with other branches around the leading front and 3 immediately proximal branches were selected, at random, on the confocal z-projection images. Next, 3 positions in each of the selected branches were defined, with distal, intermediate and proximal positions being selected in the angiogenic branch as well as in the immediately proximal branch. Then, the outer diameters of vessels and the lumen diameters at each position were measured manually.

### Quantification of localization patterns of the Golgi apparatus relative to the nucleus

In both the on-chip angiogenesis assay and the retinal angiogenesis assay, the relative position of the Golgi apparatus to the nucleus of the tip EC was examined using confocal z-projection images showing whole-mount staining of angiogenic branches with or without removal of pericytes. The relative position of the Golgi apparatus was visually classified as proximal or distal depending on whether its centroid was positioned in the tip side of the branch to the center of the nucleus, as

shown in the drawing for Fig. 5g. The ratios of tip ECs classified as either proximal or distal to the total number of tip ECs were calculated.

### Quantification of localization patterns of F-BAR protein clusters and Arp2/3 complexes and their relationships to lumen development in on-chip angiogenesis

The localization patterns of both F-BAR protein clusters and Arp2/3 complexes at the tips of angiogenic branches in on-chip assays were assessed under different culture conditions. Using confocal z-projection images showing whole-mount staining of angiogenic branches, the clusters and complexes were manually counted in the distal half branch (*d/2*) corresponding to the area ranging from the tip to the half-way point of the distance between the tip and the distal edge of the nucleus of tip EC (*d*) (Fig. 8d). To further examine relationships between their localization patterns and the degree of lumen development around the nucleus of tip EC, we estimated % Lumen occupancy. Using confocal z-projection images showing whole-mount staining of angiogenic branches, we manually traced the lumen and branch areas of the branch segment within the range between 25 μm distal and proximal to the center of the nucleus of tip EC (Fig. 8d). % Lumen occupation was then calculated by dividing the lumen area by the branch area, followed by evaluation of the relationship with the localization patterns of either F-BAR protein clusters or Arp2/3 complexes.

### Statistics and reproducibility

Statistical analyses were performed using R. Branch lengths, branch diameters, lumen area changes, Tip expandability, relative changes in branch diameter with pressure loading, lumen area change, the coverage ratios of Col-IV and laminin, the fluorescence intensities per volume of Col-IV and laminin, values of creep compliance in the PTM analysis and numbers of CIP4, TOCA1-EGFP and Arp2/3 complexes in on-chip angiogenesis or in murine retinal angiogenesis were compared between groups using the Mann-Whitney U test. Branch elongation, tip EC speed, Branch length, Tip expandability and the coverage ratio of Col-IV in on-chip angiogenesis were compared among 3 different sampling points using the Mann-Whitney U test with a Bonferroni correction. Lumen area changes before versus after pressure loading and release in on-chip angiogenesis were analyzed using the Wilcoxon signed rank test. Changes of branch elongation in on-chip angiogenesis were compared among 3 different sampling points using the Wilcoxon signed rank test with a Bonferroni correction. The extents of the radial growth of murine retinal angiogenic branches were compared between groups employing the Welch's *t* test. Frequency distributions of branch diameters and of EC movement speeds, in either on-chip angiogenesis or murine retinal angiogenesis, were compared between groups applying the Kolmogorov-Smirnov test. Rates of the Golgi apparatus having the distal position in on-chip angiogenesis or in murine retinal angiogenesis were compared between groups employing the Chi-squared test. Values of creep compliance in the PTM analysis in on-chip angiogenesis were compared among groups using the Mann-Whitney U test with a Holm correction. Correlations between % Lumen occupation and numbers of CIP4, EGFP-TOCA1 and Arp2/3 complexes were evaluated by applying the Pearson correlation coefficient. $p < 0.05$ was considered to indicate a statistically significant difference. Data are presented as means ± SD.

All experiments (except a part of experiments for the Supplementary Figs.) in this study were independently repeated more than three times with similar results. For all experiments, no statistical method was used to predetermine sample size, and no data were excluded from the analyses. For on-chip angiogenesis assays, all cultured cells were randomly assigned to different experimental groups in individual assays, and data were collected from all branches or cells observed on each chip, or randomly selected branches on each chip. For murine retinal analysis, mice or eyes to be analyzed were randomly

selected for each experimental group. Blinding during data collection was not performed since the same investigator did all experimental processes from group allocation to data analysis.

## Reporting summary

Further information on research design is available in the Nature Portfolio Reporting Summary linked to this article.

## Data availability

The authors declare that all data supporting the findings of this study are available within the main text and Supplementary Materials. Raw data to generate all graphs within the figures are provided as Source Data File. Source data are provided with this paper.

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

## Acknowledgements

We thank H. Miyoshi (BioResource Center, RIKEN) for the CSII-CMV-MCS-IRES2-Bsd lentiviral expression vector and the packaging plasmids. We are grateful to the core facility of the International Research Center for Medical Sciences (IRCMS), Kumamoto University, and to the core facility of the Frontier Science Research Center, University of Miyazaki. We thank M. Uchikawa and C. Esumi, for excellent technical assistance. We also thank R. Masuya for excellent support in the core facility of the Frontier Science Research Center, University of Miyazaki. This work was supported by Core Research for Evolutional Science and Technology (CREST) from Japan Science and Technology (JST) under Grant Number JPMJCR14W4 to K.N.; by Grants-in-Aid for Scientific Research (B) to K.N. (19H04446 and 24K03267), for Exploratory Research to K.N. (26670394), for Scientific Research for Young Scientists (Start-up) to Y.H. (22K20743), for Scientific Research (C) to K.N. (16KT0173) from the Japan Society for the Promotion of Science; research grants from the Naito Foundation to K.N., from Astellas Foundation for Research on Metabolic Disorders to K.N., from The NOVARTIS Foundation (Japan) for the Promotion of Science to K.N., from the TERUMO LIFE SCIENCE FOUNDATION to K.N., from SENSHIN Medical Research Foundation to K.N. and Y.H., from the NAKATANI FOUNDATION for advancement of measuring technologies in biomedical engineering to K.N., from the KOSÉ Cosmetology Research Foundation to K.N., from The Uehara Memorial Foundation to K.N., from the Princess Takamatsu Cancer Research Fund to K.N. and Daiichi Sankyo Foundation of Life Science to K.N.

## Author contributions

Y.H. and K.N. conceived and designed the research. Y.H., S.H., M.H. and Y.A. performed in vitro experiments including on-chip angiogenesis assays and analyzed the data. Y.H., K.N., S.O., Y.S., Y.A. and H.O. performed in vivo experiments using the murine retinal angiogenesis model, with help from Y.O. and A.U., and analyzed the data. S.F. and K.N. generated the EGFP-TOCA1 lentivirus. Y.H. performed in silico quantification analyses with help from M.H. and S.N., while Y.H., T.M. and K.N. interpreted the data and wrote the manuscript with input from all authors.

## Competing interests

The authors declare no competing interests.
