## [Transparent Peer Review file · Nature Communications]

Biomechanical control of vascular morphogenesis by the surrounding stiffness

Corresponding Author: Professor Koichi Nishiyama

Version 0:

Reviewer comments:

Reviewer #1

(Remarks to the Author)

In this study, Hanada and co-authors aimed to address how two distinct morphogenic processes – cell migration and lumen formation/expansion – are integrated and coordinated to control the rate of vessel elongation during sprouting angiogenesis. Using an on-chip angiogenesis assay system where it is possible to control intraluminal pressure and perivascular stiffness, the authors found an inverse relationship between vessel elongation and lumen expansion, whereby the increase in lumen area or diameter correlates with a decrease in vessel elongation. The authors also showed that the presence of pericytes restricted lumen dilation that is induced by increased intraluminal pressure. This finding is supported by *in vivo* studies using the mouse retina where the removal of pericytes resulted in decreased vessel diameter and increased vessel length. The authors next explored whether force balance between intraluminal pressure and abluminal/ECM stiffness controls vessel elongation versus lumen expansion. When ECM stiffness is increased, there is a reduction in vessel diameter and an increase in vessel length. They further showed that pericytes and Collagen IV increase the stiffness of ECM that ECs are embedded, and depletion of Collagen IV expression by ECs leads to increased vessel diameter and reduced vessel length. With these findings, the authors conclude that the integration of different mechanical signals – intraluminal pressure and abluminal stiffness – is required to control the balance between vessel elongation and lumen expansion.

Significance

The strength of this study is the use of the *in vitro* on-chip angiogenesis model to control specific forces – intraluminal pressure and ECM stiffness – to understand the interplay and integrating of these two mechanical forces. This study contributes to the understanding of how mechanical forces regulate blood vessel morphogenesis and shaping vessel networks, a field that is understudied.

Comments

1. The authors propose that pericytes increase the stiffness of the vascular basement membrane (VBM) to control the rate of vessel elongation. However, in certain vessels such as the zebrafish intersegmental vessels, vessel elongation and lumen expansion occur in the absence of pericytes (pericytes coverage occurs after ISVs are completely lumenized). How do you think that vessel elongation and lumen expansion are mechanically coordinated in blood vessels that lack pericytes?

2. A previous study by Stratman et al., 2017 in *Development* has shown that mural cells control the diameter of the dorsal aorta whereby the dorsal aorta narrows as mural cells wrap around the vessel. They further showed that mural cells produce Collagen IV and fibronectin to increase the stiffness of the VBM i.e. control vessel elasticity or distensibility, which is what the authors is showing in this paper. This work should be cited.

3. Although the authors indicate and discuss the importance of intraluminal pressure and ECM stiffness in controlling vessel diameter, they should also discuss the cell autonomous role of ECs in response to intraluminal pressure and the regulation of lumen expansion and vessel diameter. For example, in Gebala et al., the authors showed ECs to activate actomyosin contractility in the apical cortex to retract pressure-induced membrane blebs, thereby controlling lumen expansion and

vessel size. In Kondrychyn et al., the authors showed that ECs generate a balanced network of branched and linear actin at the cell cortex that increases cortical stiffness and resistance against blood pressure, loss of which leads to membrane blebbing and increased lumen/vessel diameter. In these studies, in which endothelial cell contractility and/or stiffness is perturbed, ECM stiffness and the surrounding tissues were not sufficient to resist intraluminal pressure.

4. This study would also benefit from a time course study with closer examination of the dynamics of pericyte coverage/movement as the vessels elongate and lumen forms. Are pericytes at the site of the vessel that undergoes lumen formation? Do pericytes themselves generate contractile forces to prevent lumen expansion? Where is Col-IV deposition relative to pericyte location on the vessel?

5. The authors show that there is a decrease in Col-IV deposition in the VBM in the absence of pericytes and also when Col4a1/a2 are knocked-down in ECs. This can also be interpreted as Col-IV is produced by both cell types – ECs and pericytes – and contribute to VBM stiffness. Is there a greater loss in VBM stiffness and increase in vessel distensibility when Col-IV is not produced by both cell types?

6. I do not understand the message in lines 36 and 37 “.... Col-IV deposition on the VBM from EC driven by pericytes accounts for spatiotemporal perivascular stiffening”. A similar sentence exists in lines 319 to 320. What is meant by “driven by pericytes”? Are the authors suggesting that pericytes increase Col-IV deposition by ECs or that pericytes remodel Col-IV in the VBM? This sentence should be clarified.

7. Does EC number influence the vessel elongation rate and/or lumen diameter? Is there a correlation between EC number and vessel diameter?

8. Fig. 5C: I do not understand how the authors quantified the “rate” rate of Col-IV coverage. The method that is used, b/a (where b is the length of Col-IV measured and a is the total length of the vessel measured) gives a ratio, not rate. The word “rate” gives the impression that an element of time has been measured but this is not the case.

9. Can the authors explain why a probability plot is used, instead of a frequency plot, in Figures 1g, 1j, 4j and 5j.

10. The title of the Fig. 2 is “Vascular lumen expansion inhibits directional tip EC movement and branch elongation during angiogenesis”. It is undisputed that there is a correlation between these two cellular processes, but there is no solid experimental manipulation to prove this. Given that it is possible to control intraluminal pressure, can the authors perform cycles of pressure changes e.g. increase-decrease-increase, and observe lumen expansion and branch elongation? This experimental design will provide more support to the authors’ statement.

11. I would preferably like to see better resolution of images showing retina vessels. Is this possible?

Reviewer #2

(Remarks to the Author)

This paper investigates an important and timely topic regarding how the physical microenvironment regulates vascular morphogenesis. The authors utilize a multidisciplinary approach to investigate this topic by combining in vitro microfluidic chip studies with murine retinal angiogenesis in vivo experiments. In this paper, the authors focus specifically on factors regulating lumen diameter and branch elongation during morphogenesis. The authors show that pericytes deposit collagen-IV around ECs, increasing the local matrix stiffness. This increase in matrix stiffness limits lumen expansion and alters EC branch elongation. These results help to address a current gap in the field regarding the role of pericytes and their mechanical regulation of vascular processes.

This paper demonstrated a few significant drawbacks, mainly in Figures 1-3. The experimental design in the first few figures does not address the hypothesis the authors are seeking to answer in the text (relationship between lumen expansion and elongation) but answers a different hypothesis not discussed, which is investigating the role of pericytes on these metrics. The beginning figures and interpretations attempt to convince the reader that there is a relationship between lumen expansion and EC branch elongation, but the data is really showing that pericytes regulate both metrics. Furthermore, the authors consistently draw conclusions of causation when only correlative data is shown. The role of pericytes is neglected in the first half of the paper and not discussed by the authors until Figures 4 and 5. The paper then jumps from neglecting the role of pericytes in the data to making pericytes the main character of the story. Once this is realized by the authors, the more compelling data and interpretations lay in Figure 4 and 5.

Figures 4 and 5 showed compelling data with improved experimental design and hypotheses. This reviewer would have liked to see more mechanistic experiments investigating how increased stiffening of the surrounding area due to the shown increase in deposition is sensed by ECs and converted into biochemical signals that regulate the changes in lumen expansion and branch elongation. The data shown here only scratches the surface of uncovering how stiffness controls vascular morphogenesis.

Major Comments:

- When summarizing Figure 1, the authors write “These results suggest a causal relationship between branch elongation and circumferential lumen expansion.” However, the authors are reporting correlation data for these metrics. The loss of function experiment in this figure is involving +/- pericytes and resulting changes in the branch morphology. The authors are

neglecting the role pericytes and PDGFR- β might play in regulating these morphological metrics and are assuming A (elongation) causes B (diameters) when it is possible C (pericytes) may be regulating both A and B which is shown by this data.

- It is not immediately clear how the graph in Figure 2c shows EC deceleration immediately after lumen expansion. When the lumen area first increases ~90 minutes, the EC moving distance increases and accelerates (slope of line increases). What is the reason for choosing these time points for the dotted lines? It would be helpful for these lines to show when the lumen expansion and deceleration begins as this is unclear and would provide more clarity on where the written interpretation is derived from.
- Stating that the data in Figure 2 shows an inhibitory effect between these two variables is a misleading interpretation that is not supported by this data. There does appear to be an inverse correlation, but there is not sufficient data to claim lumen expansion is inhibiting EC tip movement. There is no real “experiment” being run in this figure to uncover mechanisms such as inhibition. Just observations and correlation data.
- The same issue occurs in Figure 3 as in Figure 1. The independent variable in these experiments is the presence of pericytes, which is resulting in changes in EC lumen diameter and impacting the movement of ECs. From this data alone, the authors claim that “lumen emergence followed by its expansion inhibits forward directional movement of nearby ECs” is not substantiated by the presented data.
- The conclusion of Figure 4 on lines 241-244 is a critical turning point of the paper where the data supports stiffening of the surrounding matrix due to pericytes, which provides a plausible, potential explanation for changes in EC movement. However, there is a dire lack of mechanistic studies investigating how this stiffening causes these changes in ECs.

Minor Comments:

- Fig 1b – cross-sectional image of lumen would be helpful here from supplement
- The size of the standard deviation bars in Figure 1g and 1j stood out to me as a concern. Is there a plausible explanation for this degree of variability?
- Fig 2c – Typos on both y-axes and there are two i lines and no ii line.

Version 1:

Reviewer comments:

Reviewer #1

(Remarks to the Author)

The authors have addressed most of my concerns raised in the first submission. They have performed many additional experiments to strengthen the claim that increased luminal expansion decelerates endothelial cell migration, and that vascular basement stiffening reduces luminal expansion to increase branch elongation.

Below are comments to further improve the current status of the manuscript in light of the new data.

1. In the new experiments with Y27632 and Blebbistatin, which the authors used to manipulate lumen formation/expansion, it is unclear when these drugs were added. Is it before or after the initiation of new vessel sprouts/branch? Given that actomyosin-dependent contractility is required for driving membrane protrusions and endothelial cell migration, manipulating this pathway can inhibit vessel sprout/branch formation independent of lumen expansion. In fact, in the representative image shown in Fig. 3j, it appears that Y27632-treatment prevented the formation of new vessel sprouts/branches. How can the authors differentiate the effects of myosin II on endothelial cell migration and lumen expansion?

2. Annotation of Fig. 3 can be improved.

- Fig. 3e, right panel: since there is no change in pressure in this experimental setting, there is no need to include “load” and “release” in the x-axis as this is rather confusing.

- Fig. 3f and g and Fig. 3h and i look similar. To differentiate the two, I recommend writing “after pressure load” and “after pressure release”, respectively, above the graphs to differentiate between the two conditions (especially since this information is buried in the very long figure legend).

- Fig. 3i, the x-axis legend should be “after pressure release”, not “after pressure loading”.

3. Figure 7a. Representative images of collagen IV (COL-IV) coverage by immunostaining in CTLsi EC and CTLsi pericytes look different, with decreased staining observed in CTLsi pericytes. Given that these are control experiments, shouldn't the degree of COL-IV be similar? However, quantification of coverage ratio of COL-IV as shown in Fig. 7c shows similar coverage between the two controls. Perhaps a better representative image(s) should be used.

4. There is inconsistency in the representative images used to demonstrate vessel morphology after COL4si in EC in Fig. 7a and Fig. 7b. In Fig. 7a, there appears to be very few vessel sprouts/branches that form and in cases where there are vessel branches, they are short. But in Fig. 7b, the representative vessel branch in COL4si EC condition is long with many endothelial cells (as shown by DAPI staining). Which is the more representative image?

5. The experiment where the authors attempt to rescue ECM stiffness in COL4A1/A2 DKD ECs by TG treatment is very nice. Did the authors measure Creep compliance in these experiments to demonstrate that stiffness was successfully increased and to complement the existing data (Fig. 7h – l)?

6. In the revised manuscript, the authors show that increasing ECM stiffness by TG treatment (Fig. 8 a, b, e f, g) as well as co-culturing with pericytes increase CIP4 and Arp2/3 complex localization at the leading edge of tip cells, leading to cell migration. Although the authors also correlated CIP4 and Arp2/3 complex number with the presence of lumen, can the authors also perform a pressure loading experiment (similar to that in the new Fig. 3) with or without TG treatment and examine CIP4 and Arp2/3 localization? This would convincingly demonstrate that the two mechanical cues (luminal pressure and ECM stiffness) converge molecularly at the level of Arp2/3 activity to coordinate lumen expansion and vessel elongation.

7. The length of all figures legends is far too long so much so that it does not serve the purpose in aiding the reader to interpret the data since it is difficult to find the relevant information. The figure legends should be more concise and avoid repetitive phrases.

8. The word 'rate' on page 13, line 303 should be changed to 'ratio'.

9. Extended Dat Fig. 12a show an image of laminin staining around vessels in the presence (+) and absence (-) of pericytes. In the representative image of laminin staining of vessels without pericytes (middle image of lower row), it looks like UEA-1 Lectin staining, not laminin staining. Please check.

Reviewer #3

(Remarks to the Author)

This is a brief assessment of the manuscript titled "Biomechanical Control of Vascular Morphogenesis by the Surrounding Stiffness" by Hanada and colleagues. My primary focus is on the issues raised by Reviewer #2, who was unavailable for the re-review of the revised manuscript.

In my assessment, the authors have conducted a thorough and substantial revision. The inclusion of additional experimental results has effectively addressed most of the concerns raised by Reviewer #2. In particular, the new data on the interplay between matrix stiffness, endothelial sprout extension, and lumen formation provide a stronger foundation for the pericyte-related findings presented later in the manuscript. These improvements have also resolved the disconnect between the two halves of the original submission, as highlighted in Reviewer #2's comments.

Another major concern raised by Reviewer #2—the correlative nature of many of the findings—has been partially addressed. The authors have made significant efforts to tackle this issue, and it is important to acknowledge the additional data and the insights gained from it. While a conceptual gap remains between matrix stiffness/hydrodynamic pressure and the downstream biochemical signals, resolving this within the scope of a revision would be challenging.

Other concerns expressed by Reviewer #2 have been adequately addressed through revisions to the manuscript, as outlined in the (admittedly lengthy) rebuttal letter.

Overall, my primary remaining criticism is that the manuscript is not particularly easy to read. The figure legends and Discussion section are overly lengthy and would benefit from trimming and revision to enhance clarity.

Version 2:

Reviewer comments:

Reviewer #1

(Remarks to the Author)

I would like to thank the authors for their efforts in addressing all my concerns and performing the experiments that I suggested after the first round of revision. The manuscript is now in a much-improved state for publication.

While reading the manuscript, I found two very minor issues/questions that should be addressed:

- There is a typo in the y-axis legend of Fig. 1h. "chnage" -> "change"
- Fig. 9a and Fig. 9b: does one panel show the presence of pericyte and the other the absence of pericyte. If so, which is pericyte (-) and which is pericyte (+)?

Reviewer #3

(Remarks to the Author)

The authors have addressed my questions. Moreover, the revised manuscript is clearly improved and it is easier to follow the flow of the data. However, the authors might want to consider further trimming of the discussion, which is still rather long.

Overall, the article will enhance our conceptual understanding of the processes controlling endothelial sprouting and thereby makes an important contribution to the field.

Replies to the reviewer's comments

Reviewer #1 (Remarks to the Author):

In this study, Hanada and co-authors aimed to address how two distinct morphogenic processes – cell migration and lumen formation/expansion – are integrated and coordinated to control the rate of vessel elongation during sprouting angiogenesis. Using an on-chip angiogenesis assay system where it is possible to control intraluminal pressure and perivascular stiffness, the authors found an inverse relationship between vessel elongation and lumen expansion, whereby the increase in lumen area or diameter correlates with a decrease in vessel elongation. The authors also showed that the presence of pericytes restricted lumen dilation that is induced by increased intraluminal pressure. This finding is supported by in vivo studies using the mouse retina where the removal of pericytes resulted in decreased vessel diameter and increased vessel length. The authors next explored whether force balance between intraluminal pressure and abluminal/ECM stiffness controls vessel elongation versus lumen expansion. When ECM stiffness is increased, there is a reduction in vessel diameter and an increase in vessel length. They further showed that pericytes and Collagen IV increase the stiffness of ECM that ECs are embedded, and depletion of Collagen IV expression by ECs leads to increased vessel diameter and reduced vessel length. With these findings, the authors conclude that the integration of different mechanical signals – intraluminal pressure and abluminal stiffness – is required to control the balance between vessel elongation and lumen expansion.

Significance

The strength of this study is the use of the in vitro on-chip angiogenesis model to control specific forces – intraluminal pressure and ECM stiffness – to understand the interplay and integrating of these two mechanical forces. This study contributes to the understanding of how mechanical forces regulate blood vessel morphogenesis and shaping vessel networks, a field that is understudied.

First, we thank reviewer #1 for the supportive views on our study and insightful

comments. We agree that addressing the concerns raised would greatly improve our manuscript. Thus, we have performed the additional experiments suggested and have revised the manuscript accordingly.

Comment 1:

The authors propose that pericytes increase the stiffness of the vascular basement membrane (VBM) to control the rate of vessel elongation. However, in certain vessels such as the zebrafish intersegmental vessels, vessel elongation and lumen expansion occur in the absence of pericytes (pericytes coverage occurs after ISVs are completely lumenized). How do you think that vessel elongation and lumen expansion are mechanically coordinated in blood vessels that lack pericytes?

Response to Comment 1:

Thank you for your advice to reconsider the importance of perivascular cells including pericytes in forming vascular basement membrane (VBM) during the early stage of angiogenic morphogenesis. As noted by reviewer #1, in certain vessels such as intersegmental vessels (ISVs) and cranial vessels during zebrafish development, pericyte coverage reportedly begins after the establishment of a fully lumenized vascular network (*Development* 143(8):1328039, 2016; *PLOS Genetics*, 2020 <https://doi.org/10.1371/journal.pgen.1008800>). However, we believe that some perivascular cells can contribute to VBM formation in coordination with ECs by acting as a substitute for pericytes, and thereby properly and spatiotemporally maintain the perivascular stiffening, leading to integral control of vessel elongation and lumen development via a biomechanical mechanism. One of the perivascular cell types would appear to be fibroblast-like cells. Our idea is supported by prior reports. Using *in vivo* live imaging of zebrafish embryos, AM Rajan and colleagues identified fibroblast-like cells expressing *colla2* and *col5a1* genes constituting VBM to be adjacent to elongating endothelial sprouts of ISV prior to pericyte coverage, and some of the perivascular fibroblast-like cells were observed to give rise to pericytes at the later stage (*PLOS Genetics*, 2020 <https://doi.org/10.1371/journal.pgen.1008800>). These authors also conducted genetic cell ablation experiments, in which the perivascular fibroblast-like

cells were found to be functionally involved in VBM formation, thereby maintaining lumen and vessel diameter. In addition, recent studies using single cell RNAseq analyses in mice demonstrated the presence of similar perivascular fibroblast-like cells, which were not characterized by classical pericyte markers, in adult mouse brain (*Cell* 174, 999-1014.e22, 2018; *Cell* 174, 1015-1030.e16, 2018; *Science* 352, 1326-1329, 2016; *Nature* 554, 475-480, 2018), possibly suggesting the perivascular fibroblast-like cells to be similarly involved in VBM formation during cranial vascular development.

Therefore, in the discussion section of the revised manuscript, we mentioned the possible importance of these perivascular fibroblast-like cells in regulating the perivascular environment via VBM formation during the early stage of angiogenic morphogenesis as follow:

1. Added p 22, line 525 - p23, line 533: Pericytes are not always just adjacent to ECs in the angiogenic branches, as observed in cranial vessels and intersegmental vessels during development in zebrafish embryos^{41,42}. However, *in vivo* live imaging-based analyses in zebrafish embryos⁴² together with single cell RNAseq analyses in mice⁴³⁻⁴⁶ suggest that perivascular cells such as fibroblast-like cells, that are adjacent to ECs and later give rise to pericytes, can act as a substitute for pericytes to form VBM in coordination with ECs. This may allow other perivascular cells, similar to pericytes, to maintain perivascular stiffening and integrally control vessel elongation and lumen development via biomechanical processes.

Comment 2:

A previous study by Stratman et al., 2017 in Development has shown that mural cells control the diameter of the dorsal aorta whereby the dorsal aorta narrows as mural cells wrap around the vessel. They further showed that mural cells produce Collagen IV and fibronectin to increase the stiffness of the VBM i.e. control vessel elasticity or distensibility, which is what the authors is showing in this paper.

This work should be cited.

Response to Comment 2:

Thank you for this very helpful suggestion. We completely agree with reviewer #1's opinion. Therefore, we cited a report published in 2017 by Stratman et al. (*Development* 144(1):115-127, 2017) as a reference showing that, *in vivo*, mural cells including pericytes contribute to VBM formation by depositing Col-IV via interactions with ECs, thereby controlling vessel diameter. Accordingly, we changed and added the relevant text to the result and the discussion sections of the revised manuscript, as follows:

1. *p12, line 265 - 267* → *p15, line 336 - 339*: Mural cells including pericytes were previously reported to stimulate vascular basement membrane (VBM) formation via interactions with ECs in vascular morphogenesis *in vitro*³⁴ as well as in zebrafish dorsal aorta development *in vivo*³⁵.
2. *p12, line 272 - 274* → *p 15, line 344-346*: Studies producing both *in vitro* and *in vivo* data have raised the possibility that Col-IV of VBM may make a more important contribution to determining branch diameter than other VBM components^{34,35}.
3. *p16, line 366 - 368* → *p 22, line 512 - 515*: Mural cells including pericytes are well known to contribute to vascular maturation and maintenance during development, processes which are closely associated with mural cell-mediated VBM formation, including Col-IV, as well as its function^{35,37-40}.
4. Added *p 22, line 523-525*: Similarly, proper Col-IV deposition on VBM via mural cell-EC interactions reportedly controls vessel diameter in zebrafish dorsal aorta development³⁵.

Comment 3:

Although the authors indicate and discuss the importance of intraluminal pressure and ECM stiffness in controlling vessel diameter, they should also discuss the cell autonomous role of ECs in response to intraluminal pressure and the regulation of lumen expansion and vessel diameter. For example, in Gebala et al., the authors showed ECs to activate actomyosin contractility in the apical cortex to retract pressure-induced membrane blebs, thereby controlling lumen expansion and vessel size. In Kondrychyn et al., the authors showed that ECs generate a balanced network of branched and linear actin at the cell cortex that increases cortical stiffness and resistance against blood

pressure, loss of which leads to membrane blebbing and increased lumen/vessel diameter. In these studies, in which endothelial cell contractility and/or stiffness is perturbed, ECM stiffness and the surrounding tissues were not sufficient to resist intraluminal pressure.

Response to Comment 3:

Thank you for these very helpful comments. We agree with reviewer #1's opinion that, in addition to perivascular environmental factors including intraluminal pressure and ECM stiffness, the properties and functions of ECs themselves comprising vascular wall structures also contribute to the mechanical balance controlling lumen and vessel diameters, and that one of their autonomous roles in ECs is mediated by cortical actomyosin contractility which increases cortical stiffness and resistance against intraluminal pressure. Therefore, we noted the importance of these observations in the discussion part of the revised manuscript by citing previous reports (*Nat Cell Biol* 18: 443–450, 2016; *Nat. Commun* 11:5476, 2020), as follows:

1. Added p24, line 563 - p25, line 575: In addition to these surrounding physical properties, cell-autonomous roles of the constituent cells are also important for determining the mechanical balance necessary for controlling lumen expansion during the formation of not only epithelial but also endothelial tubular structures^{24,28–30}. Active actomyosin contractility in the apical cortex reportedly increases cortical stiffness and causes retraction of intraluminal pressure-induced membrane blebbing, thereby limiting lumen and branch expansions during angiogenic morphogenesis in zebrafish^{24,30}. Indeed, in the present study, suppression of apical actomyosin contractility by inhibitors of either ROCK or NMII induced both branch and lumen expansions around the tips of ECs undergoing on-chip angiogenesis and reduced branch elongation. Whether and, if so, how the cell autonomous mechanism is involved in an unbalanced mechanical state, leading to abnormal angiogenesis in diseased states, remains to be investigated.

Comment 4:

This study would also benefit from a time course study with closer examination of the dynamics of pericyte coverage/movement as the vessels elongate and lumen forms. Are

pericytes at the site of the vessel that undergoes lumen formation? Do pericytes themselves generate contractile forces to prevent lumen expansion? Where is Col-IV deposition relative to pericyte location on the vessel?

Response to Comment 4:

We appreciate reviewer #1's insightful comments regarding the role of pericytes in controlling lumen and VBM formation, which will be among the important research topics to be tackled in the near future. We fully agree that examining the dynamic relationships among pericyte coverage/movement, vessel elongation and lumen formation would very likely further reveal the roles of pericytes in angiogenic morphogenesis.

Replies to the following three specific questions are given below.

Q1. Are pericytes at the site of the vessel that undergoes lumen formation?

Q2. Do pericytes themselves generate contractile forces to prevent lumen expansion?

Our snapshot imaging analyses of whole-mount immunostaining samples show that in most, but not all, branches of murine retinal angiogenesis, pericytes were observed around the tip of the angiogenic branch that undergoes *de novo* lumen formation (Extra Fig. 1a-c, See below). Similarly, in most branches of on-chip angiogenesis, pericytes are present around the tips of branches where *de novo* lumen formation is observed (Extra Fig. 2a, b, See below). However, in comparison to murine retinal angiogenic branches (Extra Fig. 1a), pericytes do not appear to wrap around the EC branch structure well but rather exist adjacent to ECs directly in contact with the nearby ECs, at least to some extent (Extra Fig. 2a). The difference in these patterns of pericytes suggests that direct contractile forces exerted by pericytes, for instance, previously reported to control vessel tonus (*Nature* 2006, 443:700-704, *Nat Med* 23:733-741, 2017) after the developmental stage, are not the main underlying mechanism accounting for pericyte actions preventing excess lumen expansion at the site of *de novo* lumen formation seen in both retinal and on-chip angiogenesis. On the other hand, the similarity in these patterns, corresponding to where pericytes exist, suggests that interaction between ECs and pericytes with or without their direct contact may be important for controlling the emergence of the lumen and subsequent lumen expansion. Indeed, we have obtained a clue to unveiling a

candidate molecular axis that may mediate EC-pericyte interaction and thereby prevent excess lumen expansion via Col-IV deposition on VBM although data were not shown. Further elucidation is anticipated in the near future as mentioned in the discussion parts of the previous (*p 16, line 380 - p 17, line 384*) and revised manuscripts (*p 23, line 539 - 542*). Furthermore, as discussed above, investigating the dynamics of pericyte movement/coverage in relation to the dynamics of lumen development and VBM formation using time-lapse imaging, may allow us to identify novel pericyte-dependent angiogenic mechanisms in greater detail.

Q3. Where is Col-IV deposition relative to pericyte location on the vessel?

As shown in Fig.6c, e, f, and g of the revised manuscript, in most branches of both retinal and on-chip angiogenesis with pericytes, Col-IV deposition on VBM reached the position around the distal of tip EC nucleus, where the *de novo* lumen had developed and where most pericytes were located (Extra Fig.1b-d, 2a, b, See below). The data suggest that pericytes may contribute to lumen development and Col-IV deposition on VBM via interaction with nearby ECs, as described above. Indeed, in the revised manuscript, the on-chip angiogenesis studies using either *COL4A1/A2* knock-down (KD) ECs or *COL4A1/A2* KD pericytes clearly showed the importance of enhanced VBM deposition of EC-derived Col-IV in the presence of pericytes, leading to efficient branch elongation by preventing excessive lumen and branch expansions (See also response to comment 5 below, Fig.7a-e including new data).

Extra Fig. 1

Extra Fig. 1: Relationships in localization patterns among pericytes, lumen and Col-IV deposition in murine retinal angiogenesis

Representative z-projection images of whole-mount immunostaining of P4 murine retina. **a**, Pericyte (PDGFR β) is located even at the leading edge of EC branch (CD31) that is partially composed of tip EC, and wraps the EC branch. Magnified views of areas squares bounded by dots in *top panels* are shown in *bottom panels*. **b**, Distal end of lumen (evaluated by ICAM2) reaches to the portion around the tip of EC branch (UEA-1 Lectin). **c**, Example images showing pericytes (NG2) either present (*top panel*) or absent (yellow arrows in *bottom panel*) in the tip of the branch newly undergoing lumen formation (ICAM2). **d**, Pericyte (denoted by Desmin) is adjacent to the tip of EC branches (UEA-1 Lectin), around which Col-IV deposition on VBM is simultaneously observed. Scale bars: 50 μ m for **a**, **b**, **c**, and 20 μ m for **d**.

Extra Fig. 2

Extra Fig. 2: Relationships in localization patterns between pericytes and lumen in on-chip angiogenesis

Whole-mount immunostaining of on-chip angiogenic branches at the day 4 after the cultivation. **a**, Representative z-projection images. In most cases, pericytes (PDGFR β , white arrowheads) are localized adjacent to tip ECs (red arrowheads) of the EC branches (UEA-1 Lectin). **b**, Representative z-projection images of the tip of EC branch (UEA-1 Lectin) (Z projection) and the corresponding

confocal *x-y* slice images at several *z*-positions (Slice). Distal end of lumen (evaluated by Moesin) reaches to the position around the tip EC nucleus (red arrowheads), to which pericyte (identified as a UEA-1 Lectin-negative cell, white arrowheads) is localized close. Scale bars: 50 μ m for **a** and 20 μ m for **b**.

Comment 5:

The authors show that there is a decrease in Col-IV deposition in the VBM in the absence of pericytes and also when Col4a1/a2 are knocked-down in ECs. This can also be interpreted as Col-IV is produced by both cell types – ECs and pericytes – and contribute to VBM stiffness. Is there a greater loss in VBM stiffness and increase in vessel distensibility when Col-IV is not produced by both cell types?

Response to Comment 5:

Thank you for this helpful comment. In the previous manuscript, based solely on the result showing that on-chip angiogenesis using *COL4A1/A2*-KD ECs and wild-type pericytes could reproduce the effects seen in the absence of pericytes, we concluded that the EC-derived Col-IV deposition is important for stiffening VBM surrounding the branch that undergoes *de novo* lumen formation, leading to promotion of branch elongation via reductions of lumen and branch distensibility. However, as the reviewer pointed out, Col-IV produced by pericytes may also be involved in VBM formation and stiffness.

Therefore, as described in the revised manuscript, we investigated the effects of knocking Col-IV down in pericytes and compared it with those of knocking it down in ECs. We found that, in the case using wild-type ECs and *COL4A1/A2*-KD pericytes, Col-IV deposition on the VBM surrounding the branch undergoing new lumen formation was not reduced although a slight decrease in branch elongation and a slight increase in branch diameter around the tip were observed (Fig.7a-e including new data, Extended Data Fig. 13b, See below). From these results, we concluded that EC-derived Col-IV deposition, which is enhanced via interactions with pericytes (as discussed above), is the main contributor to VBM formation and stiffening around the tips of branches, and also contributes to promoting angiogenesis. We again highlighted this point in the revised manuscript. We further investigated whether the influences exerted by *COL4A1/A2* KD in ECs could be reversed by intentional stiffening of the surrounding ECM by TG

treatment. We found that the ECM stiffening resulted in partial recovery of the morphological changes in response to *COL4A1/A2* KD in ECs in terms of branch elongation and expansion with no changes in Col-IV deposition, confirming that VBM deposition of Col-IV derived from ECs leads to the promotion of branch elongation by acting as a mechanical barrier (new data, Fig.7h-1, See below).

As noted in the revised manuscript, we did not examine the effects of knock-down of Col-IV in both ECs and pericytes for the following reasons: one is that *COL4A1/A2* KD in pericytes did not significantly affect Col-IV deposition on VBM around the tips of branches (Fig. 7b, c including new data, See below). Another reason is that *COL4A1/A2* KD solely in ECs already resulted in highly marked effects on branch elongation and expansion. The reason for *COL4A1/A2* KD in pericytes inducing changes in branch length and diameter, albeit modestly, remains to be investigated.

Fig. 7

Fig. 7: Functional involvement of Col-IV deposition on VBM in angiogenic morphogenesis

Evaluation of functional involvement of Col-IV deposition on VBM in on-chip angiogenesis with pericyte coculture. **a, b**, Representative fluorescent images (**a**) and magnified confocal z-projection images (**b**) on Day 3 visualized by UEA-1 Lectin (ECs), Col-IV and DAPI (nuclei), with (*COL4 si*) and without (*CTL si*) knock-down of both *COL4A1* and *COL4A2* in either ECs (K.D. EC, left panels) or pericytes (K.D. Pericyte, right panels) using siRNAs. **c**, Quantification of Col-IV deposition on VBM (for ECs, *CTL si*: n=18 from 2 devices, *COL4 si*: n=20 from 3 devices, examined during the course of 2 independent experiments, for pericytes, (*CTL si*: n=22 from 2 devices; *COL4 si*: n=22 from 2 devices, examined during the course of 2 independent experiments). **d**, Quantification of branch

length (for K.D. ECs, *CTL* si: n=116 from 11 devices, *COL4* si: n=138 from 7 devices, examined during the course of 3 independent experiments, for K.D. pericytes, *CTL* si: n=95 from 4 devices, *COL4* si: n=69 from 5 devices, examined during the course of 3 independent experiments). **e**, Quantification of Tip expandability, an index reflecting branch diameter, around the tip of the branch (See also Fig.2g) (for K.D. ECs, *CTL* si: n=116 from 11 devices; *COL4* si: n=138 from 7 devices, examined during the course of 5 independent experiments, for K.D. pericytes, *CTL* si: n=95 from 4 devices; *COL4* si: n=69 from 5 devices, examined during the course of 2 independent experiments). **f**, Representative confocal z-projection images showing regression process of on-chip angiogenesis composed of *COL4* si ECs. In the *right panel*, Col-IV depositions on VBM (red lines), distal edge of the angiogenic branch (white dotted line) and tip cell nuclei (white arrows) were schematically superimposed on the confocal z-projection image in the *left panel*. **g**, Evaluation of changes in perivascular stiffness in on-chip angiogenesis with *COL4* si ECs by PTM analysis based on one parameter, Creep compliance (*CTL* si: n=41 from 4 devices; *COL4* si: n=45 from 4 devices, examined during the course of 3 independent experiments). **h-l**, Rescue experiments of *COL4* si in ECs by stiffening the gel with TG treatment in the on-chip angiogenesis assay. Representative fluorescent images showing the changes in ECs comprising angiogenic branches, with pericytes coculture, by knock-down of both *COL4A1* and *COL4A2* in ECs and the rescue effects on both produced by TG treatment (**h**). Box plots of branch length (**i**) and Tip expandability (**j**, See also Fig. 2g) in on-chip angiogenesis of ECs treated with *control* (*CTL*) si or both *COL4A1* and *COL4A2* si (*COL4* si) and without and with TG treatment (+TG) (for **i**, *CTL* si: n=53 from 4 devices; *COL4* si: n=53 from 5 devices; *COL4* si + TG: n=64 from 5 devices, examined during the course of 2 independent experiments, for **j**, *CTL* si: n=53 from 4 devices; *COL4* si: n=53 from 5 devices; *COL4* si + TG: n=64 from 5 devices, examined during the course of 2 independent experiments). Representative confocal z-projection images showing the differences in Col-IV deposition on VBM around areas where the *de novo* vascular lumen forms in the three groups, using *CTL* si ECs, *COL4* si ECs and *COL4* si ECs with TG treatment (**k**), and quantification based on one parameter, the coverage ratio of Col-IV (*CTL* si: n=13 from 3 devices; *COL4* si: n=19 from 4 devices; *COL4* si + TG: n=13 from 4 devices, examined during the course of 2 independent experiments). Data are expressed as means \pm SD for **c** and **l**. All box plots show the interquartile range, with the middle line defining the median, and whiskers show the minimum and maximum values, excluding outliers, for **d**, **e**, **g**, **i** and **j**. Outlier values are defined as being 1.5 times the interquartile range above and below the third and first quartile, respectively. Scale bars: 50 μ m for **b**, **f** and **k**, 100 μ m for **a**, **h**. Mann-Whitney U test for **c-e** and **g** and Mann-Whitney U test with a Bonferroni correction for **i**, **j** and **l**.

Extended Data Fig. 13b

Extended Data Fig. 13b: Expressions of *COL4A1* and *COL4A2* genes and their knock-down using siRNAs in ECs and pericytes

Quantitative PCR data for gene expressions of *COL4A1* and *COL4A2* in HUVECs and hPIPCs, treated with *COL4A* si (siRNAs for both *COL4A1* and *COL4A2* genes) or control (*CTL*) si for 2 days. Data

are expressed as the means of the relative values of $\Delta\Delta CT$ to one in CTL si (n=2 for each group, examined during the course of 2 independent experiments).

Therefore, according to the newly obtained data, we added relevant findings and rewrote the text, with the addition of the new and related data in Fig. and in Extended Data Fig., as follows:

1. Added parts of Fig. 7 and corresponding legend
2. Added Extended Data Fig. 13b and corresponding legend
3. *p 2, line 34 - 37* → *p 2, line 35 - 38*: This process was counter-regulated the presence of pericytes, which induced perivascular stiffening by promoting the deposition of EC-derived collagen-IV (Col-IV) on the vascular basement membrane (VBM), thereby preventing excessive lumen expansion.
4. *p 5, line 98 - 100* → *p 5, line 105 - 108*: We further demonstrated enhanced deposition of EC-derived type-IV collagen (Col-IV) on vascular basement membranes (VBM) to account for spatiotemporal perivascular stiffening by pericytes.
5. *p 13, line 305 - 307* → *p 16, line 378 - 380*: We knocked down both *COL4A1* and *COL4A2* in either ECs (*COL4A1/A2* DKD ECs) or pericytes using siRNA (Extended Data Fig. 13b), then carried out the on-chip angiogenesis assay with pericytes.
6. *p 14, line 313 - 316* → *p 17, line 387 - 393*: On the other hand, the double knock-down in pericytes did not significantly reduce Col-IV deposition on VBM around branches, even around the tips, although branch elongation was slightly reduced and both branches and lumens also displayed slight expansion (Fig. 7a-e). These data revealed the functional involvement of Col-IV deposition on VBM in angiogenic morphogenesis, and also suggest that mainly EC-derived Col-IV was involved in the formation and function of VBM at least around the tip portion where the *de novo* vascular lumen develops.
7. *p 14, line 318 - 321* → *p 17, line 401 - 404*: These results collectively suggest that spatiotemporally-enhanced deposition of EC-derived Col-IV on VBM in the presence of pericytes controls perivascular stiffness appropriately, which promotes branch elongation by physically restricting both lumen and branch expansions.

8. *p 15, line 335 - 338 → p 20, line 472 - 475:* We further demonstrated that pericytes mediate proper Col-IV deposition by ECs onto the VBM, which accounts for spatiotemporal perivascular stiffening and the resultant reduction in vascular distensibility.
9. *p 16, line 374 - 375 → p 36, line 520 - 523:* We further revealed that VBM stiffening via enhanced deposition of Col-IV secreted dominantly from ECs causes branch thinning in the presence of pericytes, which is consistent with a previous result showing the importance of Col-IV VBM³⁴.
10. *p 26, line 601 - 608 → p 36, line 833 - 841:* Accell siRNA Delivery Media (B-005000-500, Horizon Discovery) with SingleQuots™ Supplements and with (for HUVECs) or without (for hPIPCs). Growth Factors (CC-4176, Lonza) served as the culture medium for the knockdown procedure. The supplements and the growth factors were diluted in the medium to the concentration recommended by the manufacturer. For double knockdown of COL4A1 and COL42, cells were cultured with 0.5 (for HUVECs) or 1 (for PIPCs) μM COL4A1 siRNA and 0.5 (for HUVECs) or 1 (for PIPCs) μM COL4A2 siRNA. As the negative control, cells were cultured with 1 (for HUVECs) or 2 (for PIPCs) μM non-targeting siRNA. After a 24-hour culture period, the cells were used for the experiments.

Comment 6:

I do not understand the message in lines 36 and 37 “.... Col-IV deposition on the VBM from EC driven by pericytes accounts for spatiotemporal perivascular stiffening”. A similar sentence exists in lines 319 to 320. What is meant by “driven by pericytes”? Are the authors suggesting that pericytes increase Col-IV deposition by ECs or that pericytes remodel Col-IV in the VBM? This sentence should be clarified.

Response to Comment 6:

We apologize for the unclear expression in the previous manuscript stating that “Col-IV deposition on the VBM from EC driven by pericytes accounts for spatiotemporal perivascular stiffening”. As reviewer #1 inferred the meaning and as we discussed above,

we wanted to accurately convey to our readers that VBM deposition of Col-IV produced by ECs increases, which contributes to perivascular stiffness:

Therefore, we rewrote the sentence to clarify the meaning in the revised manuscript, as follows:

1. *p 2, line 35 - 37* → *p 2, line 35 - 38*: This process was counter-regulated the presence of pericytes, which induced perivascular stiffening by promoting the deposition of EC-derived collagen-IV (Col-IV) on the vascular basement membrane (VBM), thereby preventing excessive lumen expansion.
2. *p 5, line 98 - 100* → *p 5, line 105 - 108*: We further demonstrated enhanced deposition of EC-derived type-IV collagen (Col-IV) on vascular basement membranes (VBM) to account for spatiotemporal perivascular stiffening by pericytes.
3. *p 14, line 313 - 314* → *p 17, line 390 - 393*: These data revealed the functional involvement of Col-IV deposition on VBM in angiogenic morphogenesis, and also suggest that mainly EC-derived Col-IV was involved in the formation and function of VBM at least around the tip portion where the *de novo* vascular lumen develops.
4. *p 14, line 318 - 321* → *p 17, line 401 - 404*: These results collectively suggest that spatiotemporally-enhanced deposition of EC-derived Col-IV on VBM in the presence of pericytes controls perivascular stiffness appropriately, which promotes branch elongation by physically restricting both lumen and branch expansions.
5. *p 15, line 335 - 338* → *p 20, line 472 - 475*: We further demonstrated that pericytes mediate proper Col-IV deposition by ECs onto the VBM, which accounts for spatiotemporal perivascular stiffening and the resultant reduction in vascular distensibility.
6. *p 16, line 374 - 375* → *p 22, line 520 - 523*: We further revealed that VBM stiffening via enhanced deposition of Col-IV secreted dominantly from ECs causes branch thinning in the presence of pericytes, which is consistent with a previous result showing the importance of Col-IV VBM³⁴.

Comment 7:

Does EC number influence the vessel elongation rate and/or lumen diameter? Is there a correlation between EC number and vessel diameter?

Response to Comment 7:

Thank you for pointing out “loose ends” that needed to be resolved in the near future. In the revised manuscript, we first addressed reviewer #1’s latter question by examining whether there is a correlation between branch diameter and the number of ECs that constitute it in the on-chip angiogenesis assay. For this purpose, we estimated the number of ECs by counting the nuclei contained in the portion of the branch from the center of the tip EC nucleus to 25 μm proximal (Extra Fig. 3a, See below), which corresponds to the region of the proximal half that was analyzed for lumen area shown in Fig. 1e of the revised manuscript. As expected, we found that there was a positive correlation between the number of constituent ECs and the diameter of the branch (Extra Fig. 3b, See below). However, with only such correlational data, it is not possible to address potential causal relationships, as would be necessary to answer reviewer #1’s former question.

Extra Fig. 3

Extra Fig. 3: Relationship between size of branch diameter and number of ECs composing it
a, Setting of analyzed branch area, shown in DIC images of an angiogenic branch, in which an example of a tip EC nucleus (red arrow) composing the analyzed branch (*left panel* of **a**) and the analyzed area were superimposed (*right panel* of **a**). The number of ECs was counted within a range 25 μm proximal to the center of tip EC nucleus (blue area), and branch diameter at the center of tip EC was measured. White arrow in the *left panel* indicates the direction of branch elongation. **b**, Distribution plot showing relationship between the size of branch diameter and the number of constituent EC in the analyzed area.

As pointed out, we consider changes in EC number comprising the 3D vascular structure, which is in turn referred to as EC density in a portion of the vascular structure, can affect both EC movement speed, leading to branch elongation, and vascular diameter growth, with these two factors being dependent on each other. This is possibly due to a

homeostatic process that maintains a constant EC density. Thus, we hypothesize that, in some cases, an increase in EC density may more dominantly promote EC forward movement and branch elongation, while in other cases, there is an increase in branch diameter or in both simultaneously. Indeed, our previous data, showing 2D trajectories of EC movements in elongating branches in an *in vitro* angiogenesis assay (Extra Fig. 4 corresponding to Fig. 2B-c in *Development* 138, 4763-4776, 2011, See below), is considered to support our hypothesis that an increase in EC density may promote branch elongation, which occurs as a consequence by maintaining EC density in a certain part of the EC branch. In addition, our previous data (*Nat Commun* 13, 2594, 2022) and the present data show intraluminal pressure to be an important factor influencing branch elongation as well as both lumen and branch diameters, independently of changes in EC number. At the moment, however, the causal relationships among dynamics of EC number change, branch elongation and branch and lumen diameter growth, as demonstrated by 3D time-lapse fluorescence imaging of on-chip angiogenesis at higher spatiotemporal resolution, have yet to be elucidated.

Extra Fig. 4

[REDACTED]

Extra Fig. 4: Individual EC movement in elongating branch of an *in vitro* angiogenesis assay
The figure corresponding to the Fig. 2B-c of our previous manuscript (*Development* 138, 4763-4776, 2011). Trajectory analysis. A representative pattern of EC movements during branch elongation are shown. Individual ECs are presented in different colors. Branch elongation speed was accelerated after EC density around the tip of branch increased (red arrow).

Comment 8:

Fig. 5C: I do not understand how the authors quantified the “rate” rate of Col-IV

coverage. The method that is used, b/a (where b is the length of Col-IV measured and a is the total length of the vessel measured) gives a ratio, not rate. The word “rate” gives the impression that an element of time has been measured but this is not the case.

Response to Comment 8:

Thank you for this insightful suggestion. As reviewer #1 pointed out, we fully agree that it is better to use the word “ratio” rather than “rate” in the phrase “rate of Col-IV coverage”, as follows:

Thus, we changed “rate” to “ratio” throughout the revised version of our work.

1. Fig. 6e and g, and 7c and l, and Extended data Fig. 12b, d and corresponding legends
2. p 13, line 286 → p 16, line 359: the Col-IV coverage **ratio**
3. p 13, line 288 → p 16, line 361: The Col-IV coverage **ratio**
4. p 37, line 863 → p 49, line 1141: coverage **ratio**
5. p 37, line 866 → p 49, line 1144: the coverage **ratio**
6. p 38, line 887 → p 50, line 1167: **ratio**,
7. p 40, line 923 → p 52, line 1218: the Col-IV coverage **ratios**

Comment 9:

Can the authors explain why a probability plot is used, instead of a frequency plot, in Figures 1g, 1j, 4j and 5j.

Response to Comment 9:

In this regard, we agree with the reviewer’s suggestion that a “frequency” plot is more appropriate for presenting the relevant data.

Therefore, we changed the expression “probability” to “frequency” throughout the revised version of our work, as follows:

1. Fig. 2i, 4e and j, and 5b, and corresponding legends
2. p 35, line 811 → p 50, line 1071: **frequency** of
3. p 40, line 923 → p 53, line 1230: **frequency**

Comment 10:

The title of the Fig. 2 is “Vascular lumen expansion inhibits directional tip EC movement and branch elongation during angiogenesis”. It is undisputed that there is a correlation between these two cellular processes, but there is no solid experimental manipulation to prove this. Given that it is possible to control intraluminal pressure, can the authors perform cycles of pressure changes e.g. increase-decrease-increase, and observe lumen expansion and branch elongation? This experimental design will provide more support to the authors’ statement.

Response to Comment 10:

We appreciate the reviewer for this highly meaningful suggestion to improve the quality of the data. In order to elucidate the causal relationship between vascular lumen expansion and directional tip EC movement, we have now examined the changes in directional tip EC movement by intentionally controlling lumen and branch expansions. In the experiment, according to the reviewer’s suggestion, first, we ectopically applied additional intraluminal pressure to the vascular wall, inducing lumen and branch expansion, by increasing hydrostatic pressure, and then relieved it to induce the shrinkage of both (new data, Fig. 3a-c, See below). When possible, the load-release cycle was performed twice. We found that in most cases, in rapid response to lumen and branch expansions, forward tip EC movement and branch elongation slowed down, but then were restored rather shortly afterward (new data, Fig. 3d-g, See below and supplemental Video 8). It is noteworthy that tip ECs, which had stopped moving, gradually resumed movement after the expanded lumen and branch shrank, thereby again contributing to branch elongation (new data, Fig. 3d, e, h, i, See below). These dynamic changes in tip EC movement and branch elongation in response to externally applied intraluminal pressure and its release were similarly observed during the 2nd load-release cycle (new data, Extended Data Fig. 7, See below and supplemental Video 10). In clear contrast, in the absence of intraluminal pressure, movement of some tip ECs decelerated regardless of the presence or absence of intraluminal pressure loading, whereas most tip ECs continued to move forward over the observational period (new data, Extended Data Fig. 6, See below and supplemental Video 9). These results suggest that, at a minimum,

ectopic lumen and branch expansions inhibit forward tip EC movement and branch elongation. These results are presented in the revised version of our manuscript.

Fig. 3

Hanada et al. Figure 3

Fig. 3: Influences of interventions for lumen and branch expansion altering tip EC movement and branch elongation

a-i, Physical interventions altering lumen and branch expansions by externally loading intraluminal

pressure in on-chip angiogenesis of ECs. A schematic diagram showing the timeline of the loading and release cycles of intraluminal pressure (a). Additional pressure (41 ± 14 Pa) was loaded into intraluminal spaces of elongating branches (Day 3) for 3 hours and then released to default status. In some cases, the loading and release cycle was repeated 20 hours after the 1st pressure release. Representative DIC images showing changes of lumens and branches when loaded (*left panel*) followed by release of the (*right panel*) additional intraluminal pressure (b), and individual plots of changes in luminal area in response to the loading and release of intraluminal pressure, respectively (loading; n=12 from 3 devices, release; n=11 from 3 devices examined during the course of 3 independent experiments). White dotted lines indicate the proximal and distal boundaries of the analyzed luminal region. Time-lapse imaging of on-chip angiogenesis of ECs during the intervention (d). Representative DIC images at the elapsed time indicated at the top showing the dynamics in branch elongation, lumen development and tip EC movement before and after loading and the release of additional intraluminal pressure (*upper panel* of d, See also Supplementary Video 8) and kymograph of branch elongation (*lower panel* of d). In the DIC images, yellow arrows and red lines indicate the trajectory of the tip of the branch and the tip EC displacement, respectively. In the kymograph, red and blue dotted lines indicate time points at which additional intraluminal pressure was loaded and then released, respectively. Box plots of averaged branch elongation 3 hours before (pressure load (-)) and after (pressure load (+)) the 1st pressure loading and 3 hours after its release (pressure release (+)), merged with individual plots showing the branch elongation changes in response to loading and the release of intraluminal pressure (*left panel*), and the corresponding box and individual plots in the control group without interventions (*right panel*) (pressure loading and release: n=21 from 2 devices, control: n=19 from 2 devices examined during the course of 2 independent experiments, See also Extended Fig. 6) (e). Kymograph showing the dynamics of tip EC movement for 50 minutes before and 100 minutes after loading additional intraluminal pressure (time=0, dotted red line) (f), and box plots of averaged tip EC movement speed for 50 minutes before, 50 minutes and 50-100 minutes after the pressure load, calculated from the dynamics data in f (g, each group: n=13 from 3 devices examined during the course of 3 independent experiments). Kymograph showing the dynamics in tip EC movement for 50 minutes before and 100 minutes after the release of additional intraluminal pressure (time=0, dotted red line) (h), and box plots of averaged tip EC movement speed for 50 minutes before, 50 minutes and 50-100 minutes after pressure release, calculated from the dynamics data in h (i, each group: n=8 from 2 devices examined during the course of 2 independent experiments). Red double arrows indicate the loading phase of the intraluminal pressure in a, f, and h, and the *bottom panel* of d. j-m, Pharmacological interventions altering lumen and branch expansions. Representative fluorescent images of the branch of on-chip angiogenesis of ECs, visualized by UEA-1 Lectin, without and with treatment by a ROCK inhibitor, Y27632 (*top panel* of j), and a myosin II inhibitor, Blebbistatin (*bottom panel* of j), and without and with TG treatment of the gel. k, Localization pattern of pMLC2 in on-chip angiogenic branches of ECs (UEA-1 Lectin) around the area where the *de novo* vascular lumen (Moesin, yellow asterisks) develops. Representative confocal z-projection images, with magnified images of square areas bound by dots (z-projection) and their corresponding confocal x-y slice images at specific z-positions (slice) without and with Y27632 and without and with TG treatment. Box plots of branch length (*left panel*) and Tip expandability (*right panel*, See also Fig. 2g), in on-chip angiogenesis without and with Y27632 (l) or Blebbistatin (m) and without and with TG treatment of the gel (l, Y27632 (-) TG (-): n=117 from 3 devices; Y27632 (+) TG (-): n=108 from 3 devices; Y27632 (+) TG (+): n=102 from 3 devices, examined during the course of 3 independent experiments, m, Blebbistatin (-) TG (-): n=39 from 2 devices; Blebbistatin (+) TG (-): n=35 from 2 devices; Blebbistatin (+) TG (+): n=64 from 3 devices, examined during the course of 2 independent experiments). All box plots show the interquartile range, with the middle line defining the median, and whiskers show the minimum and maximum values, excluding outliers, for g, i, l and m. Outlier values are defined as being 1.5 times the interquartile range above and below the third and first quartile, respectively. Scale bars: 20 μ m for *top panel* of d, k, 25 μ m for b and 100 μ m for j. Wilcoxon signed rank test for e, Wilcoxon signed rank test with a Bonferroni correction for e, g and i and Mann-Whitney U test with a Bonferroni correction for l and m.

Extended Data Fig. 6

Hanada et al. Extended Data Figure 6

Extended Data Fig. 6: Time-lapse imaging of on-chip angiogenesis of ECs as control with no interventions

Physical intervention control experiments for lumen and branch expansions by externally loading intraluminal pressure in on-chip angiogenesis of ECs. **a**, Representative DIC images at the elapsed time indicated at the top showing the dynamics in branch elongation, lumen development and tip EC movement in the same timeline, before and after loading and then releasing the additional intraluminal pressure in the intervention group (Fig. 3d) (*top panel*, See also Supplementary Video 9) and kymograph of branch elongation (*bottom panel*). In the DIC images, red lines and yellow arrows indicate the trajectory of tip EC displacement and the tip of the branch, respectively. In the kymograph, red and blue dotted lines indicate time points at which additional intraluminal pressure was loaded and released, respectively, in the intervention group. Scale Bars: 20 μm for top panel of **a**. **b-e**, Quantification of tip EC movement. Kymograph showing the dynamics in tip EC movement in the same timeline, 50 minutes before and 100 minutes after loading additional intraluminal pressure in the intervention group (time=0, dotted red line) (**b**), and box plots of averaged tip EC movement speeds in the same timeline, 50 minutes before, 50 minutes and 50-100 minutes after the pressure load in the intervention group, calculated from the dynamics data in **b** (**c**, each group: n=12 from 2 devices examined during the course of 2 independent experiments). Kymograph showing the dynamics in tip EC movement in the same timeline, 50 minutes before and 100 minutes after the release of additional intraluminal pressure in the intervention group (time=0, dotted red line) (**d**), and box plots of averaged

tip EC movement speeds in the same timeline, 50 minutes before, 50 minutes and 50-100 minutes after pressure release in the intervention group, calculated from the dynamics data in **d** (**e**, each group: $n=13$ from 2 devices examined during the course of 2 independent experiments). All box plots show the interquartile range, with the middle line defining the median, and whiskers show the minimum and maximum values, excluding outliers, for **c** and **e**. Outlier values are defined as being 1.5 times the interquartile range above and below the third and first quartile, respectively. Wilcoxon signed rank test with a Bonferroni correction for **c**, **e**.

Extended Data Fig. 7

Extended Data Fig. 7: Time-lapse imaging of on-chip angiogenesis of ECs at the 2nd intervention cycle and the corresponding analyses

The second cycle of physical intervention for lumen and branch expansions by externally loading intraluminal pressure in on-chip angiogenesis of ECs. **a**, Representative DIC images at the elapsed time indicated at the top showing the dynamics in branch elongation, lumen development and tip EC movement before and after loading and then releasing additional intraluminal pressure (*top panel*, See also Supplementary Video 10) and kymograph of branch elongation (*bottom panel*). In the DIC images, red lines and yellow arrows indicate the trajectory of tip EC displacement and the tip of the branch, respectively. In the kymograph, red and blue dotted lines indicate time points at which additional intraluminal pressure was loaded and released, respectively. Scale Bars: 20 μm for top panel of **a**. **b-d**, Quantification of branch elongation and tip EC movement. Box plots of averaged branch elongation 3 hours before (pressure load (-)) and after (pressure load (+)) the 2nd pressure loading and 3 hours

after the release (pressure release (+)), merged with individual plots showing the branch elongation changes in response to loading and then the release of intraluminal pressure (pressure load and release: n=11 from 2 devices, control: n=6 from 1 devices examined during the course of 1-2 independent experiments) **(b)**. All box plots **(b)** show the interquartile range, with the middle line defining the median, and whiskers show the minimum and maximum values, excluding outliers. Outlier values are defined as being 1.5 times the interquartile range above and below the third and first quartile, respectively. Kymograph showing the dynamics in tip EC movement 50 minutes before and 100 minutes after loading additional intraluminal pressure (time=0, dotted red line) **(c)**. Kymograph showing the dynamics in tip EC movement 50 minutes before and 100 minutes after the release of additional intraluminal pressure (time=0, dotted red line) **(d)**. Wilcoxon signed rank test with a Bonferroni correction for **b**.

We attempted to further control lumen and branch expansions in two different ways. First, we restricted lumen and branch expansions via stiffening of the perivascular environment. In the previous version of our manuscript, we had already described stiffening of the surrounding ECM in response to TG treatment as inducing an increase in elongation with narrowing of the branch width in on-chip angiogenesis without pericytes (previous Fig. 4g-k). Therefore, in the revised manuscript, using the time-lapse imaging data, we additionally showed changes in the dynamics of tip EC movement and lumen expansion when the ECM was stiffened by TG treatment, and examined the relationship between lumen and branch expansions and branch elongation. We thereby found that ECM stiffening potently reduced acceleration of the forward tip EC movement in conjunction with suppression of lumen expansion (new data, Fig. 2h-l, Extended Data Fig. 4b and 5, See below), collectively supporting the hypothesis that lumen expansion causes branch elongation.

Fig. 2h-I

Hanada et al. Figure 2

Fig. 2h-i: Changes in the dynamics of tip EC movement and lumen expansion and their relationship

Effects of ECM stiffening by TG treatment on on-chip angiogenesis (Day 4). Quantification of tip EC movement and lumen development using the gel without and with TG treatment (**h-I**). Kymograph of tip EC movements (*left panel of h*, TG (-): n=11 from 3 devices; TG (+): n=10 from 3 devices examined during the course of 3 independent experiments) and typical “*move forward and stop*” pattern of tip EC movement (*right panel of h*), as in the *left panel of h* (blue arrow). Frequency distribution of EC movement speeds using the gel without or with TG treatment (**i**), calculated from the dynamics data in **h** for each group. Kymograph of representative tip EC movement (red) and lumen area change (distal (D) and proximal (P) areas in black and blue, respectively) using the gel with TG treatment (*bottom panel of j*). Serial DIC images of elongating angiogenic branches, with trajectory (red line) of the center of the EC nucleus and lumen diameter (yellow bars) at timepoints indicated by numbers in the kymograph (orange dotted line) (*top panel of j*, See also Supplementary Video 7). White arrows indicate displacements of the nuclei of tip ECs. Kymographs showing relationships in dynamics between tip EC movement (*top of k*) and the nearby lumen area change (*bottom of k*, distal (D) and proximal (P) areas in black and blue, respectively) using the gel with TG treatment. Data for 100 minutes are shown and the half-way timepoint is set as 0 minutes (orange dotted line) (n=10 from 3 devices examined during the course of 3 independent experiments). As to lumen area changes, data are shown for predominantly expanded proximal (P) or distal (D) lumens. Box and individual plots of lumen area changes for 30 minutes prior

to the half-way timepoint using the gel without and with TG treatment (values with (red) and without (blue) deceleration of tip EC movement), calculated from the dynamics data in k (I). Data are expressed as means \pm SD for *right panel* of d. All box plots show the interquartile range, with the middle line defining the median, and whiskers show the minimum and maximum values, excluding outliers, for c, f, I and *right panel* of g. Outlier values are defined as being 1.5 times the interquartile range above and below the third and first quartile, respectively. Scale bars: 20 μm for d, 25 μm for j and 100 μm for e. Mann-Whitney U test with a Holm correction for c, f, I, *right panels* of d, g and Kolmogorov-Smirnov test for i.

Extended Data Fig. 4b

Extended Data Fig. 4b: Tip EC movement patterns in elongating branches of on-chip angiogenesis using gels with TG treatment

Kymographs of individual tip EC movements in elongating angiogenic branches of on-chip angiogenesis by ECs using gel with TG treatment (examined during the course of 3 independent experiments), which correspond to the data in Fig. 2h. A “move forward and stop” pattern of tip EC movement was rarely seen in elongating angiogenic branches of on-chip angiogenesis using gels with TG treatment.

Extended Data Fig. 5

Hanada et al. Extended Data Figure 5

Extended Data Fig. 5: Temporal relationship between tip EC movement and lumen expansion in elongating branches of on-chip angiogenesis using gels with TG treatment

Kymographs showing relationships in dynamics between tip EC movement (*top*) and lumen area (*bottom*) for 50 minutes before and after the deceleration timepoint (0 minutes, orange dotted line) in the presence (*left panel*) and the absence (*right panel*) of deceleration of forward tip EC movements (deceleration: $n=3$ from 3 devices; no deceleration: $n=7$ from 3 devices, examined during the course of 3 independent experiments). For changes of lumen area, data are shown for the predominantly expanded proximal (P) or distal (D) lumen. Dynamics data indicated within squares bounded by black dots were used for analysis of changes in the lumen area for 30 minutes prior to the deceleration timepoint shown in Fig. 1h.

Next, we attempted to induce lumen and branch expansion by pharmacological interventions in addition to the aforementioned physical intervention (new data, Fig. 3j-m, See above). Previously reported studies (*Nat Cell Biol* 18: 443–450, 2016; *Nat Cell Biol* 18:311–318, 2016; *Circ Res* 119:810–826, 2016; *Nat Commun* 11:5476, 2020) showed that in both epithelial and endothelial tubulogenesis, expansion of the *de novo* lumen is limited by apical actomyosin contractility via Ras homologue gene family member A (RhoA) - Rho-associated protein kinase (ROCK) – non-muscle myosin II (NMII) signaling. Therefore, we blocked the RhoA-ROCK-NMII signaling pathway using an inhibitor of ROCK, Y27632, and an inhibitor of NMII, Blebbistatin, in on-chip angiogenesis of ECs to induce circumferential lumen and branch expansions by inhibiting apical actomyosin contractility. As expected, both inhibitor treatments induced branch and lumen expansions around the tip, and concomitantly reduced branch elongation (new

data, Fig. 3j-m, Extended Data Fig. 8, See above and below). Furthermore, the reduction in branch elongation was rescued by physically restricting branch and lumen expansions via stiffening of the surrounding ECM by TG treatment (Fig. 3j-m, see above), quite possibly suggesting branch and lumen expansions to physically inhibit branch elongation. Taken together, these data obtained employing three different interventions suggest that there is a causal link between lumen development and branch elongation, in which lumen and branch expansions inhibit forward directional movement of tip ECs, resulting in branch elongation.

Hanada et al. Extended Data Figure 8

Extended Data Fig. 8: Phosphorylated myosin light chain 2 localization in the branch around forming *de novo* lumen in on-chip angiogenesis without and with ROCK inhibitor and without

and with TG treatment

Representative confocal z-projection images in on-chip angiogenesis (Day 4), magnified images of the areas enclosed within squares bounded by white dots in the left images (z-projection), and their corresponding confocal x-y slice images at specific z-positions (slice) of the non-treated group (a), ROCK inhibitor-treated group (b) and both ROCK and TG-treated groups (c). All images were visualized by whole-mount staining with phosphorylated myosin light chain 2 (pMLC2), Moesin and DAPI (nuclei). Scale bars: 10 μm for confocal z-projection images in the right and x-y slice images and 20 μm for confocal z-projection images in the left.

Therefore, we added these new data throughout the revised version of our manuscript, as follows:

1. Added Fig. 2h-l and 3, and corresponding legends
2. Extended Data Fig. 4b and 5-8, and corresponding legends
3. Added *p 8, line 168 - 171*: To verify the possible causal relationship between branch elongation and lumen development, we investigated changes in the dynamics of directional EC movement and branch elongation when the degrees of lumen and branch expansions were externally altered in different ways, in the on-chip angiogenesis of ECs.
4. *p 11, line 247 - 250* \rightarrow *p 8, line 176 - 179*: Thus, to intentionally restrict branch and lumen expansion in the on-chip angiogenesis assay, we first increased the stiffness of ECM by treatment with Transglutaminase (TG)^{25,26}, a crosslinking enzyme of fibrin gels, without changing the biochemical compositions of the on-chip assay of ECs.
5. *p 11, line 255 - 260* \rightarrow *p 9, line 207 - p 10, line 218*: This “move forward and stop” movement pattern was repeated during branch elongation in on-chip angiogenesis of ECs using the non-treated ECM (Fig. 2h and Extended Data Fig. 4). In contrast, when the ECM was treated with TG, most of the tip ECs tended to continue moving forward, as characterized by a higher frequency of forward movement with greater speed, such that the aforementioned movement pattern was suppressed (Fig. 2h, i and Extended Data Fig. 4, See also Supplementary Video 7). However, the sizes of nearby lumens remained essentially constant, though occasional decreases were seen, regardless of whether the lumen was distal or proximal to the nuclei, and expansion was rarely noted (Fig. 2j-l and Extended Data Fig. 5), in clear contrast to the condition without TG treatment (Fig. 2i). These results indicate that restriction of excessive lumen expansion promotes branch elongation with a higher frequency of forward directional movement of tip ECs.

6. Added *p 10, line 219 - p 11, line 242*: Next, we ectopically altered the expansion states of both lumen and branch by increasing intraluminal pressure externally and then restoring it. To achieve this, additional hydrostatic pressure ($\Delta P=41\pm 14$ Pa) with culture media was loaded into the intraluminal spaces of elongating branches and it was then released 3 hours after the loading in on-chip angiogenesis of ECs.... Similar dynamic changes in directional tip EC movement and branch elongation in response to external loading and release of intraluminal pressure could be repeatedly reproduced in the second round of the pressure load-release cycle, although the number of branches in which this could be confirmed was methodologically limited (Extended Data Fig. 7, See also Supplementary Video 10). These results clearly indicate that ectopic lumen and branch expansions can inhibit branch elongation driven by forward directional EC movement.
7. Added *p 11, line 243 - p 11, line 261*: Finally, we attempted to induce lumen and branch expansions by pharmacological interventions. It was previously reported that, in both epithelial and endothelial tubulogenesis, expansion of the *de novo* lumen is limited by apical actomyosin contractility via Ras homologue gene family member A (RhoA) - Rho-associated protein kinase (ROCK) – non-muscle myosin II (NMII) signaling^{24,28-30}. Collectively, these findings strongly suggest a causal link between lumen development and branch elongation, in which lumen and branch expansions inhibit forward directional movement of the tip ECs, resulting in retardation of branch elongation.
8. *p 14, line 329 - p 15, 333 → p 20, line 465 - 470*: An observational approach combining *in vitro* time-lapse imaging of the on-chip angiogenesis assay with *in vivo* validation, together with physical and pharmacological interventions specifically intended to control lumen and branch expansions, showed that lumen emergence followed by its expansion inhibits directional movement of the tip ECs polarizing along the antero-posterior axis, resulting in delayed branch elongation.
9. Added *p 38, line 883 - 888*: In the experiments to examine the effects of an inhibitor of ROCK, Y27632 (FUJIFILM Wako Pure Chemical Corp.), and an inhibitor of NMII, Blebbistatin (Merck), on on-chip angiogenesis, EGM-2 were replaced with new culture media including Y27632 (10 nM) or Blebbistatin (10 μ M) or DMSO (Wako),

twice at a 30-minute interval one day after introducing HUVECs. Two days later, the devices were subjected to whole-mount staining.

10. Added *p 42, line 986 - p 43, line 1001*: To assess the influences of lumen and branch expansions on the dynamics of tip EC movement and branch elongation, after acquiring at least 12 hours of time-lapse imaging data, similarly, approximately 40 Pa of hydrostatic pressure was loaded onto the intraluminal space of angiogenic branches to intentionally induce lumen and branch expansions, and time-lapse imaging was then restarted (Fig. 3a). ...Furthermore, after completion of the time-lapse imaging study, the actual values of the applied hydrostatic pressure were measured using the pressure gauge through a similar hydrostatic pressure loading procedure. The average values were 41 ± 14 Pa (n=6).
11. Added *p 46, line 1072 - 1078*: In the intervention study assessing vascular intraluminal pressure, in addition to tip ECs, the tips of the angiogenic branches were simultaneously tracked in a similar manner on DIC images obtained every 15 minutes. From the tracking data, we determined the average speed (/minute) of individual tip ECs during specified periods after loading and the release of intraluminal pressure, as well as the average branch elongation for 3 hours before and after intraluminal pressure loading and 3 hours after its release.

Comment 11:

I would preferably like to see better resolution of images showing retina vessels. Is this possible?

Response to Comment 11 :

According to the reviewer's suggestion, in the revised manuscript, we replaced the images of retinal angiogenic sprouts in Fig. 1d, f of the previous manuscript with better resolution images. These are presented as Fig. 4b, d in the revised manuscript as below.

Fig. 4b, d

Finally, we again deeply appreciate the reviewer's supportive and insightful input and comments on our work. According to the reviewer's helpful suggestions, by performing additional experiments and extensively revising the manuscript based on the new data obtained, we believe we had made the manuscript significantly improved although a lot still remain to be figured out.

Reviewer #2 (Remarks to the Author):

This paper investigates an important and timely topic regarding how the physical microenvironment regulates vascular morphogenesis. The authors utilize a multidisciplinary approach to investigate this topic by combining in vitro microfluidic chip studies with murine retinal angiogenesis in vivo experiments. In this paper, the authors focus specifically on factors regulating lumen diameter and branch elongation during morphogenesis. The authors show that pericytes deposit collagen-IV around ECs, increasing the local matrix stiffness. This increase in matrix stiffness limits lumen expansion and alters EC branch elongation. These results help to address a current gap in the field regarding the role of pericytes and their mechanical regulation of vascular processes.

First of all, we thank reviewer #2 for the supportive input and insightful comments on our work. We agree that addressing the concerns raised would significantly improve the manuscript. Therefore, we performed additional experiments according to reviewer #2's helpful suggestions, and extensively revised the manuscript based on the new data obtained. The added findings are described in detail below.

This paper demonstrated a few significant drawbacks, mainly in Figures 1-3. The experimental design in the first few figures does not address the hypothesis the authors are seeking to answer in the text (relationship between lumen expansion and elongation) but answers a different hypothesis not discussed, which is investigating the role of pericytes on these metrics. The beginning figures and interpretations attempt to convince the reader that there is a relationship between lumen expansion and EC branch elongation, but the data is really showing that pericytes regulate both metrics. Furthermore, the authors consistently draw conclusions of causation when only correlative data is shown. The role of pericytes is neglected in the first half of the paper and not discussed by the authors until Figures 4 and 5. The paper then jumps from neglecting the role of pericytes in the data to making pericytes the main character of the story. Once this is realized by the authors, the more compelling data and interpretations lay in Figure 4 and 5. Figures 4 and 5 showed compelling data with improved

experimental design and hypotheses.

Response:

We appreciate reviewer #2's helpful comments. In the previous manuscript, as noted by reviewer #2, appropriate data was not presented, interpreted and logically developed, such that we were unable to clearly state the points needing clarification, mainly regarding the following two points: first, whether the lumen expansion seen during lumen development inhibits forward tip EC movement and branch elongation; and, second, whether inhibition of branch elongation is biomechanically counter-regulated by perivascular pericytes via perivascular stiffness, which may explain one of the mechanisms underlying pericyte-dependent promotion of angiogenesis. In the revised manuscript, we feel that the logical flow has been greatly improved and that the additional data have answered questions raised by the reviewer.

To more clearly articulate our claim in the revised manuscript, based on reviewer #2's suggestions, we extensively rearranged the structure of the text, modifying the order of data presentation in terms of the logical flow. In the former part, we focused on clarifying whether lumen expansion seen during lumen development inhibits forward tip EC movement and branch elongation in on-chip angiogenesis of ECs (in the absence of pericytes). First, the data showing the differences in angiogenic morphology depending on the presence versus the absence of pericytes (Fig. 1c-j in the previous manuscript), which had led to an unclear progression of logic, were moved to the latter part, in which the roles of pericytes were addressed. Next, from the time-lapse imaging data of on-chip angiogenesis of ECs, we demonstrated that there is a significant inverse correlation between lumen and branch expansions and forward tip EC movement in Fig. 1 (previous Fig. 2). Then, we attempted to clarify whether there is a causal relationship between these two parameters based on the following three intervention studies, including two newly added experiments, designed to intentionally control the lumen and branch expansions, as shown in Fig. 2 and 3 (See also response to major comment 3). One of the three approaches was to physically limit lumen and branch expansion by treating the gel with TG, as shown in Fig. 4 of the previous manuscript, and, by additionally conducting analyses of their dynamics based on time-lapse imaging data (new data, Fig. 2h-l, Extended Data Fig. 4b, 5, See below). Employing this strategy we showed that restriction

of lumen and branch expansions promotes forward tip EC movement and branch elongation (Fig. 2). Next, by loading and releasing additional hydrostatic pressure, we intentionally and physically induced lumen and branch expansions and then their relaxation. Thereby, we demonstrated that forward tip EC movement and branch elongation ceased and then resumed in rapid response to expansion and shrinkage of both lumen and branch (new data, Fig. 3a-i, Extended Data Fig. 6, 7, See below). Finally, we pharmacologically induced lumen and branch expansions by using inhibitors for Rho-associated protein kinase (ROCK)-non-muscle myosin II (NMII) signaling, which is involved in controlling lumen expansion via apical actomyosin contractility. We thereby showed that signaling inhibition delayed branch elongation accompanied by lumen and branch expansions, which was further restored by physical restriction of lumen and branch expansions in response to TG treatment of the gel (new data, Fig. 3j-m, Extended Data Fig. 8, See below). These three physical and pharmacological intervention studies strongly suggest a causal link between lumen development and branch elongation, in which lumen and branch expansions inhibit forward tip EC movement, resulting in retardation of branch elongation. Therefore, in the revised manuscript, we addressed concerns raised by review #2 regarding the potential causality between lumen development and branch elongation. In addition, we believe that we were able to further demonstrate that mechanical stimulation of lumen and branch expansions inhibits branch elongation.

Fig. 2h-I

Hanada et al. Figure 2

Fig. 2h-i: Changes in the dynamics of tip EC movement and lumen expansion and their relationship

Effects of ECM stiffening by TG treatment on on-chip angiogenesis (Day 4). Quantification of tip EC movement and lumen development using the gel without and with TG treatment (**h-l**). Kymograph of tip EC movements (*left panel of h*, TG (-): n=11 from 3 devices; TG (+): n=10 from 3 devices examined during the course of 3 independent experiments) and typical “move forward and stop” pattern of tip EC movement (*right panel of h*), as in the *left panel of h* (blue arrow). Frequency distribution of EC movement speeds using the gel without or with TG treatment (**i**), calculated from the dynamics data in **h** for each group. Kymograph of representative tip EC movement (red) and lumen area change (distal (D) and proximal (P) areas in black and blue, respectively) using the gel with TG treatment (*bottom panel of j*). Serial DIC images of elongating angiogenic branches, with trajectory (red line) of the center of the EC nucleus and lumen diameter (yellow bars) at timepoints indicated by numbers in the kymograph (orange dotted line) (*top panel of j*, See also Supplementary Video 7). White arrows indicate displacements of the nuclei of tip ECs. Kymographs showing relationships in dynamics between tip EC movement (*top of k*) and the nearby lumen area change (*bottom of k*, distal (D) and proximal (P) areas in black and blue, respectively) using the gel with TG treatment. Data for 100 minutes are shown and the half-way timepoint is set as 0 minutes (orange dotted line) (n=10 from 3 devices examined during the course of 3 independent experiments). As to lumen area changes, data are shown for predominantly expanded proximal (P) or distal (D) lumens. Box and individual plots of lumen area changes for 30 minutes prior to the half-way timepoint using the gel without and with TG treatment (values with (red) and without (blue) deceleration of tip EC movement), calculated from the dynamics data in **k** (**l**). Data are expressed as means \pm SD for *right panel of d*. All box plots show the interquartile range, with the middle line defining the median, and whiskers show the minimum and maximum values, excluding outliers, for **c**, **f**, **l** and *right panel of g*. Outlier values are defined as being

1.5 times the interquartile range above and below the third and first quartile, respectively. Scale bars: 20 μm for **d**, 25 μm for **j** and 100 μm for **e**. Mann-Whitney U test with a Holm correction for **c**, Mann-Whitney U test for **f**, **l**, right panels of **d**, **g** and Kolmogorov-Smirnov test for **i**.

Extended Data Fig. 4b

Extended Data Fig. 4b: Tip EC movement patterns in elongating branches of on-chip angiogenesis using gels with TG treatment

Kymographs of individual tip EC movements in elongating angiogenic branches of on-chip angiogenesis by ECs using gel with TG treatment (examined during the course of 3 independent experiments), which correspond to the data in Fig. 2h. A “move forward and stop” pattern of tip EC movement was rarely seen in elongating angiogenic branches of on-chip angiogenesis using gels with TG treatment.

Extended Data Fig. 5

Hanada et al. Extended Data Figure 5

Extended Data Fig. 5: Temporal relationship between tip EC movement and lumen expansion in elongating branches of on-chip angiogenesis using gels with TG treatment

Kymographs showing relationships in dynamics between tip EC movement (*top*) and lumen area (*bottom*) for 50 minutes before and after the deceleration timepoint (0 minutes, orange dotted line) in

the presence (*left panel*) and the absence (*right panel*) of deceleration of forward tip EC movements (deceleration: n=3 from 3 devices; no deceleration: n=7 from 3 devices, examined during the course of 3 independent experiments). For changes of lumen area, data are shown for the predominantly expanded proximal (P) or distal (D) lumen. Dynamics data indicated within squares bounded by black dots were used for analysis of changes in the lumen area for 30 minutes prior to the deceleration timepoint shown in Fig. 1h.

Fig. 3

Hanada et al. Figure 3

Fig. 3: Influences of interventions for lumen and branch expansion altering tip EC movement and branch elongation

a-i, Physical interventions altering lumen and branch expansions by externally loading intraluminal pressure in on-chip angiogenesis of ECs. A schematic diagram showing the timeline of the loading

and release cycles of intraluminal pressure (a). Additional pressure (41 ± 14 Pa) was loaded into intraluminal spaces of elongating branches (Day 3) for 3 hours and then released to default status. In some cases, the loading and release cycle was repeated 20 hours after the 1st pressure release. Representative DIC images showing changes of lumens and branches when loaded (*left panel*) followed by release of the (*right panel*) additional intraluminal pressure (b), and individual plots of changes in luminal area in response to the loading and release of intraluminal pressure, respectively (loading; n=12 from 3 devices, release; n=11 from 3 devices examined during the course of 3 independent experiments). White dotted lines indicate the proximal and distal boundaries of the analyzed luminal region. Time-lapse imaging of on-chip angiogenesis of ECs during the intervention (d). Representative DIC images at the elapsed time indicated at the top showing the dynamics in branch elongation, lumen development and tip EC movement before and after loading and the release of additional intraluminal pressure (*upper panel* of d, See also Supplementary Video 8) and kymograph of branch elongation (*lower panel* of d). In the DIC images, yellow arrows and red lines indicate the trajectory of the tip of the branch and the tip EC displacement, respectively. In the kymograph, red and blue dotted lines indicate time points at which additional intraluminal pressure was loaded and then released, respectively. Box plots of averaged branch elongation 3 hours before (pressure load (-)) and after (pressure load (+)) the 1st pressure loading and 3 hours after its release (pressure release (+)), merged with individual plots showing the branch elongation changes in response to loading and the release of intraluminal pressure (*left panel*), and the corresponding box and individual plots in the control group without interventions (*right panel*) (pressure loading and release: n=21 from 2 devices, control: n=19 from 2 devices examined during the course of 2 independent experiments, See also Extended Fig. 6) (e). Kymograph showing the dynamics of tip EC movement for 50 minutes before and 100 minutes after loading additional intraluminal pressure (time=0, dotted red line) (f), and box plots of averaged tip EC movement speed for 50 minutes before, 50 minutes and 50-100 minutes after the pressure load, calculated from the dynamics data in f (g, each group: n=13 from 3 devices examined during the course of 3 independent experiments). Kymograph showing the dynamics in tip EC movement for 50 minutes before and 100 minutes after the release of additional intraluminal pressure (time=0, dotted red line) (h), and box plots of averaged tip EC movement speed for 50 minutes before, 50 minutes and 50-100 minutes after pressure release, calculated from the dynamics data in h (i, each group: n=8 from 2 devices examined during the course of 2 independent experiments). Red double arrows indicate the loading phase of the intraluminal pressure in a, f, and h, and the *bottom panel* of d. j-m, Pharmacological interventions altering lumen and branch expansions. Representative fluorescent images of the branch of on-chip angiogenesis of ECs, visualized by UEA-1 Lectin, without and with treatment by a ROCK inhibitor, Y27632 (*top panel* of j), and a myosin II inhibitor, Blebbistatin (*bottom panel* of j), and without and with TG treatment of the gel. k, Localization pattern of pMLC2 in on-chip angiogenic branches of ECs (UEA-1 Lectin) around the area where the *de novo* vascular lumen (Moesin, yellow asterisks) develops. Representative confocal z-projection images, with magnified images of square areas bound by dots (z-projection) and their corresponding confocal x-y slice images at specific z-positions (slice) without and with Y27632 and without and with TG treatment. Box plots of branch length (*left panel*) and Tip expandability (*right panel*, See also Fig. 2g), in on-chip angiogenesis without and with Y27632 (l) or Blebbistatin (m) and without and with TG treatment of the gel (l, Y27632 (-) TG (-): n=117 from 3 devices; Y27632 (+) TG (-): n=108 from 3 devices; Y27632 (+) TG (+): n=102 from 3 devices, examined during the course of 3 independent experiments, m, Blebbistatin (-) TG (-): n=39 from 2 devices; Blebbistatin (+) TG (-): n=35 from 2 devices; Blebbistatin (+) TG (+): n=64 from 3 devices, examined during the course of 2 independent experiments). All box plots show the interquartile range, with the middle line defining the median, and whiskers show the minimum and maximum values, excluding outliers, for g, i, l and m. Outlier values are defined as being 1.5 times the interquartile range above and below the third and first quartile, respectively. Scale bars: 20 μ m for *top panel* of d, k, 25 μ m for b and 100 μ m for j. Wilcoxon signed rank test for e, Wilcoxon signed rank test with a Bonferroni correction for e, g and i and Mann-Whitney U test with a Bonferroni correction for l and m.

Extended Data Fig. 6

Hanada et al. Extended Data Figure 6

Extended Data Fig. 6: Time-lapse imaging of on-chip angiogenesis of ECs as control with no interventions

Physical intervention control experiments for lumen and branch expansions by externally loading intraluminal pressure in on-chip angiogenesis of ECs. **a**, Representative DIC images at the elapsed time indicated at the top showing the dynamics in branch elongation, lumen development and tip EC elongation, before and after loading and then releasing the additional intraluminal pressure in the intervention group (Fig. 3d) (*top panel*, See also Supplementary Video 9) and kymograph of branch elongation (*bottom panel*). In the DIC images, red lines and yellow arrows indicate the trajectory of tip EC displacement and the tip of the branch, respectively. In the kymograph, red and blue dotted lines indicate time points at which additional intraluminal pressure was loaded and released, respectively, in the intervention group. Scale Bars: 20 μm for top panel of **a**. **b-e**, Quantification of tip EC movement. Kymograph showing the dynamics in tip EC movement in the same timeline, 50 minutes before and 100 minutes after loading additional intraluminal pressure in the intervention group (time=0, dotted red line) (**b**), and box plots of averaged tip EC movement speeds in the same timeline, 50 minutes before, 50 minutes and 50-100 minutes after the pressure load in the intervention group, calculated from the dynamics data in **b** (**c**, each group: n=12 from 2 devices examined during the course of 2 independent experiments). Kymograph showing the dynamics in tip EC movement in the same timeline, 50 minutes before and 100 minutes after the release of additional intraluminal pressure in the intervention group (time=0, dotted red line) (**d**), and box plots of averaged

tip EC movement speeds in the same timeline, 50 minutes before, 50 minutes and 50-100 minutes after pressure release in the intervention group, calculated from the dynamics data in **d** (e, each group: n=13 from 2 devices examined during the course of 2 independent experiments). All box plots show the interquartile range, with the middle line defining the median, and whiskers show the minimum and maximum values, excluding outliers, for **c** and **e**. Outlier values are defined as being 1.5 times the interquartile range above and below the third and first quartile, respectively. Wilcoxon signed rank test with a Bonferroni correction for **c**, **e**.

Extended Data Fig. 7

Hanada et al. Extended Data Figure 7

Extended Data Fig. 7: Time-lapse imaging of on-chip angiogenesis of ECs at the 2nd intervention cycle and the corresponding analyses

The second cycle of physical intervention for lumen and branch expansions by externally loading intraluminal pressure in on-chip angiogenesis of ECs. **a**, Representative DIC images at the elapsed time indicated at the top showing the dynamics in branch elongation, lumen development and tip EC movement before and after loading and then releasing additional intraluminal pressure (*top panel*, See also Supplementary Video 10) and kymograph of branch elongation (*bottom panel*). In the DIC images, red lines and yellow arrows indicate the trajectory of tip EC displacement and the tip of the branch,

respectively. In the kymograph, red and blue dotted lines indicate time points at which additional intraluminal pressure was loaded and released, respectively. Scale Bars: 20 μm for top panel of **a**. **b-d**, Quantification of branch elongation and tip EC movement. Box plots of averaged branch elongation 3 hours before (pressure load (-)) and after (pressure load (+)) the 2nd pressure loading and 3 hours after the release (pressure release (+)), merged with individual plots showing the branch elongation changes in response to loading and then the release of intraluminal pressure (pressure load and release: n=11 from 2 devices, control: n=6 from 1 devices examined during the course of 1-2 independent experiments) (**b**). All box plots (**b**) show the interquartile range, with the middle line defining the median, and whiskers show the minimum and maximum values, excluding outliers. Outlier values are defined as being 1.5 times the interquartile range above and below the third and first quartile, respectively. Kymograph showing the dynamics in tip EC movement 50 minutes before and 100 minutes after loading additional intraluminal pressure (time=0, dotted red line) (**c**). Kymograph showing the dynamics in tip EC movement 50 minutes before and 100 minutes after the release of additional intraluminal pressure (time=0, dotted red line) (**d**). Wilcoxon signed rank test with a Bonferroni correction for **b**.

Hanada et al. Extended Data Figure 8

Extended Data Fig. 8: Phosphorylated myosin light chain 2 localization in the branch around forming *de novo* lumen in on-chip angiogenesis without and with ROCK inhibitor and without and with TG treatment

Representative confocal z-projection images in on-chip angiogenesis (Day 4), magnified images of the areas enclosed within squares bounded by white dots in the left images (z-projection), and their corresponding confocal x-y slice images at specific z-positions (slice) of the non-treated group (a), ROCK inhibitor-treated group (b) and both ROCK and TG-treated groups (c). All images were visualized by whole-mount staining with phosphorylated myosin light chain 2 (pMLC2), Moesin and DAPI (nuclei). Scale bars: 10 μm for confocal z-projection images in the right and x-y slice images and 20 μm for confocal z-projection images in the left.

Next, based on the data shown in Fig. 1-3, we furthered our understanding of the novel role of pericytes in angiogenic morphogenesis. This is highlighted in the latter part

of the revised manuscript. As pointed out by reviewer #2, pericytes can independently influence both lumen and branch expansion as well as branch elongation, and it is also possible that pericytes contribute to both processes via molecular pathways, such as the PDGF-BB–PDGFR β , reflecting independent biomechanical mechanisms. However, in this study, we focused on elucidating factors supporting our hypothesis that pericytes biomechanically promote forward tip EC movement and branch elongation via limiting lumen and branch expansion by stiffening the perivascular environment, which was based on the initially presented results. Therefore, we tackled the latter issue by presenting the *in vivo* and *in vitro* data in Fig. 4 showing that pericytes contributed to both branch diameter growth and branch elongations. Next, we presented data demonstrating a possible relationship between lumen and branch expansions and tip EC movement reflecting branch elongation, as shown in Fig. 5. These figures correspond to the previous Fig. 1 and 3, respectively. In Fig. 6, we showed the pericytes influence perivascular stiffening possibly via enhanced Col-IV VBM deposition (the data in parts of previous Fig. 4 and Fig. 5), and, in Fig. 7, we demonstrated that Col-IV deposition on VBM is functionally involved in branch elongation through restricting lumen and branch expansions due to perivascular stiffening (the data in parts of the previous Fig. 5 and new data). Notably, through additional experiments, we showed that effects of *COL4A1* and *COL4A2* knock-down in ECs on angiogenic morphogenesis, such as delayed branch elongation and expanded branch width, were reversed, at least partially, by perivascular stiffening in response to TG treatment of the gel, supporting the importance of mechanical properties of Col-IV VBM in regulating angiogenic morphogenesis (new data, Fig. 7h-l, Extended Data Fig.13b, See below). In addition, we demonstrated the importance of EC-derived Col-IV deposition in forming VBM around the tip (new data, Fig. 7a-e, See below).

Fig. 7

Hanada et al. Figure 7

Fig. 7: Functional involvement of Col-IV deposition on VBM in angiogenic morphogenesis

Evaluation of functional involvement of Col-IV deposition on VBM in on-chip angiogenesis with pericyte coculture. **a, b**, Representative fluorescent images (**a**) and magnified confocal z-projection images (**b**) on Day 3 visualized by UEA-1 Lectin (ECs), Col-IV and DAPI (nuclei), with (*COL4* si) and without (*CTL* si) knock-down of both *COL4A1* and *COL4A2* in either ECs (K.D. EC, left panels) or pericytes (K.D. Pericyte, right panels) using siRNAs. **c**, Quantification of Col-IV deposition on

VBM (for ECs, *CTL* si: n=18 from 2 devices, *COL4* si: n=20 from 3 devices, examined during the course of 2 independent experiments, for pericytes, (*CTL* si: n=22 from 2 devices; *COL4* si: n=22 from 2 devices, examined during the course of 2 independent experiments). **d**, Quantification of branch length (for K.D. ECs, *CTL* si: n=116 from 11 devices, *COL4* si: n=138 from 7 devices, examined during the course of 3 independent experiments, for K.D. pericytes, *CTL* si: n=95 from 4 devices, *COL4* si: n=69 from 5 devices, examined during the course of 3 independent experiments). **e**, Quantification of Tip expandability, an index reflecting branch diameter, around the tip of the branch (See also Fig.2g) (for K.D. ECs, *CTL* si: n=116 from 11 devices; *COL4* si: n=138 from 7 devices, examined during the course of 5 independent experiments, for K.D. pericytes, *CTL* si: n=95 from 4 devices; *COL4* si: n=69 from 5 devices, examined during the course of 2 independent experiments). **f**, Representative confocal z-projection images showing regression process of on-chip angiogenesis composed of *COL4* si ECs. In the *right panel*, Col-IV depositions on VBM (red lines), distal edge of the angiogenic branch (white dotted line) and tip cell nuclei (white arrows) were schematically superimposed on the confocal z-projection image in the *left panel*. **g**, Evaluation of changes in perivascular stiffness in on-chip angiogenesis with *COL4* si ECs by PTM analysis based on one parameter, Creep compliance (*CTL* si: n=41 from 4 devices; *COL4* si: n=45 from 4 devices, examined during the course of 3 independent experiments). **h-l**, Rescue experiments of *COL4* si in ECs by stiffening the gel with TG treatment in the on-chip angiogenesis assay. Representative fluorescent images showing the changes in ECs comprising angiogenic branches, with pericytes coculture, by knock-down of both *COL4A1* and *COL4A2* in ECs and the rescue effects on both produced by TG treatment (**h**). Box plots of branch length (**i**) and Tip expandability (**j**, See also Fig. 2g) in on-chip angiogenesis of ECs treated with *control* (*CTL*) si or both *COL4A1* and *COL4A2* si (*COL4* si) and without and with TG treatment (+TG) (for **i**, *CTL* si: n=53 from 4 devices; *COL4* si: n=53 from 5 devices; *COL4* si + TG: n=64 from 5 devices, examined during the course of 2 independent experiments, for **j**, *CTL* si: n=53 from 4 devices; *COL4* si: n=53 from 5 devices; *COL4* si + TG: n=64 from 5 devices, examined during the course of 2 independent experiments). Representative confocal z-projection images showing the differences in Col-IV deposition on VBM around areas where the *de novo* vascular lumen forms in the three groups, using *CTL* si ECs, *COL4* si ECs and *COL* si ECs with TG treatment (**k**), and quantification based on one parameter, the coverage ratio of Col-IV (*CTL* si: n=13 from 3 devices; *COL4* si: n=19 from 4 devices; *COL4* si + TG: n=13 from 4 devices, examined during the course of 2 independent experiments). Data are expressed as means \pm SD for **c** and **l**. All box plots show the interquartile range, with the middle line defining the median, and whiskers show the minimum and maximum values, excluding outliers, for **d**, **e**, **g**, **i** and **j**. Outlier values are defined as being 1.5 times the interquartile range above and below the third and first quartile, respectively. Scale bars: 50 μ m for **b**, **f** and **k**, 100 μ m for **a**, **h**. Mann-Whitney U test for **c-e** and **g** and Mann-Whitney U test with a Bonferroni correction for **i**, **j** and **l**.

Extended Data Fig. 13b

Extended Data Fig. 13b: Expressions of *COL4A1* and *COL4A2* genes and their knock-down using siRNAs in ECs and pericytes

Quantitative PCR data for gene expressions of *COL4A1* and *COL4A2* in HUVECs and hPIPCs, treated with *COL4A* si (siRNAs for both *COL4A1* and *COL4A2* genes) or control (CTL) si for 2 days. Data are expressed as the means of the relative values of $\Delta\Delta CT$ to one in CTL si (n=2 for each group, examined during the course of 2 independent experiments).

Finally, according to reviewer #2's suggestion, we clarified the mechanism by which ECs sense lumen and branch expansions, leading to cessation of forward tip EC movement and branch elongation, as shown in Fig. 8 (new data, See response to the next comment for more detailed information), by conducting additional experiments. Taken together, our current and previous data (*Nat Commun* **13**:2594, 2022) allow us to conclude that directional forward movement of tip ECs might be suppressed via impairment of F-BAR proteins localized at the leading front in response to lumen and branch expansion, followed by failure of Arp2/3 complex-mediated actin polymerization, which is counter-regulated by pericytes. Thus, ECs might sense branch and lumen expansions during lumen development via the BAR proteins essential for Arp2/3 complex actin polymerization, possibly accounting for the mechanism by which directional EC movement ceased.

This reviewer would have liked to see more mechanistic experiments investigating how increased stiffening of the surrounding area due to the shown increase in deposition is sensed by ECs and converted into biochemical signals that regulate the changes in lumen expansion and branch elongation. The data shown here only scratches the surface of uncovering how stiffness controls vascular morphogenesis.

Response:

Thank you for this comment. By conducting additional experiments according to reviewer #2's comment, we endeavored to clarify the mechanism(s) by which increased stiffening of the surrounding area due to increased deposition of Col-IV on VBM is sensed by ECs and converted into biomechanical signals that regulate changes in lumen and branch expansion and branch elongation. We previously showed that the F-BAR proteins, CIP4 and TOCA1, localized to the leading edge of tip ECs to promote Arp2/3 complex-mediated actin polymerization, which is essential for the directional movement leading to angiogenic branch elongation (*Nat Comm* **13**:2594, 2022). We

further identified CIP4 and TOCA1 to function as a mechanosensor of EC membrane stretching, in which failure of membrane localization of CIP4 and TOCA1 induced, at least, by ectopic intraluminal pressure loading-dependent vessel expansion resulted in impairment of Arp2/3 complex-driven directional movement of tip ECs (*Nat Comm* 13:2594, 2022). In the revised manuscript, based on the previous findings, we hypothesized that a similar mechanism may act during physiological lumen and branch expansions in angiogenesis to suppress directional tip EC movement and branch elongation. Therefore, we conducted new experiments to elucidate the possible mechanisms. To achieve this aim, we tested our hypothesis (new data, Fig. 8, Extended Data Fig. 14, 15, See below) by performing further experiments.

First, we evaluated the localization patterns of Arp2/3 complexes and F-BAR protein clusters in the tip ECs of on-chip angiogenesis of ECs for comparison with the degree of lumen development, and further examined the changes observed when we restricted lumen and branch expansions intentionally by stiffening the surrounding ECM with TG treatment. Whole-mount immunostaining data of on-chip angiogenesis demonstrated variations in the localization patterns of Arp2/3 complexes (detected by ARPC2) dependent on angiogenic branches, wherein some tip ECs were predominantly localized at the leading front, while others were not (Fig. 8a). On the other hand, when we restricted branch and lumen expansions by stiffening the ECM with TG treatment, more tip ECs showed predominant Arp2/3 complex localization at the leading front (Fig. 8a, b). We further found an inverse correlation between predominant Arp2/3 complex localization at the leading front of tip ECs and the degree of lumen development around the tip EC ($r=-0.4486$); in the tip ECs of the branch occupying more of the lumen (Fig. 8c and Extended Data Fig. 14), a smaller number of Arp2/3 complexes was localized at the leading front, regardless of TG treatment of the ECM gel (Fig. 8d). Similar localization patterns in CIP4 as well as TOCA1 tagged with EGFP in the N-terminus (EGFP-TOCA1) were observed (Fig. 8e-h, and Extended Data Fig. 15); CIP4 and EGFP-TOCA1, which were colocalized with Arp2/3 complexes (Fig. 8f), were more dominantly localized at the leading front of tip ECs when the ECM was treated with TG, and, in more lumen-dominated branches, smaller numbers of CIP4 and EGFP-TOCA1 were localized at the leading front of tip ECs ($r=-0.5547$, Fig. 8h and Extended Data Fig. 15). These data indicate that the localization of F-BAR protein clusters and the accumulation of ARP2/3

complexes at the leading front of tip ECs were suppressed in association with lumen development and its expansion.

Fig. 8

Hanada et al. Figure 8

Fig. 8: Loss in localization of F-BAR protein clusters and Arp2/3 complexes at the leading front of tip ECs in relation to lumen development

a-d, Relationships between Arp2/3 complex localization patterns around the tips of branches and the nearby lumen development in on-chip angiogenesis without or with TG treatment of the gel (Day 4). Representative z-projection confocal images evaluated by UEA-1 Lectin (ECs), ARP2C (Arp2/3 complex) and DAPI (Nucleus), showing different localization patterns of Arp2/3 complexes around the leading edges of angiogenic branches without (TG (-), *left and middle panels of a*) and with (TG (+), *right panel of a*) TG treatment of the gel. Areas within squares bounded by white dots are shown magnified on the right. Quantification of Arp2/3 complex localization around the leading edge of an angiogenic branch (*right panel of b*), assessed by the number of Arp2/3 complexes accumulated in the branch between the tip and the half-way point for the distance from the nuclear edge to the tip (*left panel of b*) (TG (-): n=42 branches from 4 devices; TG (+): n=26 branches from 4 devices examined during the course of 3 independent experiments). Z-projection confocal images evaluated by Moesin (Lumen) and DAPI, showing different extents of lumen development (**c**, Lumen, (-): none, (+++): well developed, (+): less developed), which corresponds to images of **a** (See also Extended Fig. 14). Distribution plot showing relationship between Arp2/3 complex localization around the leading edge of the branch and lumen development without (black circles) and with (red circles) TG treatment of the gel (*right panel of d*, TG (-): n=42 branches from 4 devices, TG (+): n=26 branches from 4 devices examined during the course of 3 independent experiments). The extent of lumen development was quantified with an index, % Lumen occupancy, calculated as the percentage of lumen occupation (red) in the 2D branch area ranging between 25 μm distal and proximal portions from the center of the nucleus of tip ECs (dotted lines), as shown in the drawing (*left panel of d*). **e-h**, Relationships between CIP4 localization patterns around the tips of branches and the nearby lumen development in on-chip angiogenesis without or with TG treatment of the gel (Day 4). Representative z-projection confocal images evaluated by UEA-1 Lectin, CIP4 and DAPI, showing different localization patterns of CIP4 around the leading edges of angiogenic branches without (TG (-), *upper and middle panels of e*) and with (TG (+), *lower panel of e*) TG treatment of the gel. Areas within squares bounded by white dots are shown magnified on the right. Line scan profiles of fluorescent intensities of CIP4 and ARPC2 along the yellow dotted line indicated in the confocal x-y slice image of the *left panel of f* (*right panel of f*), showing colocalization of CIP4 with Arp2/3 complexes. Quantification of CIP4 localization around the leading edge of an angiogenic branch, assessed in a manner similar to that employed in **b** (**g**, TG (-): n=42 branches from 4 devices; TG (+): n=26 branches from 4 devices examined during the course of 3 independent experiments). Distribution plot showing relationship between CIP4 localization around the leading edge of the branch and lumen development, assessed much as in **d**, without (black circles) and with (red circles) TG treatment of the gel (**h**, TG (-): n=42 branches from 3 devices; TG (+): n=26 branches from 3 devices examined during the course of 3 independent experiments). **i-l**, Relationship between CIP4 and Arp2/3 complex localization patterns around the tips of branches and the nearby lumen development in on-chip angiogenesis in the absence (PC (-)) or the presence (PC (+)) of pericytes (Day 4). Representative z-projection confocal images evaluated by UEA-1 Lectin, CIP4, ARPC2 and DAPI showing colocalization of CIP4 with Arp2/3 complexes around the leading edges of angiogenic branches (**i**), and images assessed by UEA-1 Lectin, ARPC2, Moesin and DAPI showing relationship between Arp2/3 complex localization pattern around the leading edge of the branch and lumen development (**k**). Regions indicated in squares bounded by white dots were magnified on the right in **i** and **k**, and the region within the square bounded by yellow dots is shown magnified in the *right panel of k*. Quantification of Arp2/3 complexes and CIP4 localization around the leading edge of an angiogenic branch, assessed in a manner similar to that employed in **b** (**l**, for Arp2/3, PC (-): n=45 branches from 4 devices; PC (+): n=44 branches from 4 devices, for CIP4, PC (-): n=37 branches from 5 devices; PC (+): n=25 branches from 5 devices, examined during the course of 3 independent experiments). Distribution plot showing relationship between ARPC2 or CIP4 localization around the leading edge of the branch and lumen development, assessed in a manner similar to that employed in **d**, in the absence (black circles) and the presence (red circles) of pericytes (**l**, for CIP4, PC (-): n=37 branches from 5 devices; PC (+): n=25 branches from 5 devices, for Arp2/3, PC (-): n=45 branches from 4 devices; PC (+): n=44 branches from 4 devices, examined during the course of 3 independent experiments). All box plots show the interquartile range, with the middle line defining the median, and whiskers show the minimum and maximum values, excluding outliers, for **b**, **g** and **j**. Outlier values are defined as being 1.5 times the interquartile range above and below the

third and first quartile, respectively. Scale bars: 10 μ m. Mann-Whitney U test for **b**, **g** and **j**.

Extended Data Fig. 14

Hanada et al. Extended Data Figure 14

Extended Data Fig. 14: Patterns of lumen development around the tip ECs in on-chip angiogenesis by ECs using gels without and with TG treatment

Representative confocal z-projection images and their corresponding confocal x-y slice images at specific z-positions (slice), visualized by whole-mount staining for Moesin (luminal surface) and DAPI (nuclei), in on-chip angiogenesis using gels without and with TG treatment (Day 4), showing different degrees of lumen development around the tip ECs (well developed: lumen (+++); less developed: lumen (+); not developed (-)). Scale bars: 10 μ m.

Extended Data Fig. 15

Hanada et al. Extended Data Figure 15

Extended Data Fig. 15: Relationship between EGFP-TOCA1 localization patterns around the tips of branches and nearby lumen development in on-chip angiogenesis without and with TG treatment of the gel

a, Representative z-projection confocal images evaluated by UEA-1 Lectin (ECs), EGFP-TOCA1, ARPC2 (ARP2/3 complex) and DAPI (nuclei), showing different localization patterns of EGFP-TOCA1 around the leading edges of on-chip angiogenic branches (Day 4) using gels without (TG (-), *upper and middle panels of a*) and with (TG (+), *lower panel of a*) TG treatment. Areas surrounded by squares with a boundary of white dots are shown magnified on the right. **b**, **c**, Line scan profiles of fluorescent intensity of EGFP-TOCA1 and ARPC2 along the yellow dotted line indicated in the confocal *x-y* slice image of the *left panel of b* (*right panel*), showing colocalization of EGFP-TOCA1 with Arp2/3 complexes (**c**). **d**, Quantification of EGFP-TOCA1 localization around the leading edges of angiogenic branches (TG (-): *n*=25 branches from 3 devices; TG (+): *n*=21 branches from 3 devices examined during the course of 3 independent experiments). All box plots show the interquartile range, with the middle line defining the median, and whiskers show the minimum and maximum values, excluding outliers. Outlier values are defined as being 1.5 times the interquartile range above and below the third and first quartile, respectively. **e**, Distribution plot showing relationship between EGFP-TOCA1 localization around the leading edge of the branch and lumen development without (black circles) and with (red circles) TG treatment of the gel (TG (-): *n*=25 branches from 3 devices; TG (+): *n*=21 branches from 3 devices examined during the course of 3 independent experiments). Scale bars: 10 μ m for *right panels of a, b* and 20 μ m for *left panels of a, b*.

Moreover, we similarly investigated the localization of CIP4 clusters and Arp2/3 complexes at the leading front of the tip ECs in on-chip angiogenesis with pericyte coculture, and examined the relationship between their localization and lumen

development. Consistent with the results we obtained when limiting branch and lumen expansions by treating the ECM with TG, a greater number of CIP4 colocalized with Arp2/3 complexes were present at the leading front of tip ECs in angiogenic branches with pericytes, as compared to those without pericytes (new data, Fig. 8i, j, See above). This finding was associated with less lumen occupation adjacent to the tip EC in the branch ($r=-0.4417$ for Arp2/3 complex, $r=-0.4638$ for CIP4, new data, Fig. 8k, l, See above). Taken together with our previous data (*Nat Commun* **13**:2594, 2022), these results strongly support our hypothesis and suggest that forward directional movement of the tip ECs might be suppressed via impairment of localization of the F-BAR proteins at the leading front in response to lumen and branch expansions, followed by failure of Arp2/3 complex-mediated actin polymerization. Furthermore, this process is also counter-regulated biomechanically by the presence of pericytes. Therefore, as one of the mechanisms, we currently speculate that ECs might sense vascular wall extension, induced by lumen and branch expansions, via the BAR proteins, and that the wall extension leads to failure in Arp2/3 complex-mediated actin polymerization at the leading front, which would be essential for forward directional EC movement. Thus, we believe that EC might indirectly sense increased perivascular stiffening through its influence on lumen and branch expansions. However, based on the present study, whether ECs directly sense the perivascular stiffening, leading to efficient branch elongation, while branch diameter narrows, remains unclear.

Therefore, we added the newly obtained data as Fig. 8 and described these novel findings in the text. In addition, in the discussion part, we elaborated on the remaining open question regarding the possibility that ECs might directly sense the perivascular stiffening, which may in turn control the cellular behaviors involved in vascular lumen formation and branch elongation. Please also see major changes regarding this important comment are below (Response to Major comment 5):

1. Added Fig. 8 and corresponding legend
2. Added Extended Data Fig. 14 and 15 and corresponding legend
3. Added *p* 2, *line* 38 - 40: Finally, we found that inhibition of forward directional movement of the tip EC during lumen development might be due to decreased localization of the F-BAR proteins and Arp2/3 complexes at the leading front.

4. Added *p 4, line 84 - 88*: Furthermore, we identified Fes/Cdc42 interacting protein-4 (CIP4) homology-Bin/Amphiphysin/Rvs (F-BAR) proteins²³, CIP4 and Transducer of Cdc42 dependent actin assembly-1 (TOCA1), as key regulators mediating the actin-related protein 2/3 (Arp2/3) complex-dependent-actin polymerization required for directional EC migration during angiogenesis⁹.
5. Added *p 5, line 108 - 110*: Finally, we demonstrated that decreased localization of the F-BAR proteins and Arp2/3 complexes at the leading front might be mechanistically involved in inhibiting the forward directional movement of tip ECs during lumen development.
6. Added *p 18, line 406 - p 20, line 456*: **Mechanism controlling tip EC movement during lumen and branch expansion**
Finally, we addressed the mechanisms by which the directional movement of tip ECs is controlled in response to lumen and branch expansions. ... Taken together with our previous data⁹, these results strongly support our hypothesis and suggest that forward directional movement of the tip ECs might be suppressed via impairment of localization of the F-BAR proteins at the leading front in response to lumen and branch expansions, followed by failure of Arp2/3 complex-mediated actin polymerization, which is counter-regulated biomechanically by the presence of pericytes.
7. Added *p 20, line 475 - p 21, line 479*: Finally, we found that the localization of F-BAR proteins and Arp2/3 complexes at the leading front of the tip ECs decreases with emergence and expansion of the nearby lumen, possibly suggesting a molecular mechanism inhibiting the forward directional migration of tip ECs during lumen development, and that this process is counter-regulated by the presence of pericytes.
8. Added *p 25, line 595 - p 26, line 601*: The present data suggest that a similar mechanism mediated by the F-BAR proteins might function in response to physiological lumen and branch expansions during lumen development, which is further controlled biomechanically by pericytes, although this possibility has yet to be confirmed by simultaneously monitoring the dynamics of the localization of the F-BAR proteins and lumen development. Moreover, the involvement of membrane tension in the mechanism remains unclear in the present study.

9. Added *p 26, line 606 - 608*: Also, in the present study, it cannot be completely denied whether ECs directly sense the perivascular stiffening, leading to efficiently elongating branch, while narrowing branch diameter.
10. Added *p 34, line 773 - 781*: **Plasmid**

A human TOCA1 cDNA was amplified by PCR using a pCS2+MT-hTocal1-wt plasmid (Addgene plasmid #33030) as the template and cloned into a pEGFP-C1 vector (Clontech, Takara Bio Inc.) to construct the pEGFP-C1-TOCA1 encoding N-terminally EGFP-tagged TOCA1 (EGFP-TOCA1). The EGFP-TOCA1 cDNA was also subcloned into the CSII-CMV-MCS-IRES2-Bsd (to construct the CSII-CMV-EGFP-TOCA1-IRES2-Bsd plasmid). The CSII-CMV-MCS-IRES-Bsd lentivirus expression vector and the packaging plasmid (pCAG-HIVgp, pCMV-VSV-G) were kindly provided by Dr. H. Miyoshi (BioResource Center, RIKEN).
11. *p 25, line 566 → p 34, line 783*: **Cell culture and lentivirus production and infection**
12. Added *p 34, line 791 - p 35, line 803*: Production of and infection with recombinant lentivirus encoding EGFP-TOCA1 were performed as we previously reported⁹. Briefly, EGFP-TOCA1 lentivirus vectors were transfected with the packaging plasmids (pCAG-HIVgp and pCMV-VSV-G) into 293T cells using Lipofectamine 3000 reagent according to the manufacturer's instructions (Thermo Fischer Scientific). Five hours after transfection, transfection media were replaced with growth media (Dulbecco's modified Eagle's medium with 10% FBS). After 48 hours, the conditioned media were collected and centrifuged at 1500 x g for 5 minutes. Then, the supernatants were filtered through a 0.45- μ m filter to remove floating cells and debris and stored at -80°C until use. The lentivirus encoding EGFP-TOCA1 were further concentrated using PEG-itTM Virus Precipitation Solution (System Biosciences) according to the manufacturer's protocol and stored at -80°C until use. HUVECs at the third or fourth passage were infected with lentivirus at the appropriate multiplicities of infection and used at passages 5-6.
13. *p 31, line 712 - 714 → p 41, line 952 - 956*: For the VBM analysis, localization patterns of F-BAR protein clusters, Arp2/3 complexes and pMLC2 as well as lumen development, fluorescent confocal images were obtained at 0.3 to 0.5 μ m intervals

along the z-axis using a 63x objective lens (Oil immersion, HC PL APO CS2, 1.40 NA, Leica).

14. Added p 51, line 1197 - p 52, line 1212: **Quantification of localization patterns of F-BAR protein clusters and Arp2/3 complexes and their relationships to lumen development in on-chip angiogenesis**

The localization patterns of both F-BAR protein clusters and Arp2/3 complexes at the tips of angiogenic branches in on-chip assays were assessed under different culture conditions. ... % Lumen occupation was then calculated by dividing the lumen area by the branch area, followed by evaluation of the relationship with the localization patterns of either F-BAR protein clusters or Arp2/3 complexes.

15. p 40, line 922 - 926 → p 52, line 1215 - 1221: Branch lengths, branch diameters, lumen area changes, **Tip expandability**, relative changes in branch diameter with pressure loading, **lumen area change**, the coverage **ratios** of Col-IV and laminin, the fluorescence intensities per volume of Col-IV and laminin, **values of creep compliance in the PTM analysis and numbers of CIP4, TOCA1-EGFP and Arp2/3 complexes** in on-chip angiogenesis or in murine retinal angiogenesis were compared between groups using the Mann-Whitney U test.

16. Added p 53, line 1235 - 1237: **Correlations between % Lumen occupation and numbers of CIP4, TOCA1-GFP and Arp2/3 complexes were evaluated by applying the Pearson correlation coefficient.**

Major Comment 1:

When summarizing Figure 1, the authors write “These results suggest a causal relationship between branch elongation and circumferential lumen expansion.” However, the authors are reporting correlation data for these metrics. The loss of function experiment in this figure is involving +/- pericytes and resulting changes in the branch morphology. The authors are neglecting the role pericytes and PDGFR- β might play in regulating these morphological metrics and are assuming A (elongation) causes B (diameters) when it is possible C (pericytes) may be regulating both A and B which is shown by this data.

Response to Major comment 1 :

We appreciate reviewer #2 for these insightful and helpful comments. We agree with reviewer #2's opinion that the data showing changes in branch length and diameter in the presence versus the absence of pericytes (shown in previous Fig. 1) does not simply imply a possible causal relationship between lumen and branch expansions and branch elongation, and that these data raise several possibilities, one of which is that pericytes may independently influence both lumen and branch expansion and branch elongations. Also, the pericyte effects might be mediated through molecular pathways, such as the PDGF-BB-PDGFR β signaling, rather than biomechanically.

As also described in the above response, in this study, we aimed to clarify whether spontaneous lumen and branch expansions during lumen development inhibit branch elongation, and we then aimed to investigate the role of pericytes in promoting branch elongation, while the lumen undergoes development. However, regrettably, due to inappropriate logical development and lack of supporting data, we were unable to properly convey our argument in the previous manuscript.

Therefore, in the revised manuscript, by conducting additional experiments and adding new data, we significantly changed the logical flow and order of data presentation in the text and figures as mentioned above. In response to major comment 1 from reviewer #2, the main points of improvement are described below. In the first half of the manuscript, we clarified whether the lumen and branch expansions during lumen development physically inhibit forward tip EC movement and branch elongation by adding new data to those addressing elucidation of causality (Fig. 1-3 including new data, See above). In the second half of our study, we demonstrated the role of pericytes in promoting branch elongation, during lumen development, while further strengthening the supporting data. In this study, based on the results obtained formerly, we focused on verifying our hypothesis that pericytes might promote branch elongation via biomechanically restricting excessive lumen and branch expansions. We added new data that strongly support our hypothesis, showing retarded branch elongation and widened branch diameter in response to perivascular softening of VBM due to *COL4* knock down in ECs in the presence of pericytes, an effect that was reversed by further counter-stiffening of the

perivascular environment using TG-treated gel with no recovery of Col-IV deposition on VBM (new data, Fig. 7h-l, See above). However, as pointed out by reviewer #2, it cannot be completely ruled out that other mechanisms, such as via molecular pathways including PDGF-BB-PDGFR β signaling rather than biomechanically, might also be involved. The open question remains unresolved in the present study. Thus, we elaborated on this point in the discussion section.

Major changes made in response to Major comment 1 in terms of the pericyte role in regulation of lumen and branch expansions and branch diameter are outlined below:

1. Fig. 1c-j \rightarrow Fig. 4a-j and corresponding legend
2. Added Fig. 4f and k and corresponding legend
3. Fig. 3 \rightarrow Fig. 5 and corresponding legend
4. Fig. 4b, c \rightarrow Fig. 6a and corresponding legend
5. Fig. 4f \rightarrow Fig. 6b and corresponding legend
6. Fig. 5a-e \rightarrow Fig. 6c-g and corresponding legend
7. Fig. 5f-l \rightarrow Fig.7a-d, f and g (partially added) and corresponding legend
8. Added Fig. 7e and h-l and corresponding legend
9. Extended Data Fig. 5a \rightarrow Extended Data Fig. 2a and corresponding legend
10. Added Extended Data Fig. 4b and corresponding legend
11. Added Extended Data Fig. 5 and corresponding legend
12. Extended Data Fig. 2 \rightarrow Extended Data Fig. 9 and corresponding legend
13. Extended Data Fig. 5b \rightarrow Extended Data Fig. 10 and corresponding legend
14. Extended Data Fig. 6 \rightarrow Extended Data Fig. 11 and corresponding legend
15. Extended Data Fig. 7 \rightarrow Extended Data Fig. 12 and corresponding legend
16. Extended Data Fig. 8 \rightarrow Extended Data Fig. 13 (includes new data) and corresponding legend
17. *p 5, line 116 - 119* \rightarrow *p 12, line 263 - 270*: **Pericyte effects on angiogenic morphogenesis driven by ECs**

The next question was whether there is an additional regulatory system further controlling branch elongation efficiently, while lumen formation progresses, in the

process of angiogenic morphogenesis seen during normal developmental stages. To address the question, we focused on morphological differences between angiogenic branches with and without pericyte coverage. Pericytes, one of the cell types comprising the vasculature, is present adjacent to ECs even in elongating angiogenic branches (Fig. 4a)³¹⁻³³.

18. *p 6, line 125 - 129* → *p 12, line 276 -282*: In on-chip angiogenesis of ECs obtained by adding pericytes, angiogenic branches covered by pericytes and with a luminal structure were induced (Fig. 4f, g), and the actions of pericytes on vascular morphogenesis observed in murine retina, such as enhanced branch elongation (Fig. 4h, i) and reduced branch diameter including around the tip of the branch (Fig. 4h, j, k), were reproducible. These observations confirmed that pericytes control branch elongation as well as lumen and branch expansion during angiogenic morphogenesis.
19. *p 8, line 173 - 188* → *p 12, line 283 - p 13, line 297*: We further analyzed tip EC movement changes in the on-chip angiogenesis assay performed with pericyte coculture. Time-lapse imaging showed that, in clear contrast to the dynamics of tip EC movement in EC-only on-chip angiogenesis, in the presence of pericytes, most of the tip ECs tended to continue moving forward, characterized by a higher frequency of forward movement with greater speed (Fig. 5a-c and Extended Data Fig. 10, See also Supplementary Video 11), which is highly similar to the pattern of tip EC movement when lumen expansion was restricted using TG-treated stiffened ECM (Fig. 2h, i and Extended Data Fig. 4, See also Supplementary Video 7). On the other hand, the sizes of nearby lumens remained essentially constant, though occasional decreases were seen, regardless of whether the lumen was distal or proximal to the nuclei, and excessive expansion was rarely noted (Fig. 5c-e), in contrast to the condition without pericytes (Fig. 1h, i). Notably, one of 20 tip ECs showed apparent deceleration of forward movement after nearby lumen expansion, despite having been cocultured with pericytes, supporting the presence of an inverse correlation between lumen expansion and tip EC forward movement driving branch elongation.
20. *p 9, line 189 - 201* → *p 13, line 298 - p 14, 312*: Also, to clarify the relationships between the forward directional movement of tip ECs and lumen and branch

expansions *in vivo*, we examined the position of the Golgi apparatus relative to the nucleus, an anterior side marker in moving cells with anterior-posterior cell polarity^{15,16} ... Taken together with the aforementioned results of TG treatment, these results also suggest that for the integrating mechanism, pericytes maintain the forward directional movement of tip ECs possibly by restricting lumen and branch expansions, although we cannot completely rule out the possibility that pericytes control branch elongation and lumen expansion independently.

21. *p 9, line 203 → p 13, line 314 - 315:* **Perivascular stiffening by pericytes via deposition of type-IV collagen on vascular basement membrane**

22. *p 9, line 207 - p 10, line 221 → p 14, line 316 - line 325:* A previous study demonstrated micro-vessel distensibility in response to increased intraluminal pressure (ΔP) to, at least theoretically, depend on physical properties of the surrounding ECM *in vivo* (Fig. 2a)²². Indeed, in a vascular distensibility test, the branches around the tip ECs rarely expanded in response to additional pressure loading when pericytes were present (-0.12 ± 2.6 % increase in branch diameter), whereas the same pressure loading caused lumen and branch expansions in the absence of pericytes (7.7 ± 2.7 % increase in branch diameter) (Fig. 6a), suggesting increased perivascular stiffening with coexistent pericytes. Thus, we determined whether pericytes stiffen the extravascular environment surrounding the tips of angiogenic sprouts, where *de novo* lumens emerge and expand, using the PTM method.

23. Previous *p 13, line 298 - p 14, line 321 → p 16, line 371 - p 17, line 404:* **Functional involvement of Col-IV deposition on VBM in angiogenic morphogenesis**

Subsequently, we examined the functional involvement of Col-IV deposition on VBM in the enhancement of branch elongation via stiffening of the perivascular structure employing interference in the gene expression of Col-IV. Consistent with the findings of a previous study³⁴, both ECs and pericytes, used for the on-chip angiogenesis assay, expressed *COL4A1* and *COL4A2*, Col-IV subunits of VBM (Extended Data Fig. 13a), though ECs are reportedly more likely to be a major contributor to Col-IV deposition on VBM³⁴. ... These results collectively suggest that

spatiotemporally-enhanced deposition of EC-derived Col-IV on VBM in the presence of pericytes controls perivascular stiffness appropriately, which promotes branch elongation by physically restricting both lumen and branch expansions.

24. p 15, line 335 - 338 → p 20, line 472 - 475: We further demonstrated that pericytes mediate proper Col-IV deposition by ECs onto the VBM, which accounts for spatiotemporal perivascular stiffening and the resultant reduction in vascular distensibility.
25. p 16, line 374 - 375 → p 22, line 520 - 523: We further revealed that VBM stiffening via enhanced deposition of Col-IV secreted dominantly from ECs causes branch thinning in the presence of pericytes, which is consistent with a previous result showing the importance of Col-IV VBM³⁴.
26. p 26, line 601 - 608 → p 36, line 833 - 841: Accell siRNA Delivery Media (B-005000-500, Horizon Discovery) with SingleQuots™ Supplements and with (for HUVECs) or without (for hPIPCs). Growth Factors (CC-4176, Lonza) served as the culture medium for the knockdown procedure. The supplements and the growth factors were diluted in the medium to the concentration recommended by the manufacturer. For double knockdown of COL4A1 and COL42, cells were cultured with 0.5 (for HUVECs) or 1 (for PIPCs) μM COL4A1 siRNA and 0.5 (for HUVECs) or 1 (for PIPCs) μM COL4A2 siRNA. As the negative control, cells were cultured with 1 (for HUVECs) or 2 (for PIPCs) μM non-targeting siRNA. After a 24-hour culture period, the cells were used for the experiments.

Major Comment 2:

It is not immediately clear how the graph in Figure 2c shows EC deceleration immediately after lumen expansion. When the lumen area first increases ~90 minutes, the EC moving distance increases and accelerates (slope of line increases). What is the reason for choosing these time points for the dotted lines? It would be helpful for these lines to show when the lumen expansion and deceleration begins as this is unclear and would provide more clarity on where the written interpretation is derived from.

Response to Major comment 2 :

First of all, we apologize for the confusion caused by incorrectly drawing the iii mark corresponding to the 110-minute timepoint in the previous Fig. 2c (orange arrow), in which the line was drawn slightly to the left of where it should have been (blue arrow and dotted line in the previous Fig. 2c of Extra Fig. 5, See below). The point iii is the deceleration timepoint of EC movement, which corresponds to time 0 indicated by the orange dotted line in the previous Fig. 3e. Based on the time-lapse imaging data, the deceleration point of EC movement was determined as described in the method section (previously, *p 35, line 821- p 36, line 832; p 42, line 1087 - 1098* in the revised manuscript). First, local maximum timepoints were identified based on the EC movement distance-time plot. The local maximum timepoint T is defined as the point satisfying $y(T - 5min) < y(T)$ and $y(T) \geq y(T + 5min)$, where $y(T)$ is the position of the EC at time T . Then, the deceleration point was selected from the local maximum points based on the following criteria:

1. $y(T) - y(T - 50min) > 5 \mu m$,
2. $y(T + 50min) - y(T) < 5 \mu m$,
3. $y(T) - y(T + 15min) > 0 \mu m$.

Extra Fig. 5

Extra Fig. 5: Left panel of Fig. 2c in the previous manuscript and left panel of Fig. 1f in the revised manuscript

Orange and blue lines indicate incorrect and correct ones in previous Fig. 2c, respectively (right panel). Corrected figure in Fig. 2f of the revised manuscript (right panel).

Therefore, in the revised manuscript, we replaced the previous Fig. 2c with revised figure as Fig. 1f, and added the description that line iii indicates the deceleration point of EC movement in the legend.

Major Comment 3:

Stating that the data in Figure 2 shows an inhibitory effect between these two variables is a misleading interpretation that is not supported by this data. There does appear to be an inverse correlation, but there is not sufficient data to claim lumen expansion is inhibiting EC tip movement. There is no real “experiment” being run in this figure to uncover mechanisms such as inhibition. Just observations and correlation data.

Response to Major comment 3 :

Thank you for these highly insightful comments. As also discussed above, we fully agree that we showed a possible inverse correlation between circumferential lumen and branch expansion and inhibition of tip EC movement and branch elongation, though we did not demonstrate causality, as shown in the previous Fig. 2 and the following Figures. To clarify the possible causal relationship between two variables, in the revised manuscript, we additionally conducted three different experiments, in which we examined the changes in directional tip EC movement or/and branch elongation when intentionally controlling lumen and branch expansions. We thereby obtained positive data suggesting the presence of causality and thus significantly reorganized the manuscript by adding the newly obtained data in the aforementioned manner. In response to the first comment made by reviewer #2, the major important points of improvement are described below (See also response to the 1st comment above).

In the revised manuscript, we describe attempting to control lumen and branch expansions in three different ways. Theoretically, micro-vessel (lumen) diameters are determined by the balance between intraluminal and extraluminal pressures and by the stiffness of the vessel wall itself as well as that of the surrounding ECM. First, we attempted to restrict lumen and branch expansions via stiffening of the perivascular environment. In the previous manuscript, we had already demonstrated that stiffening of the surrounding ECM in response to TG treatment induced an increase in elongation with

reduction of the branch width in on-chip angiogenesis without pericytes (previous Fig. 4g-k). Therefore, in the revised manuscript, using the time-lapse imaging data, we additionally showed changes in the dynamics of tip EC movement and lumen expansion when the ECM was stiffened by TG treatment, and examined the relationship between these two variables. We then showed ECM stiffening to potentially reduce acceleration of forward tip EC movement in conjunction with suppression of lumen expansion (new data, Fig. 2h-l, Extended Data Fig. 4 and 5, See also above), i.e., we obtained data collectively supporting the presence of causality, i.e., evidence that lumen expansion causes branch elongation.

Next, we physically controlled lumen and branch expansions by ectopically applying additional intraluminal pressure to the vascular wall, inducing lumen and branch expansions, by increasing hydrostatic pressure, as well as by releasing this pressure to induce their shrinkage in the on-chip angiogenesis assay (new data, Fig. 3a-c, See also above). When possible, the load-release cycle was performed twice. We observed that in most cases, in rapid response to lumen and branch expansions, forward tip EC movement and branch elongation slowed down, but then retraction was noted shortly afterward (new data, Fig. 3d-g, Extended Data Fig. 6, See also supplemental Video 8). It is noteworthy that tip EC movement, which had stopped, gradually resumed after the expanded lumen and branch shrank, thereby again contributing to branch elongation (new data, Fig. 3d, e, h, i, Extended Data Fig. 6). These dynamic changes in tip EC movement and branch elongation in response to externally applied intraluminal pressure and its release were both reproduced during the 2nd load-release cycle (new data, Extended Data Fig. 7, See also supplemental Video 10). In clear contrast, in the absence of intraluminal pressure, movement of some tip ECs decelerated regardless of the presence or absence of intraluminal pressure loading, whereas most tip ECs continued to move forward over the observational period (Extended Data Fig. 6, See also supplemental Video 9). These results suggest that, at a minimum, ectopic lumen and branch expansions inhibit forward tip EC movement and branch elongation.

Finally, we applied pharmacological interventions to induce lumen and branch expansions in addition to the physical intervention described above (new data, Fig. 3j-m, see also above). Previous studies (*Nat Cell Biol* 18: 443–450, 2016; *Nat Cell Biol* 18:311–318, 2016; *Circ Res* 119:810–826, 2016; *Nat Commun* 11:5476, 2020) showed that in

both epithelial and endothelial tubulogenesis, expansion of the *de novo* lumen is limited by apical actomyosin contractility via Ras homologue gene family member A (RhoA) - Rho-associated protein kinase (ROCK) – non-muscle myosin II (NMII) signaling. Therefore, we blocked the RhoA-ROCK-NMII signaling using an inhibitor of ROCK, Y27632, and an inhibitor of NMII, Blebbistatin, in on-chip angiogenesis of ECs to induce circumferential expansion of both lumen and branch by inhibiting apical actomyosin contractility. As expected, both inhibitor treatments induced branch and lumen expansions around the tip, while concomitantly reducing branch elongation (new data, Fig. 3j-m, Extended Data Fig. 8). Furthermore, the reduction in branch elongation was reversed by physically restricting branch and lumen expansions via stiffening of the surrounding ECM by TG treatment (Fig. 3j-m), quite possibly suggesting that branch and lumen expansions physically inhibit branch elongation. Taken together, these data obtained employing three different intervention strategies suggest that there is a causal link between lumen development and branch elongation, in which lumen and branch expansions inhibit forward directional movement of tip ECs, resulting in branch elongation.

Therefore, in the revised manuscript, we added the new data in Figures and Extended Data figures. Furthermore, in accordance with the new data, we also added or changed the text as appropriate. In addition, we introduced a new parameter, i.e., “Tip expandability”, which more specifically evaluates branch diameter around the tip where the *de novo* lumen develops, and used this parameter for the experiments shown in Fig. 2g, 3l, m, 4k and 7e and i.

Fig. 2g

Tip expandability

Major changes made in response to Major comment 3 regarding a causality between lumen and branch expansions and branch elongation are below:

12. Added Fig. 2h-l and 3, and corresponding legends
13. Extended Data Fig. 4b and 5-8, and corresponding legends
14. Added *p 8, line 168 - 171*: To verify the possible causal relationship between branch elongation and lumen development, we investigated changes in the dynamics of directional EC movement and branch elongation when the degrees of lumen and branch expansions were externally altered in different ways, in the on-chip angiogenesis of ECs.
15. *p 11, line 247 - 250* → *p 8, line 176 - 179*: Thus, to intentionally restrict branch and lumen expansion in the on-chip angiogenesis assay, we first increased the stiffness of ECM by treatment with Transglutaminase (TG)^{25,26}, a crosslinking enzyme of fibrin gels, without changing the biochemical compositions of the on-chip assay of ECs.
16. *p 11, line 255 - 260* → *p 9, line 207 - p 10, line 218*: This “*move forward and stop*” movement pattern was repeated during branch elongation in on-chip angiogenesis of ECs using the non-treated ECM (Fig. 2h and Extended Data Fig. 4). In contrast, when the ECM was treated with TG, most of the tip ECs tended to continue moving forward, as characterized by a higher frequency of forward movement with greater speed, such that the aforementioned movement pattern was suppressed (Fig. 2h, i and Extended Data Fig. 4, See also Supplementary Video 7). However, the sizes of nearby lumens remained essentially constant, though occasional decreases were seen, regardless of whether the lumen was distal or proximal to the nuclei, and expansion was rarely noted (Fig. 2j-l and Extended Data Fig. 5), in clear contrast to the condition without TG treatment (Fig. 2i). These results indicate that restriction of excessive lumen expansion promotes branch elongation with a higher frequency of forward directional movement of tip ECs.
17. Added *p 10, line 219 - p 11, line 242*: Next, we ectopically altered the expansion states of both lumen and branch by increasing intraluminal pressure externally and then restoring it. To achieve this, additional hydrostatic pressure ($\Delta P=41\pm 14$ Pa) with

culture media was loaded into the intraluminal spaces of elongating branches and it was then released 3 hours after the loading in on-chip angiogenesis of ECs.... Similar dynamic changes in directional tip EC movement and branch elongation in response to external loading and release of intraluminal pressure could be repeatedly reproduced in the second round of the pressure load-release cycle, although the number of branches in which this could be confirmed was methodologically limited (Extended Data Fig. 7, See also Supplementary Video 10). These results clearly indicate that ectopic lumen and branch expansions can inhibit branch elongation driven by forward directional EC movement.

18. Added *p 11, line 243 - p 11, line 261*: Finally, we attempted to induce lumen and branch expansions by pharmacological interventions. It was previously reported that, in both epithelial and endothelial tubulogenesis, expansion of the *de novo* lumen is limited by apical actomyosin contractility via Ras homologue gene family member A (RhoA) - Rho-associated protein kinase (ROCK) – non-muscle myosin II (NMII) signaling^{24,28-30}. Collectively, these findings strongly suggest a causal link between lumen development and branch elongation, in which lumen and branch expansions inhibit forward directional movement of the tip ECs, resulting in retardation of branch elongation.
19. *p 14, line 329 - p 15, 333 → p 20, line 465 - 470*: An observational approach combining *in vitro* time-lapse imaging of the on-chip angiogenesis assay with *in vivo* validation, together with physical and pharmacological interventions specifically intended to control lumen and branch expansions, showed that lumen emergence followed by its expansion inhibits directional movement of the tip ECs polarizing along the antero-posterior axis, resulting in delayed branch elongation.
20. Added *p 38, line 883 - 888*: In the experiments to examine the effects of an inhibitor of ROCK, Y27632 (FUJIFILM Wako Pure Chemical Corp.), and an inhibitor of NMII, Blebbistatin (Merck), on on-chip angiogenesis, EGM-2 were replaced with new culture media including Y27632 (10 nM) or Blebbistatin (10 μM) or DMSO (Wako), twice at a 30-minute interval one day after introducing HUVECs. Two days later, the devices were subjected to whole-mount staining.

21. Added p 42, line 986 - p 43, line 1001: To assess the influences of lumen and branch expansions on the dynamics of tip EC movement and branch elongation, after acquiring at least 12 hours of time-lapse imaging data, similarly, approximately 40 Pa of hydrostatic pressure was loaded onto the intraluminal space of angiogenic branches to intentionally induce lumen and branch expansions, and time-lapse imaging was then restarted (Fig. 3a). ...Furthermore, after completion of the time-lapse imaging study, the actual values of the applied hydrostatic pressure were measured using the pressure gauge through a similar hydrostatic pressure loading procedure. The average values were 41 ± 14 Pa (n=6).

22. Added p 46, line 1072 - 1078: In the intervention study assessing vascular intraluminal pressure, in addition to tip ECs, the tips of the angiogenic branches were simultaneously tracked in a similar manner on DIC images obtained every 15 minutes. From the tracking data, we determined the average speed (/minute) of individual tip ECs during specified periods after loading and the release of intraluminal pressure, as well as the average branch elongation for 3 hours before and after intraluminal pressure loading and 3 hours after its release.

23. Added p 48, line 1124 - p 49, line 1132: *Tip expandability*

To more sensitively evaluate the extent of the angiogenic branch diameter around the tip, the parameter “Tip expandability” was selected. For this, we initially calculated the straight-line distance from the base of the branch to the tip (corresponding to branch length (l)), and then drew a line intersecting with it at the half-way point to determine both ends of the width of the branch. Tip expandability ($\tan \theta$) was calculated by the following formula (See also Fig. 2g),

$$\tan \theta = (l/2) / (d/2) = l / d,$$

where d is a distance between both ends of the width of the branch at the half-way point.

Major Comment 4:

The same issue occurs in Figure 3 as in Figure 1. The independent variable in these experiments is the presence of pericytes, which is resulting in changes in EC lumen diameter and impacting the movement of ECs. From this data alone, the authors claim that “lumen emergence followed by its expansion inhibits forward directional movement

of nearby ECs” is not substantiated by the presented data.

Response to Major comment 4 :

Thank you for these kind and very helpful suggestions. We agree with reviewer #2's opinion. Therefore, as mentioned above, in the revised manuscript, we significantly changed the logical flow, and in the first half, we showed that lumen and branch expansions can cause branch elongation by conducting new on-chip angiogenesis experiments without pericytes (See also response above).

First, we hypothesized that an additional control system may exist to more efficiently control branch elongation during lumen formation processes in normal development, and that pericytes control these processes for the following reasons. One involves morphological differences between angiogenic branches with and without pericyte coverage. Morphological analyses in both murine retinal and on-chip angiogenesis assays demonstrated the action of pericytes in both promoting branch elongation and limiting lumen and branch expansions (Fig. 4; previous Fig. 1c-f). Second, time-lapse imaging analysis of on-chip angiogenesis, together with whole-mount immunostaining of murine retinal angiogenesis, suggested the existence of an inverse correlation between lumen and branch expansions and forward tip EC movement causing branch elongation (Fig. 5; previous Fig. 3), as reviewer #2 also pointed out. In particular, as the mechanism, we hypothesized that pericytes stiffen the perivascular environment, thereby biomechanically promoting branch elongation through restriction of lumen and branch expansions. This is because, in the former parts of the study added to the revised manuscript, we found that stiffening of the perivascular environment by TG treatment limits lumen and branch expansions and promotes forward tip EC movement and branch elongation (Fig. 2d-l and Extended Data Fig. 4, 5, including new data, See also response above), observations very similar to the angiogenic morphogenesis changes seen in the presence of pericytes (Fig. 4, 5), suggesting the existence of a similar underlying biomechanical mechanism. Therefore, in the following parts, we focused on verifying our hypothesis.

To this end, we first investigated whether pericytes stiffen the perivascular environment by analyzing changes in vascular distensibility that indirectly reflects

perivascular stiffness in the on-chip angiogenesis assay with coculture in the presence of pericytes. We found vascular distensibility to be reduced in the presence of pericytes (Fig. 6a; previous Fig. 4b, c), quite possibly suggesting an increase in perivascular stiffness. Furthermore, we directly measured the stiffness of the perivascular ECM using PTM analysis, and found the stiffness of the areas around ($< 5 \mu\text{m}$ from branch, Near) angiogenic branches at the tip to be increased in the presence of pericytes (Fig. 6b; previous Fig. 4f). Collectively, these results indicate that coculturing ECs with pericytes increased the stiffness of the ECM surrounding the angiogenic branches, strongly supporting our hypothesis that pericytes stiffen the perivascular environment, restricting lumen and branch expansions, possibly resulting in effective forward directional movement of tip ECs and, consequently, elongation of angiogenic branches.

We subsequently tackled the issue of how the perivascular structure around the tips of angiogenic branches became stiffened in the presence of pericytes. Based on previous reports, in this study we focused on the formation, and especially the deposition, of Col-IV, which is regarded as being a primary determinant of the mechanical properties of the VBM because of its extensive cross-linkages. We found that Col-IV deposition on VBM around the position of *de novo* lumen development was increased in the presence of pericytes in both on-chip and murine retinal angiogenesis assays (Fig. 6c-g; previous Fig. 5a-e). Additionally, in the on-chip angiogenesis assay, we demonstrated that knock-down of both *COL4A1* and *COL4A2*, Col-IV subunits of VBM, in ECs but not pericytes reduced Col-IV deposition on VBM around the branch portion where the *de novo* vascular lumen was developing, and further showed that the knock-down in ECs delayed branch elongation with an enlarged branch diameter even when cocultured with pericytes, as seen in the absence of pericytes (Fig. 7a-f including new data; previous Fig. 5f-k). Thereby, we newly confirmed EC-derived COL4 deposition on VBM around the branch portion where *de novo* lumen develops to be important for promoting branch elongation during lumen development, by showing no significant effects of *COL4A1* and *COL4A2* knock-down in pericytes on angiogenic morphogenesis in additional experiments (Fig. 7a-e and Extended Data Fig. 13b, including new data). In the knock-down experiments, we further found that reduced Col-IV deposition on VBM by *COL4A1* and *COL4A2* knock-down induced perivascular softening, and that enlarged branch diameter and delayed branch elongation by *COL4A1* and *COL4A2* knock-down in ECs was restored by stiffening of

the perivascular ECM in response to TG treatment of the gel (new data, Fig.7h-l, See also response above). These results collectively suggest that spatiotemporally-enhanced deposition of EC-derived Col-IV on VBM in the presence of pericytes controls perivascular stiffness appropriately, thereby promoting branch elongation by physically restricting both lumen and branch expansion.

Therefore, in the revised manuscript, we changed the order of the data presentation and the corresponding documents and added the new data to the text and as Fig and Extended Data Fig, as described above.

Major Comment 5:

The conclusion of Figure 4 on lines 241-244 is a critical turning point of the paper where the data supports stiffening of the surrounding matrix due to pericytes, which provides a plausible, potential explanation for changes in EC movement. However, there is a dire lack of mechanistic studies investigating how this stiffening causes these changes in ECs.

Response to Major comment 5 :

We appreciate reviewer #2 for this helpful comment aimed at improving our manuscript. In the previous manuscript, investigation of the underlying mechanism by which perivascular stiffening in the presence of pericytes enhanced forward tip EC movement and branch elongation was not addressed, and we limited ourselves to elaborating on the possibility of this mechanism in the discussion part (*p 18, line 419 - p 19, line 429*) based on our previous report (*Nat Commun 13:2594, 2022*). According to reviewer #2's comment, as described in the revised manuscript, we conducted new experiments to address the mechanism, as mentioned above (See also response and details of the data noted above). Based on our previous report (*Nat Commun 13:2594, 2022*), we hypothesized that, during lumen development, ECs sensed branch and lumen expansions via the BAR proteins which resulted in impairment of Arp2/3 complex-driven directional movement of tip ECs, followed by retardation of branch elongation, which was counter-regulated biomechanically by pericytes via stiffening of the perivascular environment.

Therefore, we speculated that ECs might indirectly sense increased perivascular stiffening through its effects on lumen and branch expansions. By conducting experiments to verify our hypothesis, we obtained data strongly supporting it (new Fig. 8), as shown above (See also the details presented above).

Therefore, in the revised manuscript, we added new Figs. and Extended Data Figs, and in accordance with the new data, we added and changed the text, as described above.

Minor comment 1:

Fig 1b – cross-sectional image of lumen would be helpful here from supplement

Response to Minor comment 1:

Thank you for this helpful suggestion. As suggested, we moved the cross-sectional image of the lumen shown in the previous Extended Data Fig. 1d to Fig. 1c in the revised manuscript.

Minor comment 2:

The size of the standard deviation bars in Figure 1g and 1j stood out to me as a concern. Is there a plausible explanation for this degree of variability?

Response to Minor comment 2:

Thank you for pointing this out. The right panels of the previous Fig. 1g, j showed the average branch diameter calculated from the width data of all branch sections from root to tip of all analyzed branches, as shown in the left panels of the previous Fig. 1g, j. The analyzed algorithm was explained in the methods section of the previous manuscript (*p 36, line 842 – p 37, line 852*). Therefore, the increased size of the standard deviation bars would appear to be reasonable.

In addition, we introduced a new parameter, i.e., “Tip expandability”, which more specifically evaluates branch diameter around the tip where the *de novo* lumen

develops, and used this parameter for the experiments shown in Fig. 2g, 3l, m, 4k and 7e and i, as described above.

Minor comment 3:

Fig 2c – Typos on both y-axes and there are two i lines and no ii line.

Response to Minor comment 3 :

We apologize for having mistyped “ii” as “i” in the previous Fig. 2c. We made the necessary correction, and it has been moved as Fig. 1f in the revised manuscript.

Finally, we again deeply appreciate the reviewer’s supportive and insightful input and comments on our work. According to the reviewer's helpful suggestions, by performing additional experiments and extensively revising the manuscript based on the new data obtained, we believe we had made the manuscript significantly improved although a lot still remain to be figured out.

Replies to the reviewer's comments

Reviewer #1 (Remarks to the Author):

The authors have addressed most of my concerns raised in the first submission. They have performed many additional experiments to strengthen the claim that increased luminal expansion decelerates endothelial cell migration, and that vascular basement stiffening reduces luminal expansion to increase branch elongation.

Below are comments to further improve the current status of the manuscript in light of the new data.

First, we would like to thank the reviewer #1 for his/her supportive evaluation of our revised manuscript, which includes additional experiments, and we are also grateful for your and insightful comments. We believe addressing the concerns raised would further improve our manuscript. Thus, we performed the additional experiments and revised the manuscript according to the reviewer #1's suggestions, as described below.

Comment 1:

In the new experiments with Y27632 and Blebbistatin, which the authors used to manipulate lumen formation/expansion, it is unclear when these drugs were added. Is it before or after the initiation of new vessel sprouts/branch? Given that actomyosin-dependent contractility is required for driving membrane protrusions and endothelial cell migration, manipulating this pathway can inhibit vessel sprout/branch formation independent of lumen expansion. In fact, in the representative image shown in Fig. 3j, it appears that Y27632-treatment prevented the formation of new vessel sprouts/branches. How can the authors differentiate the effects of myosin II on endothelial cell migration and lumen expansion?

Response to Comment 1:

Thank you for reviewer #1's important comment regarding the new experiment using Y27632 and Blebbistatin in the previous revised manuscript. Although the timing

of administration of these inhibitors was stated in the Methods section of the previous revised manuscript (*p 38, line 883-888*), it was not stated in the Results section of the text, and I apologize for the confusion regarding the timing of administration. In the experiment, administration of both inhibitors was started one day after introducing ECs to induce angiogenic branch. At this stage, most blood vessels sprouted into the ECM and extended their angiogenic branch to a certain extent. And in most angiogenic branches, this is at a stage where the vascular lumen is just beginning to form. In the present revised manuscript, therefore, we added this information to the Results section of the main text, as shown below.

In addition, as the reviewer #1 also pointed out, treatment with these inhibitors not only expand the lumen and branches, but also directly inhibit directional EC movement, and through this direct action, it may inhibit branch elongation. However, we believe that this possibility cannot be completely ruled out, regardless of when these inhibitors were started. Furthermore, we believe it is not possible to completely distinguish whether the inhibitory effect of the inhibitors on branch elongation is mediated through a direct inhibition of directional EC movement or indirect effect via lumen and branch expansion. In the previous revised manuscript, in order to distinguish between the two as much as possible, we carried out a rescue experiment of the inhibitory effect of inhibitors by limiting lumen and branch expansion through TG treatment. As a result, we found that the inhibition of branch elongation by both inhibitors was significantly suppressed by perivascular stiffening and limitation of lumen and branch expansion by TG treatment (Fig. 3j-m in the previous and present revised manuscript). These results suggested that the effects of both inhibitors reflected, at least in part, the inhibition of branch elongation via lumen and branch expansion, and thereby, in the previous and present revised manuscript, we concluded that “these experimental results suggest that lumen and branch expansion possibly serve a negative regulatory function as regards branch elongation”. However, since a direct effect of the inhibitors on directional EC movement cannot be completely denied, the following statement was added to the present revised manuscript.

Changes in the present revised manuscript were as follows:

1. *p 11, line 248 – 249 → p 11, line 248 – 251*: Therefore, we blocked the RhoA-ROCK-NMII signaling using an inhibitor of ROCK, Y27632, and an inhibitor of NMII, Blebbistatin, from one day after induction of on-chip angiogenesis of ECs, a time point when most angiogenic branches had sprouted into the ECM.
2. *Added p 11, line 259 – 261*: , although the involvement of a direct inhibitory effect of both inhibitors on directional EC movement cannot be ruled out.

Comment 2:

Annotation of Fig. 3 can be improved.

- *Fig. 3e, right panel: since there is no change in pressure in this experimental setting, there is no need to include “load” and “release” in the x-axis as this is rather confusing.*
- *Fig. 3f and g and Fig.3h and i look similar. To differentiate the two, I recommend writing “after pressure load” and “after pressure release”, respectively, above the graphs to differentiate between the two conditions (especially since this information is buried in the very long figure legend).*
- *Fig. 3i, the x-axis legend should be “after pressure release”, not “after pressure loading”.*

Response to Comment 2:

Thank you for your kind and appropriate comments and suggestions regarding the annotations in Figure 3. We agree that the annotations were indeed confusing or incorrect, so following the reviewer #1’s suggestions, we corrected the errors in the Figure annotations in Fig. 3e-i and improved them to make them easier to understand. We also changed the associated annotations in Fig. 3a, b and in Extended Data Fig.6, 7. In addition, we changed the content of the figure legends corresponding to those changes and shortened the whole thing so that it is easier to obtain the necessary information.

The specific changes for each in the present revised manuscript are as follows.

1. In the new Fig. 3a, “ ΔP ” were changed to “Pressure load”, and “Pressure release” with a blue double arrow was added.

2. In the bottom panel of the new Fig. 3d, “Pressure load” was added above the red double arrow and “Pressure release” with a blue arrow was added.
3. In the right panel of the new Fig. 3e, “load” and “release” were removed from the annotation of x-axis, and in the left panel, the annotation of x-axis was simplified.
4. In the new Fig. 3f, “Pressure load” was added above the red double arrow.
5. In the new Fig. 3g, “Pressure load” was added above the graph, and the x-axis annotation “After pressure loading” was changed to “After loading”.
6. In the new Fig. 3h, “Pressure load” was added above a red double arrow, and “Pressure release” with a blue double arrow was added.
7. In the new Fig. 3i, “Pressure release” was added above the graph, and the x-axis annotation “After pressure releasing” was changed to “After releasing”.
8. In the new Extended Data Fig. 6c, the x-axis annotation “After pressure loading” was changed to “After loading”.
9. In the new Extended Data Fig. 6e, the x-axis annotation “After pressure loading” was changed to “After releasing”.
10. In the bottom panel of the new Extended Data Fig. 7a, “Pressure load” was added above the red double arrow, and “Pressure release” with a blue arrow was added.
11. In the right panel of the new Extended Data Fig. 7b, “load” and “release” were removed from the annotation of x-axis, and in the left panel, the annotation of x-axis was simplified.
12. In the new Extended Data Fig. 7c, “Pressure load” was added above the red double arrow.
13. In the new Extended Data Fig. 7d, “Pressure load” was added above the red double arrow, and “Pressure release” with a blue double arrow was added.

Fig. 3

Fig. 3: Influences of interventions for lumen and branch expansion altering tip EC movement and branch elongation

a-i, Physical interventions altering lumen and branch expansions by externally loading intraluminal pressure in on-chip angiogenesis of ECs. **a**, A schematic diagram showing the timeline of the loading and release cycles of intraluminal pressure. **b, c**, Changes of lumens and branches when loaded followed by release of the additional intraluminal pressure. **b**, Representative DIC images. White dotted lines indicate the proximal and distal boundaries of the analyzed luminal region. **c**, Quantification of the changes in luminal area (loading: $n=12$; release: $n=11$, from 3 independent experiments). **d-i**, Time-lapse analyses of on-chip angiogenesis during the intervention. **d**, Representative serial DIC images at the elapsed time indicated at the top before and after loading and the release of additional intraluminal pressure (*top*, See also Supplementary Video 8) and kymograph of branch elongation (*bottom*). In the DIC images, yellow arrows and red lines indicate the trajectory of the tip of the branch and the tip EC displacement, respectively. In the kymograph, red and blue dotted lines indicate time points at which additional intraluminal pressure was loaded and then released, respectively. **e**, Box plots of averaged branch elongation and the individual plots 3 hours before (pressure change (-)) and after (load) the 1st pressure loading and 3 hours after its release (release) (*left*), and the corresponding box and individual plots in the control group without interventions (*right*) (pressure loading and release: $n=21$, control: $n=19$, from 2 independent experiments, See also Extended Data Fig. 6). **f-i**, Kymograph showing the dynamics of tip EC movement for 50 minutes

before and 100 minutes after loading (f) and releasing (h) additional intraluminal pressure (time=0, dotted red line), and the quantification showing in box plots of averaged tip EC movement speed (g, each group: n=13 from 3 independent experiments; i, each group: n=8 from 2 independent experiments).

Extended Data Fig. 6

Hanada et al. Extended Data Figure 6

Extended Data Fig. 6: Time-lapse imaging of on-chip angiogenesis of ECs as control with no interventions

b-e, Quantification of tip EC movement. **b**, Kymograph showing the dynamics in tip EC movement in the same timeline, 50 minutes before and 100 minutes after loading additional intraluminal pressure in the intervention group (time=0, dotted red line). **c**, Box plots of averaged tip EC movement speeds in the same timeline, 50 minutes before, 50 minutes and 50-100 minutes after the pressure load in the intervention group (each group: n=12 from 2 independent experiments). **d**, Kymograph showing the dynamics in tip EC movement in the same timeline, 50 minutes before and 100 minutes after the release of additional intraluminal pressure in the intervention group (time=0, dotted red line).

Extended Data Fig. 7

Hanada et al. Extended Data Figure 7

Extended Data Fig. 7: Time-lapse imaging of on-chip angiogenesis of ECs at the 2nd intervention cycle and the corresponding analyses

The second cycle of physical intervention for lumen and branch expansions by externally loading intraluminal pressure. **a**, Representative serial DIC images at the elapsed time indicated at the top showing the dynamics in branch elongation, lumen development and tip EC movement before and

after loading and then releasing additional intraluminal pressure (*top*). See also Supplementary Video 10) and kymograph of branch elongation (*bottom*). In the DIC images, red lines and yellow arrows indicate the trajectory of tip EC displacement and the tip of the branch, respectively. In the kymograph, red and blue dotted lines indicate time points at which additional intraluminal pressure was loaded and released, respectively. Scale Bars: 20 μm . **b.** Box plots of averaged branch elongation 3 hours before (pressure change (-)) and after (load) the 2nd pressure loading and 3 hours after the release (release), merged with individual plots showing the branch elongation changes in response to loading and then the release of intraluminal pressure (pressure load and release: n=11, control: n=6, from 1-2 independent experiments). All box plots show the interquartile range, with the middle line defining the median, and whiskers show the minimum and maximum values, excluding outliers. Outlier values are defined as being 1.5 times the interquartile range above and below the third and first quartile, respectively. **c.** Kymograph showing the dynamics in tip EC movement 50 minutes before and 100 minutes after loading additional intraluminal pressure (time=0, dotted red line). **d.** Kymograph showing the dynamics in tip EC movement 50 minutes before and 100 minutes after the release of additional intraluminal pressure (time=0, dotted red line). Wilcoxon signed rank test with a Bonferroni correction (**b**).

Comment 3:

Figure 7a. Representative images of collagen IV (COL-IV) coverage by immunostaining in CTLsi EC and CTLsi pericytes look different, with decreased staining observed in CTLsi pericytes. Given that these are control experiments, shouldn't the degree of COL-IV be similar? However, quantification of coverage ratio of COL-IV as shown in Fig. 7c shows similar coverage between the two controls. Perhaps a better representative image(s) should be used.

Response to Comment 3:

Thank you for your kind comment and suggestions. As the reviewer #1 pointed out, we agree that we selected inappropriate images that gave the impression that vascular basement membrane deposition of Col-IV was reduced in the CTL si-treated pericyte group in Fig. 7a compared to the CTL si-treated EC group in the previous revised manuscript. The image used in Fig. 7a was taken with a fluorescence microscope. The image of pericytes treated with CTL si were in focus at the tip of the angiogenic sprout but out of focus at the base of the angiogenic branch. This is the reason why deposition of Col-IV in the vascular basement membrane appeared to have decrease in the CTL si-treated pericyte group of the previous Fig. 7a. On the other hand, to quantify Col-IV coverage, we used images taken with a confocal laser microscope, so that all samples could be analyzed without any focal plane issues.

Therefore, in the present revised manuscript, we replaced the image of CTL si-treated pericyte in Fig.7a with a more appropriate one.

Fig. 7a

Fig. 7: Functional involvement of Col-IV deposition on VBM in on-chip angiogenic morphogenesis with pericytes

a, b, Representative fluorescent images (a) and magnified confocal z-projection images (b) on Day 3, with (COL4 si) and without (CTL si) knock-down of both COL4A1 and COL4A2 in either ECs (K.D. EC, left) or pericytes (K.D. Pericyte, right) using siRNAs.

Comment 4:

There is inconsistency in the representative images used to demonstrate vessel morphology after COL4si in EC in Fig. 7a and Fig. 7b. In Fig. 7a, there appears to be very few vessel sprouts/branches that form and in cases where there are vessel branches, they are short. But in Fig. 7b, the representative vessel branch in COL4si EC condition is long with many endothelial cells (as shown by DAPI staining). Which is the more representative image?

Response to Comment 4:

Thank you for your very constructive feedback, which will enable us to present our data better. As the reviewer #1 pointed out, when EC are treated with Col4 si, most vascular angiogenic branches become shorter. In some cases, as shown in Fig. 7f of the previous and present revised manuscript, many branches regressed and can no longer be

recognized as branches, which we believe is one of the reasons why the number of angiogenic branches appears to be low at first glance. We agree that, in the previous revised manuscript, we selected the most elongated branches among these and therefore provided inappropriate information regarding the changes in angiogenic branches composed of Col4 si-treated ECs.

Therefore, in the present revised manuscript we replaced the images ones showing more typical angiogenic branches formed by Col4 si-treated ECs.

Fig. 7b

Fig. 7: Functional involvement of Col-IV deposition on VBM in on-chip angiogenic morphogenesis with pericytes

a, b, Representative fluorescent images (**a**) and magnified confocal z-projection images (**b**) on Day 3, with (*COL4* si) and without (*CTL* si) knock-down of both *COL4A1* and *COL4A2* in either ECs (*K.D. EC, left*) or pericytes (*K.D. Pericyte, right*) using siRNAs.

Comment 5:

The experiment where the authors attempt to rescue ECM stiffness in COL4A1/A2 DKD ECs by TG treatment is very nice. Did the authors measure Creep compliance in these experiments to demonstrate that stiffness was successfully increased and to complement the existing data (Fig. 7h – l)?

Response to Comment 5:

Thank you for your kind comments and helpful suggestions to improve the quality of the manuscript. In the previous revised manuscript, in an experiment to rescue the changes in angiogenic branches using Col4 si-treated ECs by TG treatment of ECM, no experiment was conducted to confirm the stiffness around the angiogenic branches using the PTM method. As the reviewer #1 commented, we agree that confirming whether TG treatment really increases ECM stiffness would complement the current data, so we conducted in the present revised manuscript. As expected, when Col4 si-treated ECs were used, the ECM surrounding the angiogenic branches became softer, and in addition, it was shown that the stiffness of the ECM was indeed increased by treating the ECM with TG.

Therefore, in this revised manuscript, the PTM analysis data was added as Fig. 7m. Accordingly, the contents were added to the main text and figure legend. Specifically, it is as follows.

1. *p 17, line 400 – 401* → *p 17, line 402 – 403*: completely, by using TG-treated stiffened ECM (confirmed by the PTM analysis), with no changes in Col-IV deposition on the VBM (Fig. 7h-m).

Fig. 7m

Fig. 7: Functional involvement of Col-IV deposition on VBM in on-chip angiogenic morphogenesis with pericytes

h-m. Rescue experiments of *COL4* si in ECs by stiffening the gel with TG treatment. **g.** Representative fluorescent images showing the morphogenetic changes in angiogenic branches of *COL4A1* and *COL4A2* double knock-down ECs and pericytes and the rescue effects of TG treatment. **i, j.** Quantification of the rescue effects by TG treatment, shown in box plots of branch length (**i**) and Tip expandability (**j**) (for **i**, *CTL si*: n=53; *COL4 si*: n=53; *COL4 si* + TG: n=64, from 2 independent experiments, for **j**, *CTL si*: n=53 from; *COL4 si*: n=53; *COL4 si* + TG: n=64, from 2 independent experiments).

Comment 6:

In the revised manuscript, the authors show that increasing ECM stiffness by TG treatment (Fig. 8 a, b, e f, g) as well as co-culturing with pericytes increase CIP4 and Arp2/3 complex localization at the leading edge of tip cells, leading to cell migration. Although the authors also correlated CIP4 and Arp2/3 complex number with the presence of lumen, can the authors also perform a pressure loading experiment (similar to that in the new Fig. 3) with or without TG treatment and examine CIP4 and Arp2/3 localization? This would convincingly demonstrate that the two mechanical cues (luminal pressure and ECM stiffness) converge molecularly at the level of Arp2/3 activity to coordinate lumen expansion and vessel elongation.

Response to Comment 6:

Thank you for your constructive comments regarding the new Figure 8 added to the previous revised manuscript. As you pointed out, by showing that the localization

of Arp2/3 complexes and BAR proteins at leading edge of angiogenic branches disappears as the vascular lumen and branches expand due to additional intraluminal pressure load, and that this disappearance is suppressed by perivascular stiffening due to TG treatment, we believed that we more convincingly demonstrate that the activity of Arp2/3 molecules is involved in the control of branch elongation by mediating the action of lumen and branch expansion, which is determined by the balance between intraluminal pressure and perivascular stiffness. Therefore, in the present revised manuscript, we conducted additional experiments to demonstrate this. Although this was not a similar method of dynamically applying intraluminal pressure into the vascular lumen as shown in Figure 3 of the previous revised manuscript, we loaded a similar hydrostatic pressure of approximately 40 Pa by placing a capillary glass filled with an appropriate amount of culture medium in the device channel on the vascular lumen (Extended Data Fig. 17a), as we previously reported in our manuscript (*Nat Commun* 13: 2594, 2022), and then observed in the change of the localization pattern of BAR protein and Arp2/3 complex at the leading edge of the angiogenic branch. Furthermore, we investigated how its localization change was further affected by TG treatment of the ECM. As a result, the localization of BAR protein and Arp2/3 complex at the leading edge of angiogenic branches disappeared as the lumen and branches expanded due to intraluminal pressure load, however, as expected, this disappearance was suppressed by hardening the perivascular area with TG treatment and thereby limiting the lumen and branch expansion. (new Fig. 8h-j in the present revised manuscript).

Therefore, in the present revised manuscript, the data was added as Fig. 8h-j and Extended Data Fig. 17, and the relevant information was added to the methods, results, and Figure legend in the main text as follows. Also, Fig. 8f of the previous revised manuscript was moved to new Extended Data Fig. 16, and previous Fig. 8i-l were moved to the new Fig. 9a-d.

1. *p 18, line 421 – 422* → *p 18, line 424 – 426*: both, when lumen and branch expansions were restricted intentionally by stiffening the surrounding ECM with TG treatment, when additional intraluminal pressure was loaded, and both.
2. *p 19, line 433 – 439* → *p 19, line 437 – 447*: well as TOCA1 tagged with EGFP in the N-terminus (EGFP-TOCA1) were observed (Fig. 8e-g, and Extended Data Fig.

15); CIP4 and EGFP-TOCA1, which were colocalized with Arp2/3 complexes (Extended Data Fig. 15b, c and 16), were more dominantly localized at the leading front of tip ECs when the ECM contained TG (Fig. 8f and Extended Data Fig. 15d), and, in more lumen-dominated branches, smaller numbers of CIP4 and EGFP-TOCA1 were localized at the leading front of tip ECs ($r=-0.5547$, Fig. 8g and Extended Data Fig. 15e). Furthermore, when additional intraluminal pressure was loaded, the majority of the Arp2/3 complex and F-BAR protein disappeared from the leading front of tip ECs in conjunction with lumen and branch expansion, resulting in retarded branch elongation, which was inhibited by stiffening the ECM with the TG treatment (Fig. 8h-j and Extended Data Fig 16b).

3. *p 19, line 448 – 451* → *p 19, line 457 – 459*: compared to those without pericytes (Fig. 9a-c), in conjunction with less lumen occupation adjacent to the tip ECs in the branch ($r=-0.4638$ for CIP4, $r=-0.4417$ for Arp2/3 complex, Fig. 9d).
4. Added *p 42, line 985 – 990*: In experiments to evaluate the changes in localization patterns of the Arp2/3 complex and F-BAR protein by intraluminal pressure loading, approximately 40 Pa of hydrostatic pressure was loaded by placing capillary glass tubes (Hirshman) filled with the media (approximately 11mm height from the center of vascular lumen) at the outlet and the inlet of channel 4 (Extended Data Fig. 17a), similarly as we previously reported⁹.

Fig. 8e-j

Hanada et al. Figure 8

Fig. 8: Loss in localization of F-BAR protein clusters and Arp2/3 complexes at the leading front of tip ECs in relation to lumen development

e-g. Relationships between CIP4 localization patterns around the tips of branches and the nearby lumen development in on-chip angiogenesis without (-) or with (+) TG treatment of the gel. **e.** Representative z-projection confocal images, showing different localization patterns of CIP4. Areas within squares bounded by white dots are shown magnified on the right. **f, g.** Quantification of CIP4 localization (**f**) and distribution plot (**g**) showing relationship between CIP4 localization and lumen development (**f, g**; TG (-): n=42 branches; TG (+): n=26 branches, from 3 independent experiments).

h-i. Changes in localization patterns of CIP4 and Arp2/3 complexes around the tip of on-chip angiogenic branches without or with TG treatment of the gel by intraluminal pressure loading. **h.** Representative z-projection confocal images, showing different localization patterns of CIP4 and Arp2/3 complexes. Areas within squares bounded by white dots are shown magnified on the right. **i, j.** Quantification of CIP4 and Arp2/3 complex localization (**i**) and distribution plot (**j**) showing relationship between CIP4 or Arp2/3 complex localization and lumen development (**i, j**; for CIP4, Pressure (P) (-)/TG (-): n=25 branches; P (+)/TG (-): n=20 branches; P (+)/TG (+): n=22 branches,

from 3 independent experiments, for Arp2/3, P (-)/TG (-): n=25 branches; P (+)/TG (-): n=25 branches; P (+)/TG (+): n=25 branches, from 3 independent experiments). All box plots show the interquartile range, with the middle line defining the median, and whiskers show the minimum and maximum values, excluding outliers (b, f and i). Outlier values are defined as being 1.5 times the interquartile range above and below the third and first quartile, respectively. Scale bars: 20 μ m. Mann-Whitney U test (b, f) and Mann-Whitney U test with a Bonferroni correction (i).

Fig.9

Hanada et al. Figure 9

Fig. 9: Localization patterns of F-BAR protein clusters and Arp2/3 complexes at the leading front of tip ECs in the presence of pericyte

a-d, Relationship between CIP4 and Arp2/3 complex localization patterns around the tips of branches and the nearby lumen development in on-chip angiogenesis in the absence (PC (-)) or the presence (PC (+)) of pericytes. **a**, **c**, Representative z-projection confocal images showing colocalization of CIP4 with Arp2/3 complexes (**a**), and images showing relationship between Arp2/3 complex localization pattern and lumen development (**b**). Regions indicated in squares bounded by white dots were magnified on the right in **a** and **b**, and the region within the square bounded by yellow dots is shown magnified in the *right panel* of **b**. **c**, **d**, Quantification of CIP4 and Arp2/3 complexes localization (c) and distribution plot showing relationship between CIP4 or Arp2/3 complexes localization around and lumen development (d) (c, d, for CIP4, PC (-): n=37 branches; PC (+): n=25 branches, for Arp2/3, PC (-): n=45 branches; PC (+): n=44 branches, from 3 independent experiments). All box plots show the interquartile range, with the middle line defining the median, and whiskers show the minimum and maximum values, excluding outliers (c). Outlier values are defined as being 1.5 times the interquartile range above and below the third and first quartile, respectively. Scale bars: 20 μ m. Mann-Whitney U test (c).

Extended Data Fig. 16

Hanada et al. Extended Data Figure 16

Extended Data Fig. 16: Colocalization of CIP4 with Arp2/3 complexes at the leading edge of on-chip angiogenic branch

Line scan profiles of fluorescent intensities of CIP4 and ARPC2 along the yellow dotted line indicated in the confocal x-y slice image (left), showing colocalization of CIP4 with Arp2/3 complexes (right).

Extended Data Fig. 17

Hanada et al. Extended Data Figure 17

Extended Data Fig. 17: Effects of intraluminal pressure load on branch elongation in on-chip angiogenesis with and without TG treatment of the gel

a, Intraluminal pressure loading into luminal space of the angiogenic branches by placing capillaries filled with culture media (11 mm in height). **b, c**, Changes in branch elongation of on-chip angiogenesis without or with TG treatment of the gel by intraluminal pressure loading. **b**, Representative DIC images. Scale bars: 100 μm . **c**, Quantification of branch length (Pressure (P) (-)/TG (-): n=90 branches; P (+)/TG (-): n=82 branches; P (+)/TG (+): n=94 branches, from 3 independent experiments). All box plots show the interquartile range, with the middle line defining the median, and whiskers show the minimum and maximum values, excluding outliers. Outlier values are defined as being 1.5 times the interquartile range above and below the third and first quartile, respectively. Mann-Whitney U test with a Bonferroni correction.

Comment 7:

The length of all figures legends is far too long so much so that it does not serve the purpose in aiding the reader to interpret the data since it is difficult to find the relevant information. The figure legends should be more concise and avoid repetitive phrases.

Response to Comment 7:

Thank you for your kind comment. We agree with the reviewer #1 that the figure legends are very long and it is very difficult to find the necessary information in them. Therefore, we made an effort to rewrite the legend more concisely, taking particular care to avoid repetitive phrases. The same applies to the legends in the Extended Data Figure, which was similarly corrected in the present revised manuscript. Listing the corrections below would be too extensive, so we did not extract the changes and listed them below. We would appreciate it if you could take the time to look at the main text and Extended Data Figures directly. All changes are highlighted in yellow.

Comment 8:

The word 'rate' on page 13, line 303 should be changed to 'ratio'.

Response to Comment 8:

We are very sorry. Last time, we received a comment that it was desirable to change “rate” to “ratio”, and we thought we made the necessary changes to all parts of the manuscript, but we missed this change. We made the change in the text and the related Figure 5 legend of the present revised manuscript as follows.

1. p 13, line 303 → p 13, line 305: showed the ratio of tip ECs with distal localization of the Golgi apparatus to be reduced
2. p 68, line 1516 → p 66, line 1459: Quantification of the ratio of tip ECs having the Golgi apparatus

Comment 9:

Extended Dat Fig. 12a show an image of laminin staining around vessels in the presence (+) and absence (-) of pericytes. In the representative image of laminin staining of vessels without pericytes (middle image of lower row), it looks like UEA-1 Lectin staining, not laminin staining. Please check.

Response to Comment 9:

We are very sorry. As the reviewer #1 pointed out, in Extended Data Fig. 12 of the previous revised manuscript, we made a simple mistake and presented images of UEA-1 lectin staining instead of laminin staining. Therefore, in the present revised manuscript, we replaced the incorrect image of UEA-1 lectin staining with the correct image of laminin staining.

Extended Data Fig. 12a

Extended Data Fig. 12: Laminin deposition on VBM around the tip at the site of *de novo* vascular lumen formation in on-chip angiogenesis and in murine retinal angiogenesis
a, Representative confocal z-projection images (z-projection) and confocal *x-y* slice images at specific *z*-positions (slice) of on-chip angiogenic branches with and without pericyte coculture.

Reviewer #3 (Remarks to the Author):

This is a brief assessment of the manuscript titled “Biomechanical Control of Vascular Morphogenesis by the Surrounding Stiffness” by Hanada and colleagues. My primary focus is on the issues raised by Reviewer #2, who was unavailable for the re-review of the revised manuscript.

In my assessment, the authors have conducted a thorough and substantial revision. The inclusion of additional experimental results has effectively addressed most of the concerns raised by Reviewer #2. In particular, the new data on the interplay between matrix stiffness, endothelial sprout extension, and lumen formation provide a stronger foundation for the pericyte-related findings presented later in the manuscript. These improvements have also resolved the disconnect between the two halves of the original submission, as highlighted in Reviewer #2’s comments.

Thank you for reviewing our revised manuscript on behalf of previous reviewer #2. We would also like to express our sincere gratitude for your evaluation of the previous revised manuscript, which was significantly revised based on numerous additional experiments made in accordance with the previous reviewer #2’s suggestions.

Another major concern raised by Reviewer #2—the correlative nature of many of the findings—has been partially addressed. The authors have made significant efforts to tackle this issue, and it is important to acknowledge the additional data and the insights gained from it. While a conceptual gap remains between matrix stiffness/hydrodynamic pressure and the downstream biochemical signals, resolving this within the scope of a revision would be challenging.

As current reviewer #3 has pointed out, we have no particular disagreement with the fact that the revised manuscript does not fully demonstrate the molecular mechanism by which the lumen and branch expansion caused by increased intraluminal pressure impairs the directional movement of ECs, thereby suppressing branch elongation. We believe that this molecular mechanism is an important issue that needs to be addressed in

the future, and we stated this in the discussion of the previous as well as present revised manuscript (*p25, line 595 – p26, line 608* in the previous revised manuscript and *p25, line 581 – 594* in the present revised manuscript).

Other concerns expressed by Reviewer #2 have been adequately addressed through revisions to the manuscript, as outlined in the (admittedly lengthy) rebuttal letter.

Overall, my primary remaining criticism is that the manuscript is not particularly easy to read. The figure legends and Discussion section are overly lengthy and would benefit from trimming and revision to enhance clarity.

Thank you for your kind comment. We agree with the reviewer #2 that the figure legends and discussion are very long. Therefore, we made an effort to rewrite the legend and discussion more concisely, taking particular care to avoid repetitive phrases. The same applies to the legends in the Extended Data Figure, which was similarly corrected in the present revised manuscript. Listing the corrections below would be too extensive, so we did not extract the changes and listed them below. We would appreciate it if you could take the time to look at the main text and Extended Data Figures directly. All changes are highlighted in yellow.

Replies to the reviewer's comments

Reviewer #1 (Remarks to the Author):

I would like to thank the authors for their efforts in addressing all my concerns and performing the experiments that I suggested after the first round of revision. The manuscript is now in a much-improved state for publication.

First, we would like to thank the reviewer #1 for his/her supportive evaluation and heartwarming words on the revised manuscript during the revision process. We were encouraged by the reviewer's comments and were able to improve our manuscript.

Comment 1:

While reading the manuscript, I found two very minor issues/questions that should be addressed:

- There is a typo in the y-axis legend of Fig. 1h. "chnage" -> "change".

Response to Comment 1:

Thank you very much for your suggestion. We changed the miss typing from "change" to "change" in the legend of Fig. 1h.

Comment 2:

- Fig. 9a and Fig. 9b: does one panel show the presence of pericyte and the other the

Response to Comment 2:

Thank you very much for your suggestion. In Figure 9a and Figure 9b, we only showed data when pericytes were present, so we noted this in the Figure legend of the revised manuscript.

Reviewer #3 (Remarks to the Author):

The authors have addressed my questions. Moreover, the revised manuscript is clearly improved and it is easier to follow the flow of the data. However, the authors might want to consider further trimming of the discussion, which is still rather long.

Overall, the article will enhance our conceptual understanding of the processes controlling endothelial sprouting and thereby makes an important contribution to the field.

Thank you very much for the reviewer #3 for his/her kind evaluation and heartwarming words on the revised manuscript. Based on the reviewer #3's suggestions, we have made every effort to edit the discussion section to make it more concise and clearer.